# Century Long Reconstruction of Gridded Phosphorus Surplus Across Europe (1850 – 2019)

Masooma Batool[1], Fanny J. Sarrazin[1,2], and Rohini Kumar[1]

[1]UFZ-Helmholtz Centre for Environmental Research, Department of Computational Hydrosystems, Leipzig, Germany
[2]Université Paris-Saclay, INRAE, UR HYCAR, 92160 Antony, France

**Correspondence:** Masooma Batool (masooma.batool@ufz.de) and Rohini Kumar (rohini.kumar@ufz.de)

**Abstract.** Phosphorus (P) surplus in soils significantly contributes to the eutrophication and degradation of water quality in surface waters worldwide. Despite extensive European regulations, elevated P levels persist in many water bodies across the continent. Long-term annual data on soil P surplus (the difference between P inputs and outputs) are essential to understand these levels and guide future management strategies. This study reconstructs and analyzes the annual long-term P surplus for
both agricultural and non-agricultural soils from diffuse sources across Europe at a 5 arcmin ($\approx$ 10 km at the equator) spatial resolution from 1850 to 2019. The dataset includes 48 P surplus estimates that account for uncertainties arising from different methodological choices and coefficients in major components of the P surplus. Our results indicate substantial changes in P surplus magnitude over the past 100 years, underscoring the importance of understanding a long-term P surplus. Specifically, the total P surplus across the EU-27 has tripled over 170 years, from 1.19 ($\pm 0.28$) $kg\ ha^{-1}$ of physical area in 1850 to around
2.48 ($\pm 0.97$) $kg\ ha^{-1}$ of physical area $yr^{-1}$ in recent years. We evaluated the plausibility and consistency of our P surplus estimates by comparing them with existing studies and identified potential areas for further improvement. Notably, our dataset supports aggregation at various spatial scales, aiding in the development of targeted strategies to address soil and water quality issues related to P. The P surplus reconstructed dataset is available at https://doi.org/10.5281/zenodo.11351028 (Batool et al., 2024).

## 1  Introduction

Phosphorus (P), an essential nutrient for plant growth, presents a paradox: while agricultural soils contain large P reserves, these are largely inaccessible to plants, necessitating external inputs in organic or inorganic forms (Panagos et al., 2022a; Wang et al., 2015; Zou et al., 2022). Since the 1920s, agricultural intensification in Europe, characterized by increased use of P mineral fertilizers, has resulted in significant P accumulation in soils (Einarsson et al., 2020). This accumulation exceeds
the immediate needs of plants, leading to excess P or P surplus (the difference between P inputs and outputs) with significant environmental impacts, including water quality degradation, harm to human health, and threats to biodiversity (Muntwyler et al., 2024; Wu et al., 2022; Guejjoud et al., 2023; Brownlie et al., 2022; Schoumans et al., 2015). Excessive P inputs to the environment are recognized as one of the greatest threats to planetary boundaries, underlining the urgent need to reduce them (Muntwyler et al., 2024; Steffen et al., 2015). In response, the European Union (EU) has enacted directives aimed at P surplus

mitigation, including the Water Framework Directive (Directive 2000/60/EC) (European Commission, 2000a), the Urban Waste Water Treatment Directive (European Union, 1991), and the recent Farm to Fork Strategy (European Commission) as part of the EU Green Deal (European Commission, 2019). These initiatives face the challenge of P legacies that is, accumulated P surplus in soil that is not immediately available for plant uptake and that is responsible for high P levels in the environment despite reductions in P inputs. P legacies not only increase the risk of eutrophication but also represent a significant untapped secondary P resource that could reduce reliance on primary P mineral fertilizers (Pratt and El Hanandeh, 2023; Brownlie et al., 2022). The finite and unevenly distributed nature of geological P deposits, with key producers like China, the USA, Russia, and Morocco generating 60% of global output (Ritchie et al., 2022; Schoumans et al., 2015), further underscores the importance of optimizing the use of legacy P resources.

A comprehensive understanding of the long-term P surplus is therefore critical to understanding these P legacies, which is essential for improving future land and water management practices. Existing databases covering the European domain provide P budgets (the difference between P inputs and outputs), but are often constrained by limited temporal coverage or low spatial resolution and focus only on agricultural areas comprising cropland and/or pasture. Specifically, FAOSTAT (Food and Agriculture Organization Corporate Statistical Database) (Ludemann et al., 2023) and Zou et al. (2022) offer global annual P budgets for croplands from 1961-2020 across over 200 countries, assessing P budgets as the difference between P inputs (mineral fertilizer, animal manure, seeds) and P outputs (crop P removal). Ringeval et al. (2024) enhance this analysis by providing a granular global dataset of agricultural P flows from 1900 to 2018 at a 0.5° gridded spatial resolution. At the European level, Muntwyler et al. (2024) offer current (2011-2019 average) and future projections (2020-2029 and 2040-2049) of P budgets in agricultural soils at a higher spatial resolution of 1 $km^2$ by employing a process-based biogeochemical model (DayCent). Furthermore, Panagos et al. (2022a) provide the agricultural P budget for the EU27 and the UK, averaging 2011-2019 data at NUTS (Nomenclature of Territorial Units for Statistics) 2 (regional scale) and country scale, while Einarsson et al. (2020) present the agricultural P budget for the EU28 for 2013 at the NUTS-2 level based on empirical methods. Additionally, there are subnational P budgets available for some countries, such as France (Guejjoud et al., 2023), Poland (Kopiński et al., 2006), Sweden (Bergström et al., 2015), and Turkey (Özbek, 2014). Summaries of P budgets and their components in existing studies are provided in Table 2 and Table 3, which also shows that different databases consider different components of the P surplus budget.

Nutrient budgets tend to have large uncertainties (Zhang et al., 2021; Ludemann et al., 2023). Uncertainties in P budgets can stem from limited knowledge about the distribution of mineral fertilizers and animal manure on cropland and pasture and about the P removal coefficients, among other factors (Ludemann et al., 2023). As a result, the different studies of Table 2 and Table 3 adopted different schemes to allocate mineral fertilizer and animal manure to cropland and different coefficient values. While some studies explicitly consider uncertainties (e.g., Guejjoud et al. (2023); Antikainen et al. (2008); Lun et al. (2018); Muntwyler et al. (2024); Ringeval et al. (2024); Ludemann et al. (2023); Panagos et al. (2022b), listed in Tables 2 and 3), the majority do not. Ignoring this uncertainty could lead to inaccurate assessments of P dynamics and, consequently, flawed policy recommendations (Oenema et al., 2003). Recent studies, such as Guejjoud et al. (2023); Ringeval et al. (2024), Sarrazin et al. (2024) and Zhang et al. (2021) underscore the need for uncertainty-aware nutrient datasets to support quantification of

nutrient budgets and robust water quality assessments. Additionally, previous studies (Zou et al., 2022; Ludemann et al., 2023) developing long-term, country-scale nutrient budgets did not consider P inputs from atmospheric deposition and chemical weathering and excluded fodder crops (e.g., alfalfa, green maize), potentially underestimating nutrient removal in various European countries (Panagos et al., 2022a).

To address these limitations, we present here a database of yearly long-term P budgets, termed "P surplus" - defined as the difference between P inputs (mineral fertilizer, animal manure, atmospheric deposition and chemical weathering) and P removals (crop and pasture removals), covering both agricultural (cropland and pastures) and non-agricultural soils at a 5 arcmin (1/12°; approximately 10 km at the equator) spatial resolution from 1850 to 2019 across Europe, focusing only on diffuse sources. Our dataset quantifies uncertainties arising from methodological choices in major P surplus components, such as mineral fertilizer and animal manure distribution to cropland and pasture and crop removal coefficients. The dataset integrates information at various spatial levels (country and grid level) to construct the different P surplus components. The importance of constructing a long-term dataset is underscored by the large changes in P surplus magnitude over the past 100 years. Additionally, we account for P surplus in non-agricultural areas, which, although decreased threefold over a century (from 15% around 1850 to 5% in recent years) across the EU-28 from our estimates, might still play an important role in countries with higher proportions of non-agricultural areas, such as those in Northern Europe. We therefore specifically integrate atmospheric deposition and chemical weathering to provide a more complete picture of P surplus. Our dataset characterizes soil surplus P budget, analogous to the nitrogen (N) surplus budget at the soil surface (Oenema et al., 2003). With the gridded database provided here, we provide the flexibility to aggregate the P surplus at any spatial scale relevant. This flexibility supports subnational studies and transboundary analyses of river basins where nutrient dynamics and management practices cross political boundaries and are needed for the design of land and water management strategies. We also investigate the consistency and plausibility of our P surplus estimates by comparing them against existing P budget datasets (Ludemann et al., 2023; Zou et al., 2022; Lun et al., 2018; Guejjoud et al., 2023; DEFRA, 2022; Verbič and Sušin, 2022; Eurostat, 2024; Einarsson et al., 2020). We further discuss possible avenues for a comprehensive characterization of uncertainty in the P surplus, with our reconstruction methodology paving the way for exploring alternative assumptions in P surplus estimates. Notably, our P surplus dataset has been developed consistently with the recently published long-term, N surplus dataset (Batool et al., 2022), enabling joint analysis of N and P budgets across Europe, thereby facilitating holistic nutrient management studies.

## 2   Methods and Datasets

Here, we describe our approach to reconstruct a long-term yearly time series for the components of P surplus on a 5 arcmin grid from 1850 to 2019 (refer to the detailed workflow in Figure 1). We gathered and standardized a variety of databases covering different periods (1850–1960, 1961–2019, and the year 2000), at varying intervals (snapshots, decadal, yearly), and across different spatial scales (gridded data, national averages, global level trends). We ensured that the uncertainties arising from methodological differences and coefficient values were incorporated for key components of P surplus. Consequently, we generated 48 gridded datasets of P surplus by combining two fertilizer estimates, six animal manure estimates, and two

cropland and two pasture P removal estimates. Supplementary Table S2 and S3 outlines the specific combinations of these estimates used to create the 48 unique P surplus datasets.

We used FAOSTAT (FAOSTAT, 2024) country-level data, which provides a comprehensive dataset for various variables, such as animal manure and mineral fertilizer and covers the period during 1961-2019 worldwide. Additionally, we incorporated the recent dataset from Ludemann et al. (2023), which spans 1961 to 2019 and includes information on the allocation of mineral fertilizer and animal manure to cropland. While FAOSTAT provides data on total agricultural areas using national statistics, Ludemann et al. (2023) derives cropland estimates by integrating FAOSTAT with Eurostat and various national datasets specific to European countries. We also considered the distribution patterns of mineral fertilizer and animal manure to cropland from Zou et al. (2022), which, while using major datasets from FAOSTAT, offers detailed insights into the application of P fertilizer across different crop types. Additionally, we employed the previously reconstructed gridded database by Batool et al. (2022) to account for land use (both agricultural and non-agricultural), crop-specific harvested areas, and crop production for both fodder and non-fodder crops.

In the following sections, we first outline the definition of P surplus in both agricultural and non-agricultural soils. Next, we present a summary of the methodology used to reconstruct the land use types, including agricultural land, namely cropland and pasture, and non-agricultural land, including non-vegetated areas, semi-natural vegetation, forest, and urban areas. Crop-specific harvested areas for non-fodder and fodder crops are also defined. We refer to Batool et al. (2022) for detailed methodologies. Finally, we describe the steps employed to reconstruct P inputs, including fertilizer, manure, atmospheric deposition, and chemical weathering, and P outputs, focusing on P removal from cropland and pastures. For the clarity and ease of reference, all variables used in the equations in the Methods section are listed in the Table A1 at the end of the manuscript together with their descriptions and units.

## 2.1 P surplus

We calculated P surplus as the difference between P inputs and P outputs (Ludemann et al., 2023; Zou et al., 2022). The total P surplus is composed of contributions from both agricultural (cropland and pasture) and non-agricultural areas (semi-natural vegetation, forest, non-vegetated regions, and urban areas), as described in equation (1) (with all variables expressed in $kg\ ha^{-1}$ of physical area $yr^{-1}$):

$$Surp_{\text{soil}}(i,y) = Surp_{\text{agri}}(i,y) + Surp_{\text{NonAgri}}(i,y) \tag{1}$$

Here, $i$ represents the grid cell; $y$ indicates the year; $Surp_{\text{soil}}$ is the total P surplus; $Surp_{\text{agri}}$ is the P surplus from agricultural areas; and $Surp_{\text{NonAgri}}$ is the P surplus from non-agricultural areas. The following sections elaborate on the components of P surplus.

### 2.1.1 P surplus in agricultural soils

P surplus in agricultural soils includes the surplus from cropland ($Surp_{\text{cr}}$) and pasture ($Surp_{\text{past}}$). The surplus in these areas is determined by the difference between inputs to cropland and pasture ($Inp_{\text{cr}}$ and $Inp_{\text{past}}$) from mineral fertilizers ($FERT_{\text{cr}}$ and

$FERT_{\text{past}}$), animal manure ($MAN_{\text{cr}}$ and $MAN_{\text{past}}$), chemical weathering ($CW_{\text{cr}}$ and $CW_{\text{past}}$), and atmospheric deposition ($DEP_{\text{cr}}$ and $DEP_{\text{past}}$), against outputs from harvested crops ($Rem_{\text{cr}}$) and animal grazing and cutting of grass($Rem_{\text{past}}$). These relationships are represented by equations (2–8) (all variables are in $kg\ ha^{-1}$ of physical area $yr^{-1}$):

$$Surp_{\text{agri}}(i,y) = Surp_{\text{cr}}(i,y) + Surp_{\text{past}}(i,y) \tag{2}$$

$$Surp_{\text{cr}}(i,y) = Inp_{\text{cr}}(i,y) - Out_{\text{cr}}(i,y) \tag{3}$$

$$Inp_{\text{cr}}(i,y) = FERT_{\text{cr}}(i,y) + MAN_{\text{cr}}(i,y) + DEP_{\text{cr}}(i,y) + CW_{\text{cr}}(i,y) \tag{4}$$

$$Out_{\text{cr}}(i,y) = Rem_{\text{cr}}(i,y) \tag{5}$$

$$Surp_{\text{past}}(i,y) = Inp_{\text{past}}(i,y) - Out_{\text{past}}(i,y) \tag{6}$$

$$Inp_{\text{past}}(i,y) = FERT_{\text{past}}(i,y) + MAN_{\text{past}}(i,y) + DEP_{\text{past}}(i,y) + CW_{\text{past}}(i,y) \tag{7}$$

$$Out_{\text{past}}(i,y) = Rem_{\text{past}}(i,y) \tag{8}$$

### 2.1.2  P surplus in non-agricultural soils

P surplus in non-agricultural soils ($Surp_{\text{NonAgri}}$) includes contributions from forests ($Surp_{\text{For}}$), semi-natural vegetation ($Surp_{\text{NatVeg}}$), urban areas ($Surp_{\text{Urban}}$), and non-vegetated regions ($Surp_{\text{NonVeg}}$). In forested areas, P surplus is calculated from inputs such as chemical weathering ($CW_{\text{For}}$) and atmospheric deposition ($DEP_{\text{For}}$). For semi-natural vegetation, P surplus is derived from inputs through atmospheric deposition ($DEP_{\text{NatVeg}}$) and chemical weathering ($CW_{\text{NatVeg}}$). In non-vegetated areas, P surplus

is determined by inputs from atmospheric deposition and chemical weathering, denoted by ($DEP_{\text{NonVeg}}$, and $CW_{\text{NonVeg}}$). In urban areas, P surplus is determined by inputs from atmospheric deposition, denoted by ($DEP_{\text{Urban}}$. These relationships are outlined in equations (9–13) (all variables are in $kg\ ha^{-1}$ of physical area $yr^{-1}$):

$$Surp_{\text{NonAgri}}(i,y) = Surp_{\text{For}}(i,y) + Surp_{\text{NatVeg}}(i,y) + Surp_{\text{NonVeg}}(i,y) + Surp_{\text{Urban}}(i,y) \tag{9}$$

$$Surp_{\text{For}}(i,y) = DEP_{\text{For}}(i,y) + CW_{\text{For}}(i,y) \tag{10}$$

$$Surp_{\text{NatVeg}}(i,y) = DEP_{\text{NatVeg}}(i,y) + CW_{\text{NatVeg}}(i,y) \tag{11}$$

$$Surp_{\text{NonVeg}}(i,y) = DEP_{\text{NonVeg}}(i,y) + CW_{\text{NonVeg}}(i,y) \tag{12}$$

$$Surp_{\text{Urban}}(i,y) = DEP_{\text{Urban}}(i,y) \tag{13}$$

## 2.2  Land use types

We gathered a series of datasets to generate annual estimates of agricultural and non-agricultural areas. Within agriculture,
we considered cropland (fodder and non-fodder crops) and pasture land. These estimates are crucial for reconstructing the P surplus, particularly in deriving crop-specific fertilizer application rates and the allocation of animal manure and mineral fertilizer to cropland and pastures. We refer to Batool et al. (2022) for detailed steps and equations for the reconstruction of land use types. Below, we provide a summary.

### 2.2.1 Reconstruction of the agriculture area (cropland and pasture)

Cropland is defined as land used for the cultivation of crops, including arable crops and land under permanent crops (Ramankutty et al., 2008; FAOSTAT, 2021b). Pasture area is the land under permanent meadow and pasture and is defined as land used permanently (five years or more) to grow herbaceous forage crops, either cultivated or naturally occurring (e.g., wild prairie or grazing land) (FAOSTAT, 2021b). To represent the spatial distribution of cropland and pasture areas, we utilized the dataset from Ramankutty et al. (2008), which provides gridded estimates at a 5-arcminute resolution for the year 2000. These gridded values serve as the baseline for cropland and pasture area in our analysis. To account for temporal changes in cropland and pasture areas, we used data from the History Database of the Global Environment (HYDE version 3.2) (Goldewijk et al., 2017). HYDE provides global decadal estimates of cropland and pasture areas from 1700 to 2000, as well as annual values from 2000 to 2017. We generated annual time series of cropland and pasture areas for the period 1850–2019 using linear interpolation for the decadal estimates. For the years 2018 and 2019, we used the same values as 2017 due to a lack of available data.

To combine the data from Ramankutty et al. (2008) and from HYDE, we first calculated temporal ratios for the HYDE data for each grid cell using the year 2000 as the reference year. These ratios represent the relative change in cropland $R_{\text{HYDE-cr}}$ (-) and pasture area $R_{\text{HYDE-past}}$ (-) over time, normalized to the year 2000:

$$R_{\text{HYDE-cr}}(i, y_{1850,-,2019}) = \frac{A_{\text{HYDE-cr}}(i, y_{1850,-,2019})}{A_{\text{HYDE-cr}}(i, y_{2000})} \tag{14}$$

$$R_{\text{HYDE-past}}(i, y_{1850,-,2019}) = \frac{A_{\text{HYDE-past}}(i, y_{1850,-,2019})}{A_{\text{HYDE-past}}(i, y_{2000})} \tag{15}$$

Where $A_{\text{HYDE-cr}}(ha))$ and $A_{\text{HYDE-past}}(ha))$ are the gridded cropland and pasture areas, respectively.

Next, we applied these normalized ratios to the baseline gridded values from Ramankutty et al. (2008) to derive annual cropland and pasture areas for each grid cell, as follows:

$$A_{\text{cr}}(i, y_{1850,-,2019}) = A_{\text{Ramankutty-cr}}(i, y_{2000}) \times R_{\text{HYDE-cr}}(i, y_{1850,-,2019}) \tag{16}$$

$$A_{\text{past}}(i, y_{1850,-,2019}) = A_{\text{Ramankutty-past}}(i, y_{2000}) \times R_{\text{HYDE-past}}(i, y_{1850,-,2019}) \tag{17}$$

Where $A_{\text{Ramankutty-cr}}(ha)$ and $A_{\text{Ramankutty-past}}(ha)$ are the gridded cropland and pasture areas from Ramankutty et al. (2008) for the year 2000, and $A_{\text{cr}}(ha)$ and $A_{\text{past}}(ha)$ are the estimated cropland and pasture areas.

We harmonized our reconstructed cropland and pasture areas with FAOSTAT data available at country-level, which provides consistent information from 1961–2019. To do so, we calculated country-level ratios for cropland and pasture areas by comparing FAOSTAT data with the sum of our gridded estimates for each country. The ratios were calculated as follows:

$$R_{A_{\text{cr}}}(u, y_{1961,-,2019}) = \frac{A_{\text{FAO-cr}}(u, y_{1961,-,2019})}{\sum_{i=1}^{n_u} A_{\text{cr}}(i, y_{1961,-,2019})} \tag{18}$$

$$R_{A_{\text{past}}}(u, y_{1961,-,2019}) = \frac{A_{\text{FAO-past}}(u, y_{1961,-,2019})}{\sum_{i=1}^{n_u} A_{\text{past}}(i, y_{1961,-,2019})} \tag{19}$$

Whereas $R_{A_{\mathrm{cr}}}(-)$ is the country-level ratio of cropland area, $A_{\mathrm{FAO\text{-}cr}}(ha)$ represents the country-level cropland area from FAOSTAT, $n_u$ is the number of grid cells in country $u$, and $\sum_{i=1}^{n_u} A_{\mathrm{cr}}(ha)$ is the sum of the gridded cropland areas in country $u$ in year $y$. Similarly, $R_{A_{\mathrm{past}}}(-)$ is the ratio of pasture area, $A_{\mathrm{FAO\text{-}past}}(ha)$ is the country-level pasture area from FAOSTAT, and $\sum_{i=1}^{n_u} A_{\mathrm{past}}(ha)$ is the sum of the gridded pasture areas.

We applied these ratios to adjust our gridded estimates to match FAOSTAT's country-level data (all variables, except for ratios, are in $ha$):

$$A_{\mathrm{cr}}^{\mathrm{cor}}(i, y_{1961,-,2019}) = R_{A_{\mathrm{cr}}}(u, y_{1961,-,2019}) \times A_{\mathrm{cr}}(i, y_{1961,-,2019}) \tag{20}$$

$$A_{\mathrm{past}}^{\mathrm{cor}}(i, y_{1961,-,2019}) = R_{A_{\mathrm{past}}}(u, y_{1961,-,2019}) \times A_{\mathrm{past}}(i, y_{1961,-,2019}) \tag{21}$$

Whereas $A_{\mathrm{cr}}^{\mathrm{cor}}$ represents the corrected gridded cropland, $R_{A_{\mathrm{cr}}}$ is the country-level ratio of cropland area as given in equation 18, and $A_{\mathrm{cr}}$ is the original gridded cropland area as derived in equation 16. Similarly, $A_{\mathrm{past}}^{\mathrm{cor}}$ represents the corrected gridded pasture area, $R_{A_{\mathrm{past}}}$ is the country-level ratio of pasture area as shown in equation 19, and $A_{\mathrm{past}}$ is the original gridded pasture area as derived in equation 17.

For years prior to 1961, we used the same ratios as of 1961 to maintain consistency. In cases where FAOSTAT data were not available before 1992 (e.g., for Estonia, Croatia, Lithuania, Latvia, and Slovenia), we used the ratios from the year 1992 for the period 1850–1991. For countries like Luxembourg and Belgium, and Slovakia and Czech Republic, which were reported as single entities in historical records, we used combined ratios for the respective periods. Finally, the total agricultural area $A_{\mathrm{agri}}^{\mathrm{cor}}$ ($ha$) for each grid cell was calculated by summing the corrected cropland $A_{\mathrm{cr}}^{\mathrm{cor}}$ and pasture areas $A_{\mathrm{past}}^{\mathrm{cor}}$ (all variables are in $ha$):

$$A_{\mathrm{agri}}^{\mathrm{cor}}(i, y_{1850,-,2019}) = A_{\mathrm{cr}}^{\mathrm{cor}}(i, y_{1850,-,2019}) + A_{\mathrm{past}}^{\mathrm{cor}}(i, y_{1850,-,2019}) \tag{22}$$

We ensured physical consistency by checking that the agricultural area in each grid cell did not exceed the total physical area of the grid cell. In rare cases where this condition was violated due to inconsistencies in data sources (e.g., FAOSTAT (FAOSTAT, 2021b), HYDE (Goldewijk et al., 2017), and Ramankutty et al. (2008)), we redistributed the excess agricultural area to neighboring grid cells.

### 2.2.2 Reconstruction of the non-agriculture area

The non-agricultural area in a grid cell was calculated as the remaining area after allocating cropland and pasture areas. We used the classification of land cover categories from global land cover (GLC) (Bartholomé and Belward, 2005) that is available at a spatial resolution of 300 m. GLC includes 23 land cover classes that we grouped into 5 categories namely, cropland, semi-natural-vegetation (i.e. vegetation not planted by humans but influenced by human actions (Di Gregorio, 2005) including tree, shrub-land, herbaceous cover, Lichen and mosses), forest (broad-leaved, evergreen and deciduous forest), non-vegetation (bare areas, water bodies) and urban area. The proportions of these categories were then applied to the non-agricultural area to estimate their annual development from 1850 to 2019.

### 2.2.3 Reconstruction of crop-specific harvested area

We acquired gridded crop-specific harvested areas from Monfreda et al. (2008) for 175 different crops representing the year 2000. Among these, we selected 17 major non-fodder crops for which mineral fertilizer application rates are available (Heffer et al., 2017) and which are widely grown across Europe, as well as six fodder crop categories. Below we provide a more detailed overview on the selected crops (see also Table 4). These selected crops cover most of the cropland across Europe. The harmonization process ensures that the total cropland area aligns with FAOSTAT estimates.

To generate annual time series of crop-specific harvested areas, we applied the temporal dynamics of cropland areas, adjusting the spatial distribution of crops based on the Monfreda et al. (2008) dataset, while referencing FAOSTAT's country-level data to ensure consistency over time. The crop-specific harvested areas $A_{\text{crops}}$ ($ha$) were harmonized with FAOSTAT data $A_{crops_{FAO}}$ ($ha$) using a ratio-based approach. The ratio $R_A$ (-) between FAOSTAT country-level data and the sum of gridded estimates was calculated as follows:

$$R_A(u,c,y) = \frac{A_{crops_{FAO}}(u,c,y)}{\sum_{i=1}^{n_u} A_{\text{crops}}(i,c,y)} \tag{23}$$

This ratio was then applied to adjust the gridded estimates of crop-specific harvested areas for each grid cell, ensuring harmonization with FAOSTAT data:

$$A_{\text{crops}}^{\text{cor}}(i,c,y) = A_{\text{crops}}(i,c,y) \times R_A(u,c,y) \tag{24}$$

Where $A_{\text{crops}}^{\text{cor}}$ is the corrected crop-specific harvested areas for grid cell $i$, crop $c$, and year $y$.

For years prior to 1961, we applied the ratio from 1961 to maintain consistency across all years:

$$A_{\text{crops}}^{\text{cor}} = A_{\text{crops}}(i,c,y_{1850,-,1960}) \times R_A(u,c,y_{1961}) \tag{25}$$

This method ensured that the crop-specific harvested areas were harmonized with FAOSTAT country-level data, with each grid representing multiple crops.

For fodder crops, we utilized country-level data from Einarsson et al. (2021), available from 1961 to 2019 for 26 European countries. This dataset includes six fodder crop categories, namely: temporary grassland, lucerne, other leguminous plants, green maize, root crops (forage beet, turnip, etc.), and other fodder plants harvested from cropland. For the period 1850–1960, we applied the temporal dynamics of reconstructed cropland areas to estimate fodder crop areas. These estimates were harmonized with FAOSTAT's cropland totals to avoid discrepancies. For countries with missing data, we filled gaps by extrapolating ratios from neighboring countries with similar climatic and geographical conditions or using aggregated ratios from comparable regions.

## 2.3 P Inputs

Our estimates of P inputs include mineral fertilizer, animal manure, atmospheric deposition, and chemical weathering for both agricultural and non-agricultural soils at a gridded scale between 1850- 2019.

### 2.3.1 Mineral fertilizer

The quantity of fertilizer used on croplands and pastures is generally calculated based on application rates that vary across specific crops and pastures (West et al., 2014; Lu and Tian, 2017). Sattari et al. (2016) emphasized the need to consider the P cycle in pastures and its connection to croplands. P, like nitrogen (N), is a major limiting nutrient in agriculture. It is taken up by plants from croplands and is also removed from pastures through grazing, requiring replacement through inputs such as mineral fertilizer and animal manure to sustain crop and grass production (Sattari et al., 2012). Despite this, there is considerable uncertainty concerning how fertilizer is distributed between croplands and pastures (Zhang et al., 2021). To estimate these uncertainties, we generated two gridded estimates for fertilizer application by employing two distinct sets of application rates for croplands and pastures, which were then used to refine the country-level fertilizer data to a gridded format.

### 2.3.2 Country-level fertilizer applied to soil

For the period 1961 to 2019, we utilized FAOSTAT (FAOSTAT, 2023b) dataset on fertilizer applied to agricultural soils available at country-level. FAOSTAT provides P fertilizer inputs for agricultural use in the form of phosphate ($P_2O_5$), which we converted to elemental P using a molar mass conversion ratio of 0.436.

For countries without data before 1992, such as Lithuania, Croatia, Latvia, Estonia, Ukraine, and Belarus, we estimated P fertilizer application ($Pfer_{Missing_{East_{EU}}}$ ($kg\ yr^{-1}$)) during 1961–1991 by applying the temporal dynamics of Eastern European countries with available data (Czechoslovakia, Hungary and Bulgaria), as shown in Equation (26):

$$Pfer_{Missing_{East_{EU}}}(u, y_{1961-1991}) = \frac{Pfer_{East_{EU}}(y_{1961-1991})}{Pfer_{East_{EU}}(y_{1992})} \times Pfer_{Missing_{East_{EU}}}(u, y_{1992}) \tag{26}$$

where $u$ is country; $Pfer_{East_{EU}}$ ($kg\ yr^{-1}$) represents the total fertilizer application for Czechoslovakia, Hungary and Bulgaria. Additionally, Belgium and Luxembourg are reported as a single entity in FAOSTAT from 1961 to 1999, with separate country estimates available from 2000 onward. Similarly, Czechia and Slovakia are reported together under Czechoslovakia from 1961 to 1992. Before 2000 for Belgium and Luxembourg, and before 1993 for Czechia and Slovakia, we applied the historical dynamics of the combined entities.

Regarding the time period of 1850 – 1960, when country-level P fertilizer data from FAOSTAT were unavailable, we utilized the temporal dynamics from Cordell et al. (2009) that provides global estimates of phosphate rock production during 1800 – 2000. These estimated P inputs were normalized to align with FAOSTAT data starting in 1961, using 1961 as a reference year for consistency. The global temporal dynamics was then applied across all countries in our study domain for 1850–1960, proportionally scaling the values based on each country's 1961 estimate. This approach allowed us to generate a temporally coherent dataset, using global phosphate rock production as a proxy for P inputs from fertilizer during the period of limited data availability. The completed annual country-level fertilizer data are referred to as $Pfer_{soil}(u, y_{1850-2019})$ ($kg\ yr^{-1}$).

### 2.3.3 Allocation of fertilizer to croplands and pastures

For fertilizer allocation, we considered that fertilizer is applied to 100% of the cropland and pasture, since we did not have more detailed data to determine the spatial variability of fertilizer application rates within a given country. To convert the annual fertilizer amounts at the country level to grid-level distributions for croplands and pastures, we employed two distinct application rate sets to address uncertainties in the spatial patterns of fertilizer distribution within each country. Initially, we determined country-specific fertilizer application rates for various crops and grassland using data from the International Fertilizer Industry Association (IFA; https://www.ifastat.org). These rates were adjusted using two alternative methodologies, detailed in the following sections.

We sourced country-level data on fertilizer usage for different crop types and grassland ($Pfer_{crops}$ $Pfer_{grass}$, respectively, measured in $kg\ yr^{-1}$) from the IFA for 2014-2015 (Heffer et al., 2017). The IFA provides national-level rates for P fertilizer use in the form of $P_2O_5$ across 13 crop categories. For our analysis, we used IFA data corresponding to 17 non-fodder crops and 6 fodder crops. The non-fodder crops include cereals (wheat, maize grains and silage, rice, millet, rye, oats, sorghum, barley, triticale, and buckwheat), oil seeds (soybeans, rapeseed, sesame, and sunflower seeds), roots and tubers (potatoes) and sugar crops (sugar beet). The fodder crops include temporary grasslands and pastures for silage, hay, and grazing. For crops not explicitly specified by the IFA, such as pulses, we assumed fertilizer applications equivalent to those for soybeans, as both are leguminous crops. We converted IFA fertilizer application rates to phosphorus by applying a conversion factor of 0.436, as per equation 27. It is important to note that the IFA provides data on P fertilizer usage at the EU-28 level rather than for individual European countries, alongside figures for Belarus, Russia and Ukraine.

We calculated fertilizer application rates by combining IFA fertilizer usage data with FAOSTAT's crop-specific harvested area ($A_{crops_{FAO}}$ $(ha)$) and grassland area ($A_{grass_{FAO}}$ $(ha)$). Grassland areas encompass temporary grasslands (represented as one of the six fodder crop categories) as well as permanent pastures, with a single fertilizer application rate applied consistently across all grassland uses for grazing and forage production. This enabled us to derive country-specific fertilizer application rates for individual non-fodder crops ($Pfer_{crops_{Rate}}$ $(kg\ ha^{-1}$ of crop harvested areas $yr^{-1}))$ and grasslands ($Pfer_{grass_{Rate}}$ $(kg\ ha^{-1}$ of grassland areas $yr^{-1}))$, as illustrated in equations (28–29):

$$Pfer_{crops}(u, c, y_{2015}) = P_2O_5 fer_{crops}(u, c, y_{2015}) \times 0.436 \tag{27}$$

$$Pfer_{crops_{Rate}}(u, c, y_{2015}) = \frac{Pfer_{crops}(u, c, y_{2015})}{A_{crops_{FAO}}(u, c, y_{2015})} \tag{28}$$

$$Pfer_{grass_{Rate}}(u, y_{2015}) = \frac{Pfer_{grass}(u, y_{2015})}{A_{grass_{FAO}}(u, y_{2015})} \tag{29}$$

where $u$ is country; $c$ is non-fodder crop; $y_{2015}$ is the base year 2015; and $P_2O_5 fer_{crops}$ refers to the fertilizer usage derived from IFA for different non-fodder crop types in the form of phosphate.

For pastures and all six fodder crops, the fertilizer application rates were set to match those of grasslands, as indicated in equation (29). For countries not included in the IFA dataset, we used the EU-28 average fertilizer application rates for individual crops and pastures similar to Batool et al. (2022). Further, it is important to note that fertilizer application rates for non-fodder crops are based on fertilizer use per unit of corresponding harvested area to represent crop-specific fertilizer inputs.

For grassland, encompassing both temporary and permanent pastures, the fertilizer application rate is calculated using total grassland area, and ensuring consistent application of this rate across all relevant grassland areas.

To capture spatial variations, we applied the country-level fertilizer rates for non-fodder crops ($Pfer_{crops_{Rate}}$ ($kg\ ha^{-1}$ of crop harvested areas $yr^{-1}$)) and grasslands ($Pfer_{grass_{Rate}}$ ($kg\ ha^{-1}$ of grassland areas $yr^{-1}$)) to gridded areas of non-fodder crops, pastures, and fodder crops ($A_{crops}^{cor}$, $A_{past}^{cor}$, and $A_{fodder}$ respectively, ($ha$)) over the period from 1850 to 2019. This approach provided annual fertilizer application amounts for each crop type (non-fodder and fodder), pastures, and the overall total ($Pfer_{crops}$, $Pfer_{fodder}$, $Pfer_{past}$, $Pfer_{soil}$, respectively ($kg\ yr^{-1}$)) for each grid cell, as summarized in equations (30–33):

$$Pfer_{crops}(i,c,y_{1850-2019}) = Pfer_{crops_{Rate}}(u,c,y_{2015}) \times A_{crops}^{cor}(i,c,y_{1850-2019}) \tag{30}$$

$$Pfer_{past}(i,y_{1850-2019}) = Pfer_{grass_{Rate}}(u,y_{2015}) \times A_{past}^{cor}(i,y_{1850-2019}) \tag{31}$$

$$Pfer_{fodder}(i,c,y_{1850-2019}) = Pfer_{grass_{Rate}}(u,y_{2015}) \times A_{fodder}(i,c,y_{1850-2019}) \tag{32}$$

$$Pfer_{soil}(i,y_{1850-2019}) = \sum_{c=1}^{n_c} Pfer_{crops}(i,c,y_{1850-2019}) + Pfer_{past}(i,y_{1850-2019}) + \sum_{c=1}^{n_c} Pfer_{fodder}(i,c,y_{1850-2019}) \tag{33}$$

Next, the fertilizer application totals (as computed in equation (33)) were adjusted to ensure consistency with the country-level fertilizer amounts applied to soil during $1850-2019$, as reconstructed in earlier steps ($Pfer_{soil}$ ($kg\ yr^{-1}$)). This involved calculating an adjustment factor (a ratio) of the country-level fertilizer amount to the aggregated gridded fertilizer amount. The derived ratio was then applied to the fertilizer application rates of individual crops and grasslands ($Pfer_{crops_{Rate}}$ ($kg\ ha^{-1}$ of crop harvested areas $yr^{-1}$) and $Pfer_{grass_{Rate}}$ ($kg\ ha^{-1}$ of grassland areas $yr^{-1}$), respectively), leading to adjusted/corrected fertilizer application rates for crops and grasslands ($Pfer_{crops_{Rate}}^{cor}$ ($kg\ ha^{-1}$ of crop harvested areas $yr^{-1}$) and $Pfer_{grass_{Rate}}^{cor}$ ($kg\ ha^{-1}$ of grassland areas $yr^{-1}$), respectively), as given by equations (34–35):

$$Pfer_{crops_{Rate}}^{cor}(u,c,y_{1850-2019}) = Pfer_{crops_{Rate}}(u,c,y_{2015}) \times \frac{Pfer_{soil}(u,y_{1850-2019})}{\sum_{i=1}^{n_u} Pfer_{soil}(i,y_{1850-2019})} \tag{34}$$

$$Pfer_{grass_{Rate}}^{cor}(u,y_{1850-2019}) = Pfer_{grass_{Rate}}(u,y_{2015}) \times \frac{Pfer_{soil}(u,y_{1850-2019})}{\sum_{i=1}^{n_u} Pfer_{soil}(i,y_{1850-2019})} \tag{35}$$

where $u$ is country; $c$ is non-fodder crop; $y_{2015}$ is the base year 2015 and $n_u$ refers to the number of grid cells in country u.

To distribute the fertilizer application amounts between croplands and pastures, we employed two distinct sets of application rates to address methodological uncertainties. The first approach utilized IFA-derived application rates, which were subsequently adjusted using equations (34) and (35). The second approach involved further refining these rates to reflect the partitioning data provided by (Ludemann et al., 2023). According to Ludemann et al. (2023) data, a majority of the countries apply 100% of their fertilizer to croplands. This percentage differs for a few European countries, the proportions are as follows: 90% for Austria, Finland, France, Germany, the Netherlands, and Poland; 70% for Slovenia, Switzerland, the United Kingdom, and Luxembourg; and 30% for Ireland.

Ultimately, using both sets of fertilizer application rates, we calculated the gridded fertilizer quantities applied to croplands and pastures ($Pfer_{cr}$ and $Pfer_{past}$ in $kg\ yr^{-1}$, respectively), using the gridded areas of non-fodder crops $A_{crops}^{cor}$ ($ha$), fodder

crops ($A_{fodder}$) ($ha$), and pastures ($A_{\text{past}}^{\text{cor}}$ ($ha$). In the first method, employing the application rates from equations (34) and (35), the equations are formulated as follows:

$$Pfer_{cr}(i, y_{1850-2019}) = \sum_{c=1}^{n_c} (Pfer_{\text{crops}_{Rate}}^{\text{cor}}(u, c, y_{1850-2019}) \times A_{\text{crops}}^{\text{cor}}(i, c, y_{1850-2019}))$$

$$+ \sum_{c=1}^{n_c} (Pfer_{\text{grass}_{Rate}}^{\text{cor}}(u, y_{1850-2019}) \times A_{fodder}(i, c, y_{1850-2019})) \tag{36}$$

$$Pfer_{past}(i, y_{1850-2019}) = Pfer_{\text{grass}_{Rate}}^{\text{cor}}(u, y_{1850-2019}) \times A_{\text{past}}^{\text{cor}}(i, y_{1850-2019}) \tag{37}$$

where $u$ is country; $c$ is non-fodder crop and $n_c$ refers to the number of grid cells for crops c.

In the second method, the fertilizer application rates in equations (36) and (37) ($Pfer_{\text{crops}_{Rate}}^{\text{cor}}$ and $Pfer_{\text{grass}_{Rate}}^{\text{cor}}$) were replaced with adjusted rates derived from Ludemann et al. (2023).

For each method, the total gridded fertilizer amount applied to the soil was calculated by summing the fertilizer used

for croplands and pastures. This process yielded two distinct datasets reflecting methodological uncertainties in the spatial distribution within a country, while maintaining consistent country-level totals across both datasets.

### 2.3.4 Animal manure

P excretion by livestock, commonly referred to as manure production, is typically estimated using both P excretion rates and livestock number data. In the following, a detailed methodology of livestock numbers construction in the period 1850-2019

is first explained, which is then used to derive P manure production using P excretion coefficients based on previous studies (Sheldrick et al., 2003; Lun et al., 2018) (see Table 1). The resulting manure can be managed in various ways, such as being left on pasture or collected, stored, and subsequently applied to cropland and pasture soils. Given the absence of specific P data, we used nitrogen (N) data from FAOSTAT (FAOSTAT, 2022) and Einarsson et al. (2021) as proxies for estimating P manure applied to soil. From these two datasets, we employed three different methodologies for distributing animal manure to

cropland and pastures, resulting in a total of six estimates of P inputs from animal manure.

### 2.3.5 Country-level livestock counts

Initially, we utilized the FAOSTAT dataset to obtain country-level data on livestock counts (numbers) for eleven animal categories (asses, camels, cattle, chickens, goats, mules, sheep, pigs, buffaloes, ducks, and horses) from 1961 to 2019 (FAOSTAT, 2022). To extend this dataset back to 1850, we referred to historical data from Mitchell (1998), which provided livestock counts

for different animal categories in East and West Europe from 1890 to 1998 at ten-year intervals. We combined these continental datasets to form a comprehensive European dataset. We then generated the annual time series of the livestock counts for Europe for the period 1890 – 1960 using linear interpolation between every two ten-year estimates.

For the pre-1890 period (1850–1889), we inferred livestock numbers by associating them with the animal manure production dataset of (Zhang et al., 2017). This dataset is derived using the spatial distribution of livestock counts from the Global Live-

365 stock Impact Mapping System (GLIMS) (Robinson et al., 2014) and N excretion coefficients from the Intergovernmental Panel

on Climate Change (IPCC) (Dong et al., 2020) at a 5 arcmin spatial resolution for the time period 1860 – 2014. Since Zhang et al. (2017) dataset does not provide information before 1860 or after 2015, we extrapolated the 1860 data backward to 1850 and assumed constant manure production for this decade. Specifically, we calculated the ratio of animal manure production ($R_{man}$ (-)) for 1850-1889 relative to 1890 and applied this to estimate livestock numbers ($L$) ($head$):

$$R_{man}(u, y_{1850-1889}) = \frac{man(u, y_{1850-1889})}{man(u, y_{1890})} \tag{38}$$

$$L(u, l, y_{1850-1889}) = R_{man}(u, y_{1850-1889}) \times L(u, l, y_{1890}) \tag{39}$$

Here, $l$ is the livestock category. $R_{man}$(-) represents the ratio of animal manure production between 1850–1889 and the base year 1890. $L$ ($head$) is the estimated livestock number, adjusting earlier data to align with known values from 1890. For unaccounted three animal categories in Mitchell (1998) (chickens, camels, and ducks), we calculated the manure production ratio ($R_{\text{man}}$(-)) relative to the first year with available data from FAOSTAT i.e. for the year 1961 instead of 1890. This process allowed us to create a comprehensive time series of all eleven livestock categories across Europe from 1850 to 2019.

### 2.3.6  Spatial distribution of livestock counts

To spatially distribute livestock numbers, we employed the Gridded Livestock of the World database (GLW3) for the year 2010, which offers global livestock density data at a 5 arcminute resolution (Gilbert et al., 2018). For species like mules, not directly covered in GLW3, we proportionally allocated their numbers based on the distribution of similar animals (sheep or goats), a method supported by previous studies (Vermeulen et al., 2017). We then aggregated this gridded data to the country level, establishing a weighted ratio ($Wratio$) (-) for each livestock category ($LGLW$) ($head$) within each grid cell as given in equation 40. Subsequently, we applied these weighted ratios to disaggregate the country-level livestock time-series, yielding annual, gridded dataset of livestock numbers ($L$) ($head$) during 1850-2019 as in equation 41:

$$Wratio(i, l, y_{2010}) = \frac{LGLW(i, l, y_{2010})}{\sum_{i=1}^{n_u} LGLW(i, l, y_{2010})} \tag{40}$$

$$L(i, l, y_{1850-2019}) = L(u, l, y_{1850-2019}) \times Wratio(i, l, y_{2010}) \tag{41}$$

where $Wratio(i, y_{2010})$ is the weighted ratio of livestock numbers ($LGLW$) ($head$) provided by GLW3 per grid cell ($i$) to total country ($u$) level; $y_{2010}$ is the base year 2010; $L$ ($head$) is gridded dataset of livestock counts.

### 2.3.7  P manure production

After deriving the gridded livestock counts ($L$) ($head$) during 1850-2019, we estimated P manure production ($P_{man}$) ($kg\ head^{-1}\ yr^{-1}$) for each individual animal categories by multiplying the livestock count ($L$) ($head$) calculated in equation 41 with the P manure excretion coefficient (see Table1) ($P_{coeff}$) ($kg\ head^{-1}\ yr^{-1}$) as expressed in equation 42.

$$P_{man}(i, l, y_{1850-2019}) = L(i, l, y_{1850-2019}) \times P_{coeff}(l) \tag{42}$$

In the next step, we adjusted the P manure produced (calculated in equation 42) for each livestock category to ensure that
it is consistent with FAOSTAT data (FAOSTAT, 2022). We used the FAOSTAT dataset as reference database for country-
level information, due to its consistent availability for the period $1961-2019$ and global coverage across a range of variables
required to estimate the P surplus. To match our estimate of P manure produced, we first derived the amount of nitrogen (N)
excreted in manure from FAOSTAT for the time period $1961-2019$ for each livestock category (FAOSTAT, 2022). Then, we
converted these N content to P content using a P/N ratio (see Table1). Afterwards, for each year ($y$), country ($u$) and livestock
category ($l$), we calculated country-level correction factor (as ratios) ($R_{P_{man}}$ (-)) for P manure produced between those given
in FAOSTAT ($P_{man_{FAO}}$) ($kg\ head^{-1}\ yr^{-1}$) and those estimated in our study ($P_{man}$) ($kg\ head^{-1}\ yr^{-1}$), as summarized in
equations 43:

$$R_{P_{man}}(u,l,y_{1961-2019}) = \frac{P_{man_{FAO}}(u,l,y_{1961-2019})}{\sum_{i=1}^{n_u} P_{man}(i,l,y_{1961-2019})} \tag{43}$$

where $y_{1961}$ is the year 1961; $y_{1961-2019}$ is the year (in the period $1961-2019$). $u$ is country; $n_u$ is the number of grid cell in
the u-th country.

Then, we applied these calculated ratio ($R_{P_{man}}$) (-) to our gridded estimates of P manure produced ($P_{man}$) ($kg\ head^{-1}\ yr^{-1}$)
of equation 42. The resulting gridded P manure produced ($P_{man}^{cor}$) ($kg\ head^{-1}\ yr^{-1}$) can be given in equation 44 as:

$$P_{man}^{cor}(i,l,y_{1961-2019}) = R_{P_{man}}(u,l,y_{1961-2019}) \times P_{man}(i,l,y_{1961-2019}) \tag{44}$$

As FAOSTAT does not provide estimates before 1961, we applied the same ratio as of 1961 for the time period $1850-1960$
as given in equation 45:

$$P_{man}^{cor}(i,l,y_{1850-1960}) = R_{P_{man}}(u,y_{1961}) \times P_{man}(i,l,y_{1850-1960}) \tag{45}$$

For the countries for which FAOSTAT data are missing before 1992, such as Croatia, Estonia, Latvia, Lithuania and Slovenia,
we applied the same ratio as of the year 1992 for the period $1850-1991$. Furthermore, for the countries like Belgium and
Luxembourg, Czech Republic and Slovakia, FAOSTAT maintains single (combined) values of reported variables for the past
records (prior to 1993 for Czechoslovakia and before 2000 for Belgium-Luxembourg). In our estimation of country-specific
ratios we took care of these details, and accordingly applied a single ratio factor for the adjoining countries and records.
Finally, the total P manure ($P_{man}$) ($kg\ yr^{-1}$) was derived as a sum of the FAOTSTAT harmonized P manure produced for each
livestock category as mentioned in equation 46:

$$P_{man}(i,y_{1850-2019}) = \sum_{l=1}^{n_l} P_{man}(i,l,y_{1850-2019}) \tag{46}$$

where $y_{1961}$ is the year 1961; $y_{1850-1960}$ is the year (in the period $1850-1960$)

We accounted for different fates of P manure including those left on pasture by grazing animals, can be collected, stored and
then applied to soils (cropland and pasture). Given the absence of specific P data, we derived these contributions based on proxy

information of N manure given by FAOSTAT (FAOSTAT, 2022) and the European study of Einarsson et al. (2021). The FAO-STAT dataset calculates N excretion based on country-level livestock counts and regional-level values of typical animal mass and N excretion rates from the Intergovernmental Panel on Climate Change (IPCC) (Dong et al., 2020). In contrast, Einarsson et al. (2021) estimates N excretion by assuming proportionality to slaughter weights, following the methodology of Lassaletta et al. (2014). Specifically, from FAOSTAT, we used the 'Treated manure N' estimates, which represent the quantity of manure processed through specific manure management systems (e.g., lagoons, slurry, solid storage) prior to N loss in these systems (FAOSTAT, 2023c). Since P losses in these systems are minimal (FAOSTAT, 2023c), we considered that the entire amount of treated P manure is applied to soil. It is important to clarify that in this context, the term 'Treated' refers exclusively to manure management and does not extend to fertilizers, which are directly distributed to cropland and pasture areas without similar classification. We then calculated the country-level ratios ($R_{Nman_{treat}/prod}$) (-) of 'Treated manure ($Nman_{treat}$) ($kg\ yr^{-1}$)' to 'Total excreted manure ($Nman_{prod}$) ($kg\ yr^{-1}$)' and 'Manure left on pasture ($Nman_{left}$) ($kg\ yr^{-1}$)' to 'Total excreted manure ($Nman_{prod}$) ($kg\ yr^{-1}$)' for the years 1961-2019. These ratios, denoted as $R_{Nman_{treat}/prod}$ and $R_{Nman_{left}/prod}$ (-), respectively, were determined as follows:

$$R_{Nman_{treat}/prod}(u, y_{1961-2019}) = \frac{Nman_{treat}(u, y_{1961-2019})}{Nman_{prod}(u, y_{1961-2019})}, \tag{47}$$

$$R_{Nman_{left}/prod}(u, y_{1961-2019}) = \frac{Nman_{left}(u, y_{1961-2019})}{Nman_{prod}(u, y_{1961-2019})}. \tag{48}$$

From Einarsson et al. (2021), we calculated 'Treated manure' by summing 'Applied to cropland', 'Applied to permanent grassland', and 'Lost from houses and storage'. Similar to FAOSTAT, we derived ratios of 'Treated manure' to 'Excreted total' and 'Excreted grazing on permanent grassland' to 'Excreted total' for every European country for the period 1961-2019. Utilizing these ratios from two datasets, we estimated the spatial distribution of treated manure (that is applied to croplands and pastures) and manure left on pastures across the different grid cells. The treated manure ($Man_{treat}$) and manure left ($Man_{left}$) on pastures, expressed in $kg\ yr^{-1}$, were calculated by applying the above ratios to the gridded P manure production data ($P_{man}$ ($kg\ yr^{-1}$)), as shown below:

$$Man_{treat}(i, y_{1961-2019}) = R_{Nman_{treat}/prod}(u, y_{1961-2019}) \times P_{man}(i, y_{1961-2019}), \tag{49}$$

$$Man_{left}(i, y_{1961-2019}) = R_{Nman_{left}/prod}(u, y_{1961-2019}) \times P_{man}(i, y_{1961-2019}). \tag{50}$$

For the historical period of 1850–1960, we applied the 1961 ratios to the earlier manure production data ($P_{man}$ ($kg\ yr^{-1}$)) to estimate both treated ($Man_{treat}$ ($kg\ yr^{-1}$)) and left manure ($Man_{left}$ ($kg\ yr^{-1}$)), assuming that these management practices remained consistent over time:

$$Man_{treat}(i, y_{1850-1960}) = R_{Nman_{treat}/prod}(u, y_{1961}) \times P_{man}(i, y_{1850-1960}), \tag{51}$$

$$Man_{left}(i, y_{1850-1960}) = R_{Nman_{left}/prod}(u, y_{1961}) \times P_{man}(i, y_{1850-1960}). \tag{52}$$

We thus reconstructed the annual time series of two gridded datasets comprising of treated manure ($Man_{treat}$ $kg$ $yr^{-1}$) and manure left on pasture ($Man_{left}$ $kg$ $yr^{-1}$) across Europe for the period 1850–2019.

### 2.3.8 Distribution of treated manure between cropland and pasture

To allocate the manure applied to soil (derived from equations 49 and 52) from FAOSTAT and Einarsson et al. (2021) datasets between cropland and pasture, we employed three distinct methodologies to account for uncertainties. First, based on approaches from previous studies on the distribution of manure (Xu et al., 2019; Batool et al., 2022), we assumed equal distribution rates for cropland and pasture within each grid cell. Consequently, the manure applied to cropland ($Man_{cr}$) ($kg$ $ha^{-1}$ of physical area $yr^{-1}$) is calculated by dividing the treated manure ($Man_{treat}$ ($kg$ $yr^{-1}$)) by the physical area ($A_{\text{grid}}$ ($ha$)), and

then multiplied by the proportion of cropland area ($Prop_{A_{\text{cr}}}$ (-)) within the grid cell, as outlined in equation 53. The proportion of cropland area is calculated by dividing the cropland area ($ha$) by the total cropland and pasture area ($ha$) within the grid cell. Similarly, the manure designated for pasture application ($Man_{app_{past}}$) ($kg$ $ha^{-1}$ of physical area $yr^{-1}$) is determined by dividing the treated manure ($Man_{treat}$ ($kg$ $yr^{-1}$)) by the physical area ($A_{\text{grid}}$ ($ha$)), and then multiplying by the proportion of pasture area ($Prop_{A_{\text{past}}}$ (-)), as detailed in equation 54. The proportion of pasture area is calculated by dividing the pasture

area ($ha$) by the total cropland and pasture area ($ha$). Finally, the total manure allocated to pastures ($Man_{past}$) ($kg$ $ha^{-1}$ of physical area $yr^{-1}$) is then calculated by adding the manure applied to pastures ($Man_{app_{past}}$) ($kg$ $ha^{-1}$ of physical area $yr^{-1}$) and the manure left on pastures by grazing animals ($Man_{left}$) ($kg$ $yr^{-1}$) normalized by the grid's physical area ($A_{\text{grid}}$) ($ha$), as expressed in equation 55.

$$Man_{cr}(i, y_{1850-2019}) = \frac{Man_{treat}(i, y_{1850-2019}) \times Prop_{A_{\text{cr}}}(i, y_{1850-2019})}{A_{\text{grid}}(i)}, \tag{53}$$

$$Man_{app_{past}}(i, y_{1850-2019}) = \frac{Man_{treat}(i, y_{1850-2019}) \times Prop_{A_{\text{past}}}(i, y_{1850-2019})}{A_{\text{grid}}(i)}, \tag{54}$$

$$Man_{past}(i, y_{1850-2019}) = Man_{app_{past}}(i, y_{1850-2019}) + \frac{Man_{left}(i, y_{1850-2019})}{A_{\text{grid}}(i)}. \tag{55}$$

Second, we distributed the manure applied to soil based on country-level data on manure application proportions to cropland and pasture, as reported by Ludemann et al. (2023). Accordingly, a majority of the countries apply nearly 100% of their manure to croplands, with particular values for European nations such as 90% for Austria, Finland, France, Germany, the Netherlands,

and Poland, and 70% for Slovenia, Switzerland, the United Kingdom, and Luxembourg, while Ireland applies 30%. Using this information, we calculated the country-level ratios of manure applied to cropland and pasture relative to the total manure application. Subsequently, we adjusted the gridded manure application rates to cropland and pasture (equations 53 and 54, respectively) using the respective country-scale ratios. In the third method for manure application, we allocated the manure applied to soil (as calculated in equations 49 and 51) using the time-varying national proportions of nitrogen (N) manure

applied to both cropland and pasture, as provided by Einarsson et al. (2021). This study (Einarsson et al., 2021) used national-level information specific to each country to assign stored manure across cropland and pasture for different animal types.

We modified our gridded manure applications for cropland and pasture (equations 53 and 54) to align with the proportions estimated by Einarsson et al. (2021).

Overall, by integrating two distinct data sources ((FAOSTAT, 2022) and Einarsson et al. (2021)) alongside three manure distribution methods between croplands and pastures, we developed six separate gridded manure estimates for our database. These estimates reflect the uncertainties in our reconstruction, which are due to the different selections of the underlying data sets and methods. Each method highlights different aspects of manure allocation: the equal distribution assumption adjusts with cropland and pasture area changes over time, while the country-specific ratios from Ludemann et al. (2023) use fixed, national-level allocations. The third method, based on Einarsson et al. (2021), uses time-varying N based proportions as a proxy for P manure distribution. Supplementary Figure S1 illustrates these proportion of animal manure allocated to cropland and pasture under each method, highlighting the differences and capturing the uncertainties embedded in our approach. By combining these varied assumptions, our estimates provide a comprehensive view of manure distribution across cropland and pasture, allowing for a nuanced analysis of P surplus uncertainty.

### 2.3.9 Atmospheric deposition

In our study, we assessed P inputs from atmospheric deposition for different land types, including agricultural land (cropland and pastures) and non-agricultural land. To estimate P deposition for agricultural soils, we used the dataset provided by Ringeval et al. (2024) which represents global atmospheric deposition rates of P to cropland and pasture from 1900 to 2018 at a spatial resolution of 0.5 degrees. This dataset accounts for various sources, including mineral dust, primary biogenic aerosol particles, sea salt, natural combustion, and anthropogenic combustion (e.g., agricultural residue burning, forest fires, logging fires, and fossil fuel burning) (Ringeval et al., 2024). We adjusted this dataset to the spatial resolution of 5 arc minutes required for our study using nearest neighbour interpolation. For the historical period from 1850 to 1899, we projected the deposition rates backwards from 1900, assuming that they are constant across this period. Similarly, for 2019, we extrapolated the data from 2018. Then, the P deposition rates were multiplied by the corresponding land use areas for cropland and pasture to quantify the P inputs from atmospheric deposition on these land types.

For non-agricultural land, we used the dataset from Wang et al. (2017), which provides the global total (for both agricultural and non-agricultural areas) atmospheric deposition of nitrogen (N) and P from various deposition processes for the years between 1980 and 2013. This dataset, which contains snapshots for specific years (1980, 1990, 1997, followed by an annual series until 2013), was linearly interpolated to create an annual series for the period 1980-2013. For earlier years (1850-1979) we used 1980 deposition rates, while for the most recent period (2014-2019) we used 2013 data. We recognize that assuming constant P deposition rates over the past years is an oversimplification, partly due to lack of observations and reliable datasets. To determine the P deposition rates on non-agricultural soils, we calculated the difference between the total atmospheric P deposition rates from Wang et al. (2017) and the agricultural soil deposition rates from Ringeval et al. (2024).

### 2.3.10 Chemical weathering

P inputs from chemical weathering refer to the natural release of P from rocks and minerals into the soil. This process is influenced by factors such as the type of rock (lithology), temperature, and soil properties (Panagos et al., 2022a). Hartmann and Moosdorf (2011); Hartmann et al. (2014) developed a global database of P release from chemical weathering by incorporating lithological and runoff information. The dataset allows for understanding of how P is released from various types of rocks under different environmental conditions.

For our study, we used the European-specific rates of P release from chemical weathering in $kg\ ha^{-1}$ of physical area $yr^{-1}$ taken from the global dataset of Hartmann et al. (2014). These P release rates ($CW_{Rate}$) ($kg\ ha^{-1}$ of physical area $yr^{-1}$) were combined with the GLiM (Global Lithological Map) (Hartmann and Moosdorf, 2012) lithographic maps to obtain their spatial distribution across European landscapes. We then multiplied these rates ($CW_{Rate}$) ($kg\ ha^{-1}$ of physical area $yr^{-1}$) by the respective gridded land use areas in $ha$ within our study region, excluding urban areas, to estimate the P inputs in $kg\ yr^{-1}$ from chemical weathering on a gridded scale, which we then divided by physical area ($A_{\mathrm{grid}}\ (ha)$) to derive the estimates in $kg\ ha^{-1}$ of physical area $yr^{-1}$, as given in equations 56-59.

$$CW_{cr}(i, y_{1850-2019}) = \frac{A_{\mathrm{cr}}^{\mathrm{cor}}(i, y_{1850-2019}) \times CW_{Rate}(i)}{A_{\mathrm{grid}}(i)} \tag{56}$$

$$CW_{past}(i, y_{1850-2019}) = \frac{A_{\mathrm{past}}^{\mathrm{cor}}(i, y_{1850-2019}) \times CW_{Rate}(i)}{A_{\mathrm{grid}}(i)} \tag{57}$$

$$CW_{For}(i, y_{1850-2019}) = \frac{A_{\mathrm{For}}(i, y_{1850-2019}) \times CW_{Rate}(i)}{A_{\mathrm{grid}}(i)} \tag{58}$$

$$CW_{NatVeg}(i, y_{1850-2019}) = \frac{A_{\mathrm{NatVeg}}(i, y_{1850-2019}) \times CW_{Rate}(i)}{A_{\mathrm{grid}}(i)} \tag{59}$$

where $CW_{cr}$, $CW_{past}$, $CW_{For}$, and $CW_{NatVeg}$ refer to P inputs from chemical weathering for areas covered by cropland, pasture, forest, and natural vegetation, respectively, in $kg\ ha^{-1}$ of physical area $yr^{-1}$; $A_{\mathrm{cr}}^{\mathrm{cor}}$, $A_{\mathrm{past}}^{\mathrm{cor}}$, $A_{\mathrm{For}}$, $A_{\mathrm{NatVeg}}$, and $A_{\mathrm{grid}}$ represent the gridded areas of cropland, pasture, forest, natural vegetation, and the grid's physical area, respectively, in $ha$; and $CW_{Rate}$ denotes the P release rate from chemical weathering, based on lithological data, in $kg\ ha^{-1}$ of physical area $yr^{-1}$.

Finally, the total P from chemical weathering is obtained by summing above individual estimates (equations 56-59).

## 2.4 P outputs

This section outlines the reconstruction steps for estimating P removal from croplands and pastures. Additionally, we provide a summary of the approach used to estimate gridded crop production, which is essential for calculating P removal from harvested crops (for detailed methodology, see Batool et al. (2022)).

### 2.4.1 P removal from cropland

The P removal from cropland ($Rem_{cr}$ ($kg\ ha^{-1}$ of physical area $yr^{-1}$)) is calculated by summing the P removal across all crop types. This is achieved by multiplying the crop production ($Pro_{crops}$ ($t\ yr^{-1}$)) by the specific P content of each crop ($P_{content}(c)$ ($kg\ t^{-1}$)) and then dividing by physical area ($A_{grid}$ ($ha$)), as described in equation (60).

$$Rem_{cr}(i, y_{1850-2019}) = \frac{\sum_{i=1}^{n_c}(Pro_{crops}(i, c, y_{1850-2019}) \times P_{content}(c))}{A_{grid}(i)} \tag{60}$$

Given the variability of $P_{content}$ for crops in different studies (Ludemann et al., 2023; Hong et al., 2012; Guejjoud et al., 2023; Panagos et al., 2022a; Einarsson et al., 2020; Lun et al., 2018; Zou et al., 2022; Antikainen et al., 2008), we considered the resulting uncertainty by creating two scenarios. The first scenario applies the minimum values of P content from the literature to estimate the lower bound of P removal, while the second scenario uses the maximum values to estimate the upper bound. Table 4 lists the specific P content values for each crop used in our analysis.

### 2.4.2 Crop production

We compiled country-level crop production data from FAOSTAT for 1961–2019 (FAOSTAT, 2021a), covering 17 crops excluding fodder crops (as mentioned above; see Table 4). Fodder crop data were obtained from Einarsson et al. (2021) for 26 European countries during 1961–2019. We followed the methodology of Batool et al. (2022) for reconstructing the crop production development across Europe. We provide here a brief overview of the basics of these reconstructions, and interested readers can refer to Batool et al. (2022) for more details.

For the period 1850–1960, we compiled wheat production data from Bayliss-Smith and Wanmali (1984) as cited in Our World in Data (OWD) (OWD, 2021) at country-level, which provided wheat yields for selected years. The annual wheat yield data was determined by linear interpolation, whereby wheat production was calculated as the product of wheat yield and harvested area. For other crops during 1850–1960, we used the temporal dynamics of wheat production referenced to the base year 1961. Specifically, the country-level ratio of wheat production from Bayliss-Smith and Wanmali (1984) as cited in Our World in Data (OWD) (OWD, 2021) during the period 1850–1960 relative to wheat production from FAOSTAT (FAOSTAT, 2021a) for the base year (1961) was applied to estimate the crop production of other considered crops from FAOSTAT. Similar methodology was applied to reconstruct the annual production of fodder crops using the country-level estimates provided by Einarsson et al. (2021). We downscaled country-level crop production ($Pro_{crops}$) ($t\ yr^{-1}$)) using the gridded Monfreda et al. (2008) dataset ($Pro_{crops_{Monfreda}}$) ($t\ yr^{-1}$)) (as given in equation 61, which provides the respective crop production data at 5 arcmin spatial resolution for the base year around 2000. This approach maintained spatial heterogeneity and consistency in crop production estimates.

$$Pro_{crops}(i, c, y_{1850,-,2019}) = \frac{Pro_{crops\,Monfreda}(i, c, y2000)}{\sum_{i=1}^{n_u} Pro_{crops\,Monfreda}(i, c, y2000)} \times Pro_{crops}(u, c, y_{1850,-,2019}) \tag{61}$$

Here, $u$ refers to a given country and $n_u$ to the total number of grid cells within a country.

The temporal alignment between wheat production and other crop categories was assessed using scatter plots and correlation coefficients for the EU28 region (Supplementary Figure S2). Most crops showed a reasonable correlation with wheat produc-

tion, indicating consistent temporal dynamics across different crop types. These results supports the use of wheat production dynamics as a proxy for other crops during the reconstruction period (1850–1960). However, variations in correlation strength among crops suggest that future refinements could benefit from incorporating additional crop-specific data where available.

### 2.4.3 P removal from pasture

For pastures, P removal ($Rem_{\text{past}}$ in $kg\ ha^{-1}$ of physical area $yr^{-1}$) was calculated as the amount of grass harvested and grazed, utilizing a method from prior studies (Bouwman et al., 2005, 2009). This approach relies on phosphorus use efficiency (PUE), where P removal from pastures is determined by multiplying a P removal coefficient $Rem_{\text{past}}^{coeff}$ (-) by the P inputs to pastures ($Inp_{past}$ in $kg\ ha^{-1}$ of physical area $yr^{-1}$), as described in equation (62):

$$Rem_{\text{past}}(i, y_{1850-2019}) = Rem_{\text{past}}^{coeff} \times Inp_{past}(i, y_{1850-2019}) \tag{62}$$

Since PUE values can vary among studies, similar to N use efficiency (NUE), we accounted for this uncertainty by considering different P removal coefficients. To address these uncertainties, we considered two approaches. In the first approach, we assumed a value of 0.6 for $Rem_{\text{past}}^{coeff}$ based on (Bouwman et al., 2005, 2009). In the second approach, we used NUE values provided by (Kaltenegger et al., 2021) as a proxy for PUE. Accordingly, we assumed $Rem_{\text{past}}^{coeff}$ values of 0.4 and 0.5 for countries located in Eastern and Western Europe, respectively. By applying these approaches, we derived two distinct datasets

for P removal from pastures, each reflecting different assumptions about PUE to account for the associated uncertainties. Many studies have generally focused on the cropland P surplus budget (Table 3) and accordingly they do not consider P removal from pasture areas. Therefore, our dataset allows for a more comprehensive view of P dynamics in agricultural landscapes.

## 3 Results

### 3.1 Spatio-temporal variation in P surplus, P inputs and P outputs

In our study, we developed 48 estimates of P surplus across Europe with a spatial resolution of 5 arc minutes (1/12°), accounting for uncertainties within the main P surplus components. Specifically, we analyzed two separate datasets for fertilizer, six datasets for animal manure, two datasets for P removal from croplands and two datasets for P removal from pasture. The averages of P fluxes (P surplus, inputs, and outputs) for 1850–2019 are presented at the grid level in Fig 2, with units expressed as $kg; ha^{-1}$ of physical area $yr^{-1}$. Additionally, Fig 3 shows the contribution of non-agricultural P surplus to the total P

surplus, while Fig 4 depicts the average of the 48 P surplus estimates at various aggregation levels. Uncertainties in these estimates are highlighted in Fig 5.

The spatio-temporal variations in our P surplus, inputs, outputs at the gridded level is illustrated in Fig 2 for the selected years: 1900, 1930, 1960, 1990 and 2015 (See Supplementary Figure S3 for the corresponding variations in mineral fertilizer and animal manure). These plots show that, while Northern Europe consistently exhibits a positive P surplus with relatively stable

P inputs and outputs, most of Central and Western Europe experiences variable P fluxes dynamics over time. For example, in 1900 and 1930, there are notable areas in Central and Western Europe with negative P surplus (P deficit), where P outputs

exceeds P inputs, particularly in agricultural regions. As time progresses, the pattern shifts. By 1960 and 1990, the P surplus becomes more positive across these regions. During this time periods, Northern Europe continues to show a positive P surplus, with values ranging from approximately 0 to 4 $kg\ ha^{-1}$ of physical area $yr^{-1}$, and with a balanced P inputs and outputs between 2–4 $kg\ ha^{-1}$ of physical area $yr^{-1}$. Conversely, the mid-latitude areas, particularly in Central and Western Europe, exhibit higher P surplus and inputs, from 10 to over 18 $kg\ ha^{-1}$ of physical area $yr^{-1}$, with moderate outputs (4–14 $kg\ ha^{-1}$ of physical area $yr^{-1}$) in most of the grids, whereas Southern Europe presents moderate P surplus and outputs, between 4 and 8 $kg\ ha^{-1}$ of physical area $yr^{-1}$, with higher P inputs (10–16 $kg\ ha^{-1}$ of physical area $yr^{-1}$). Notably, industrialized countries like Germany, France, and the Netherlands experienced a peak in P surplus and inputs around 1990, followed by a decline except in the Netherlands, where P surplus exceeded 20 $kg\ ha^{-1}$ of physical area $yr^{-1}$. P outputs in some regions also continued to rise. By 2015, an increase in grid cells with negative P surplus (P deficit) was observed, particularly in areas like central France and Germany, reflecting a situation where P outputs exceeds P inputs, similar to a century ago, as can be seen in central France and Germany. Central European countries mainly rely on mineral fertilizers, except regions like the Netherlands, Belgium, and Denmark, where animal manure dominates due to high livestock densities (See Supplementary Figure S3). Overall, over the period from 1850 to 2019, our analysis identifies large temporal fluctuations in P fluxes across most European regions, except for the north, where P fluxes levels have remained stable at a low level. This underscores the importance of long-term datasets in capturing such variations.

Furthermore, cumulative P fluxes, including P surplus, inputs, and outputs, are presented for four distinct time periods, which we term as following: (i) 1850–1920 (Pre-modern agriculture), (ii) 1921–1960 (Industrialization before the Green Revolution), (iii) 1961–1990 (Green Revolution and synthetic fertilizer expansion), and (iv) 1991–2019 (Environmental awareness and policy intervention phase) (Supplementary Figure S4). These plots revealed marked shift in P dynamics across Europe over time. During 1850-1920, P surplus was relatively low, averaging 8-10 $t\ yr^{-1}$ in much of the Central and Eastern Europe, with some Western Europe regions like France, the Netherlands, and Denmark exceeding 16 $t\ yr^{-1}$. Northern Europe typically showed much lower values of 2-4 $t\ yr^{-1}$. In the subsequent period (1921–1960), P inputs began to rise modestly, averaging 50-70 $t\ yr^{-1}$, driven by early industrialization and chemical fertilizer use, though P surplus remained moderate due to relatively high P outputs. The Green Revolution period (1961–1990) saw a sharp increase in P inputs, exceeding 80 $t\ yr^{-1}$ in many regions due to agricultural intensification, resulting in substantial P surplus, with most areas surpassing 18 $t\ yr^{-1}$. In the most recent phase (1991–2019), P inputs declined steadily due to improved agricultural practices and environmental policies like the EU Nitrates Directive, while P outputs increased, narrowing P surplus. In some Western and Eastern Europe, P surplus even turned negative, reflecting P mining. These temporal and spatial trends highlight the importance of sustainable nutrient management practices and policies in reducing P surplus over time. Moving forward, strategies like reallocating nutrients inputs based on regional needs, improving the integration of crop and livestock systems could help to further optimize nutrient use efficiency. Such measures, coupled with continued monitoring of P indicators-P surplus and PUE- are essential to address P-related environmental challenges and promote sustainable agricultural practices (Zou et al., 2022).

The peak in P surplus observed around 1980 likely aligns with the intensified fertilizer use of the Green Revolution (Supplementary Figure S5). The subsequent decline in P surplus after 1990 reflects multiple factors, including policy shifts in

Western Europe (e.g., Nitrate Directive (Directive 91/676/EEC) (European Commission, 2000b) and Water Framework Directive (Directive 2000/60/EC) (European Commission, 2000a)), regional legislations that restricted P fertilization (Amery and Schoumans, 2014)), economic adjustments, and increased awareness of sustainable nutrient management (Ludemann et al., 2023; Senthilkumar et al., 2012; Cassou, 2018). Country-specific legislation also played a role, since a few European countries, including the Netherlands, Ireland, Norway, and Sweden, have specific legislation limiting P applications (Bouraoui et al., 2011). In some cases, the decrease in P surplus began even earlier, as in Denmark and the UK, where P was not a major limiting factor for crop yield since soil P levels had likely reached sufficient levels for crop production without additional inputs (Bouraoui et al., 2011). On the other side, in Central and Eastern European regions, the collapse of the Soviet Union and subsequent (agro-)economic restructuring may have led to reduced P inputs, as indicated by a sharp drop in fertilizer use (Csathó et al., 2007; Ludemann et al., 2023) (Supplementary Figure S5) and subsequently reflected in corresponding P surplus budgets. Such distinct P surplus patterns observed across Europe appear to have been shaped by these combined influences, and disentangling the different factors will require careful consideration in future studies. On a global scale, Zou et al. (2022) discussed the distinct roles of socioeconomic and environmental factors governing the dynamics of long-term P surplus evolution across different countries.

The importance of non-agricultural P surplus is highlighted in Fig 3, which illustrates its contribution to total P surplus. Northern European countries, such as Norway, Sweden, and Finland, show a higher contribution of non-agricultural P surplus, with 30–60% contribution across 70% of grid cells during the entire period (1850–2019). Central and Western Europe exhibit more variable contributions over time. For example, in 1900 and 1930, the non-agricultural contribution in these regions ranged between 10–30%, but it decreased to around 10% by 1990, with further declines in recent years. Southern Europe, meanwhile, displayed a moderate and stable contribution of up to 20% from 1960 to 2019. Supplementary Figures S6 and S7 provide additional insights, showing the contribution of non-agricultural P surplus at both the country level and on a decadal scale. Northern and Eastern European countries demonstrate increasing contributions over time, such as Estonia (from 15% in 1850–60 to 30% in 2010–19) and Sweden (from 35% to 40% over the same period). Meanwhile, countries like Belgium, the Netherlands, and Switzerland show a consistent decrease in contribution throughout the period, such as Switzerland dropping from 40% in 1850–60 to 5% in 2010–19. Understanding these dynamics is critical for devising holistic nutrient management strategies that account for the role of non-agricultural P sources. By incorporating non-agricultural P surplus data, our dataset enables a more comprehensive understanding of P fluxes across Europe.

The availability of gridded P surplus data enables detailed analysis at sub-national-scale within the European "Nomenclature of Territorial Units for Statistics" (NUTS), as illustrated in Fig 4. This figure underscores the importance of breaking down the P surplus data to the sub national level. Such a breakdown is crucial as it reveals spatial heterogeneity that are otherwise masked by country level averages. For example, our 2015 analysis shows that France's national P surplus (NUTS 0) appears moderate at 1 $kg\ ha^{-1}$ of NUTS physical area $yr^{-1}$; however, at NUTS 1 and NUTS 2 levels, regional disparities become evident. Specifically, Brittany in northwest France emerges as a hotspot with a P surplus exceeding 15 $kg\ ha^{-1}$ of NUTS physical area $yr^{-1}$, significantly above the national average. Furthermore, our gridded data allows for tracking P surplus changes over time in river basins that span multiple countries. This possibility is particularly valuable as river basins represent a crucial

spatial unit for water quality modelling and land-water management. Our results, shown in the right panel of Fig 4, illustrate the temporal dynamics of P surplus in different river basins. From 1930 to 1960, a steady increase in P surplus was observed in most of these river basins. This trend was followed by a significant increase around 1990, after which there was a marked decline. In the Danube river basins, for instance, which covers numerous Southeastern and Central European countries, the P surplus increased from 3 $kg\ ha^{-1}$ of NUTS physical area $yr^{-1}$ in 1930 to 5 $kg\ ha^{-1}$ of NUTS physical area $yr^{-1}$ in 1960, representing a 1.5-fold increase. This trend continued from 1960 to 1990, with P surplus values rising from 5 $kg\ ha^{-1}\ yr^{-1}$ in 1960 to 8 $kg\ ha^{-1}$ of NUTS physical area $yr^{-1}$ in 1990. After 1990, however, there was a sharp decline, with the P surplus decreasing fourfold to 2 $kg\ ha^{-1}$ of NUTS physical area $yr^{-1}$ by 2015.

Figure 5 (and Supplementary Figures S8 and S9) illustrates the time series of agricultural and total P surplus and its contributing components (P inputs and P outputs) with the variation of the uncertainty range (defined as the difference between the maximum and minimum of our 48 P surplus estimates) over time and regions. Here specifically, we analyze P surplus data for the EU-27, Germany, and the Danube River Basin. Generally, for agricultural P surplus during the period from 1850 to 1930, the uncertainty intervals (represented by grey ribbons in Fig 5a-c) were comparable in size to the mean estimates (depicted by red lines) for both the EU-27 and the Danube River Basin. In Germany, however, the uncertainty intervals were more than double the mean values, reflecting high variability in P surplus estimates. Between 1930 and 1950, the uncertainty intervals increased at a moderate rate. From 1950 to 1990, the relative size of the uncertainty intervals compared to the mean estimates decreased by approximately 2.5 times in all three regions. By 1990, it was approximately 39% of the mean for both the EU-27 and the Danube River Basin, and 44% for Germany. After 1990 and until 2010, the uncertainty range began to stabilize in all three regions, indicating a more consistent level of variability in the later years. In the last decade, however, the uncertainty interval showed a two-fold increase compared to the mean value for Germany, an increase by a factor of around 3.5 for the Danube River Basin, while for the EU-27, there was a relatively slight increase. Regarding the absolute differences between the maximum and minimum P surplus estimates, the uncertainty intervals (represented by grey ribbons) showed a consistent increase from 1850 to 1950, ranging between 2-4 $kg\ ha^{-1}$ of agricultural area $yr^{-1}$ for the EU-27 and 3-4 $kg\ ha^{-1}$ of agricultural area $yr^{-1}$ for the Danube River Basin (see Fig 5a,c). In Germany, the disparity nearly tripled, rising from 3 $kg\ ha^{-1}$ of agricultural area $yr^{-1}$ in 1850 to 8 $kg\ ha^{-1}$ of agricultural area $yr^{-1}$ by 1950. From 1950 to 1990, these values continued to grow for Germany, the EU-27, and the Danube River Basin, peaking at nearly 14 $kg\ ha^{-1}$ of agricultural area $yr^{-1}$ in Germany during the 1980s and about 9 $kg\ ha^{-1}$ of agricultural area $yr^{-1}$ for both the EU-27 and the Danube River Basin. Post-1990, the uncertainty levels stabilized at approximately 7 $kg\ ha^{-1}$ of agricultural area $yr^{-1}$ for the EU-27 and the Danube River Basin and around 11 $kg\ ha^{-1}$ of agricultural area $yr^{-1}$ for Germany. A similar temporal pattern in uncertainty ranges was observed for total P surplus across the EU-27, Germany, and the Danube River Basin (Fig 5d-f).

To assess the uncertainty in P surplus estimates, we calculated the coefficient of variation (CV, %), defined as the ratio of the standard deviation to the mean across our 48 P surplus estimates. This analysis, shown in Supplementary Figure S10, offers insights into how relative uncertainty has evolved over time. The CV was highest in the early period (1850–1920) for many countries, including Germany and France and then declined significantly during the mid-20th century (1950–1990). However, in recent decades, relative uncertainty has increased again, especially in countries like Spain and Italy.

In addition, we examined the absolute uncertainty ranges (calculated as maximum minus minimum) of P surplus estimates for each year, comparing these against the ranges of key components, including fertilizer, manure, and P output (Supplementary Figures S11–S13). The results indicate that in central, eastern, and Mediterranean countries such as Germany, Spain, Italy, Slovakia, Slovenia, Poland, and Portugal, fertilizer input uncertainty aligns closely with P surplus uncertainty, identifying fertilizer as potentially a primary driver of variation in these regions (Supplementary Figure S11). In contrast, manure inputs show a more variable relationship with P surplus uncertainty across countries, with generally weaker associations than fertilizer. However, in livestock-intensive regions such as Ireland and the Netherlands, manure uncertainty strongly contributes to P surplus variation (Supplementary Figure S12). For P outputs, associations with P surplus uncertainty are moderate to strong in countries including Germany, France, Spain, the UK, the Netherlands, and Italy, suggesting that output variability also plays a role in P surplus uncertainty, especially in areas with high agricultural productivity (Supplementary Figure S13). Overall, fertilizer inputs emerge as the dominant factor influencing P surplus uncertainty, although the impact of P outputs and manure inputs also varies by region, reflecting distinct agricultural practices. These preliminary findings emphasize the substantial spatial and temporal variability in P surplus uncertainties and underscore the value of ensemble datasets in capturing comprehensive nutrient flows. Further statistical analyses would be required to investigate the factors controlling the uncertainties in P surplus in future studies.

### 3.2 Technical evaluation of reconstructed P surplus

To evaluate the spatial and temporal consistency and plausibility of our P surplus dataset, we conducted a comparison of our 48 P surplus estimates with existing datasets, acknowledging the absence of direct P surplus observations. Initially, we utilized global databases available at the country level, including those from the Food and Agricultural Organization of the United Nations (FAOSTAT) (Ludemann et al., 2023), Zou et al. (2022) for the period 1961–2019, and Lun et al. (2018) for the period 2002–2010. The dataset of FAOSTAT (Ludemann et al., 2023) is taken as a global reference dataset that provides nutrient budgets on cropland for 205 countries and territories. The second dataset of Zou et al. (2022) provides P budget and P use efficiency (PUE) by country and crop type, using multiple variables from FAOSTAT such as crop yield to estimate P crop removal, animal N manure database to estimate P inputs from animal manure. The estimates from FAOSTAT (Ludemann et al., 2023) and Zou et al. (2022) differ in terms of distribution of mineral fertilizer and animal manure to cropland and pasture. Finally, the third evaluation dataset is based on Lun et al. (2018) who provides P inputs and P outputs of agriculture systems from 2002 to 2010, based on which we estimated the corresponding country-wise P surplus estimates. A second level assessment was conducted to check the country-level consistency. Here we compared our P-budget with country specific estimates for France (Guejjoud et al., 2023)(1920 − 2019), the United Kingdom (DEFRA (2022)) (1990, 1995 and 2000 − 2019), Slovenia (Verbič and Sušin, 2022) (1992 − 2019); and individual EU27 countries and UK were compared to estimates given by (Einarsson et al., 2020) for 2013 and by Eurostat (2024) for 2010 − 2019, respectively. Lastly, we also compared our estimates at sub-national level of NUTS 2 with the dataset provided by (Einarsson et al., 2020).

At continental-level, the uncertainty intervals for our P surplus (assessed as one standard deviation around the mean of the 48 estimates) agrees well with values derived from FAOSTAT (Ludemann et al., 2023) and Zou et al. (2022). Specifically,

our average cropland P surplus for EU-27 countries over the period $1961-2019$ is equal to $1.70 \pm 0.49$ Tg of P, whereas FAOSTAT (Ludemann et al., 2023) provides a value of 1.85 Tg of P and Zou et al. (2022) reports 2.06 Tg of P. For the period $2002-2010$, our estimated cropland P surplus for the EU-27 is $0.82 \pm 0.46$ Tg of P, Lun et al. (2018) reports a value of 0.76 Tg of P. Regarding country-specific studies, DEFRA for UK (DEFRA, 2022) reported an average value (during 1990, 1995, and 2000–2019) of 7.13 $kg\ ha^{-1}$ of agricultural area $yr^{-1}$ for agricultural P surplus for the United Kingdom, which falls within our uncertainty bound of $5.82 \pm 2.11$ $kg\ ha^{-1}$ of agricultural area $yr^{-1}$. Further, Senthilkumar et al. (2012) reported a declining trend in France during 1990–2006, with agricultural P surplus reduced by almost 75% in 2006 compared to the estimate in 1990 (from 17.5 $kg\ ha^{-1}$ of agricultural area in 1990 to 4.4 $kg\ ha^{-1}$ of agricultural area in 2006). Our estimates show a similar decline, with P surplus decreasing by 66% (from 17 $kg\ ha^{-1}$ of agricultural area in 1990 to 5.71 $kg\ ha^{-1}$ of agricultural area in 2006).

To assess the discrepancies between the P surplus from our study and the FAOSTAT data (Ludemann et al., 2023) (as well as those from Zou et al. (2022) and Lun et al. (2018)), we calculated the country-specific relative difference $Diff$ (%), as follows: equation 63:

$$Diff = \left( \frac{Reference_{Psur} - Study_{Psur}}{Reference_{Psur}} \right) \times 100 \tag{63}$$

Here, $Reference_{Psur}$ represents the average P budget reported by FAOSTAT (Ludemann et al., 2023) during the time period $1961-2019$ (in $kg\ ha^{-1}$ of cropland area $yr^{-1}$) (respectively, Zou et al. (2022) between $1961-2019$ in $kg\ ha^{-1}$ of cropland area $yr^{-1}$ and Lun et al. (2018) between $2002-2010$ in $tonne$) and $Study_{Psur}$ denotes one of our 48 P surplus estimates averaged over the relevant time period.

We provide here the mean and standard deviation (SD) for the 48 values of $Diff$(%) calculated for each of our 48 P surplus datasets. We also show the full range (minimum-maximum) of the 48 values of $Diff$(%), highlighting the full extent of discrepancies between different estimates. We found that, in about 68% of the countries, the mean $Diff$ between our estimates and FAOSTAT was contained in the range $\pm30\%$ (see Figure 6c). The differences in P surplus across these countries ranged from a minimum of -160% to a maximum of 112%. This broad range of discrepancies underscores the variability in our estimates compared to FAOSTAT data. Notable deviations of more than $\pm50\%$ in mean estimates were found in several countries. For instance, Slovakia (SK) exhibited the relative $Diff$ of $-238\% \pm 159\%$ (mean $\pm$ std. dev.), with a range varying from -485% to 45%. Norway (NO) showed a mean $Diff$ of $-131\%(\pm 24\%)$, ranging from -178% to -90%. Similarly, Lithuania (LT) had a mean $Diff$ of $-84\% \pm 118\%$, with a range from -275% to 114%. Croatia's (EL) mean $Diff$ was $-84\% \pm 50\%$, with a range from -145% to 26%. Czechia (CZ) displayed a mean $Diff$ of $-61\% \pm 199\%$, with a range from -395% to 250%. While the mean estimates show large differences between our and FAOSTAT (Ludemann et al., 2023) dataset, the ensemble realisations (48 estimates) show a large uncertainty ranging from negative to positive values of $Diff$. This indicates that the uncertainty estimates in our data sets can capture the values given by FAOSTAT (Ludemann et al., 2023) for almost all EU countries analysed. Here, the differences identified reflect the importance of taking uncertainty into account when reconstructing the P-surplus. Our datasets – in contrast to those of FAOSTAT (Ludemann et al., 2023) – use different approaches to account for uncertainties

in the main components of the P surplus, e.g. in the application of manure and fertilisers on cropland and in the P removal coefficients, while uncertainties are not accounted for in FAOSTAT. In addition, FAOSTAT does not take into account P inputs from atmospheric deposition and chemical weathering (Ludemann et al., 2023), which are minor components of the P surplus but could still lead to differences.

With respect to Zou et al. (2022), around 56% of the EU countries show a mean *Diff* in the range $\pm$ 30% (see Figure 7c), with a full range of -131% to 98%. Some countries, such as Czechia (CZ; $Diff = 291\% \pm 218$), and Slovakia (SK; $Diff = 314\% \pm 93$), presented higher discrepancies on average but also exhibited larger uncertainties. Specifically, the range of minimum and maximum *Diff* for Czechia is from -34% to 672%, while for Slovakia it ranges from 156% to 469%. This indicates substantial variability and highlights the challenges in achieving consistent P surplus estimates across different datasets and methodologies. These discrepancies might be due, among other factors, to differences in the distribution of mineral fertilisers and animal manure on cropland and pastures. Further, in Lun et al. (2018), only 37% of countries showed the mean *Diff* in the range $\pm$ 30% (see Figure 8c), with a full range of -212% to 231%. The discrepancies between our P surplus estimates and those provided by Lun et al. (2018) can be partly attributed to their inclusion of P removal from leaching, considered as 12.5% of P inputs to cropland, which is not accounted for in our estimates. This additional removal component leads to difference, where around 50% of the European countries show a negative *Diff*, suggesting that the higher P surplus in our estimates may be related to the exclusion of leaching, among other factors. Furthermore, the considerable spread in the data again underscores the variability introduced by methodological choices used in reconstructing the P surplus budget.

To evaluate how well the temporal patterns of our P budget align with existing datasets, we calculated the correlation coefficient ($r$). We report the mean and standard deviation of 48 $r$ values derived from each of our 48 P surplus estimates. We observed a strong, positive correlation with different available datasets, particularly with FAOSTAT Ludemann et al. (2023) for the period 1961–2019, where the mean $r$ value was 0.89 $\pm$0.11 (mean $\pm$ std. dev.) (see Figure 6b). Notably, approximately 70% of the countries exhibited high correlation coefficients, ranging from 0.90 to 1.0, with none recording values below 0.4 (Figure 6a). Similarly, the dynamics of our cropland P budgets were strongly aligned with findings from Zou et al. (2022) during the same period, achieving a mean $r$ of 0.76 $\pm$0.11 (see Figure 7b). Approximately 75% of the countries showed good consistency with Zou et al. (2022), with mean $r$ values between 0.80 and 1.0 (Figure 7a). However, lower correlations were observed for Lithuania (LT; 0.37 $\pm$ 0.15), Bosnia and Herzegovina (BA; 0.39 $\pm$ 0.13), and Belgium (BE; 0.49 $\pm$ 0.11), with Ireland (IE) recording the lowest at 0.10 $\pm$ 0.19. The discrepancy for Ireland is likely due to different assumptions and methodologies used in the distribution of mineral fertilizer and animal manure on cropland. Zou et al. (2022), for example, assumes that 66% of the animal manure is applied to cropland in Ireland, leading to higher P input estimates. In contrast, one of our estimates, based on FAOSTAT (Ludemann et al., 2023), assumes only 30% of the animal manure and mineral fertilizer is applied to cropland, with our other two approaches showing a range between 10–25%. Additionally, comparison with Lun et al. (2018) over the period 2002–2010 revealed a high correlation ($r = 0.80 \pm 0.09$; see Figure 8b), with approximately 75% of the countries having $r$ values between 0.70 and 1.0 (Figure 8a). Notable outliers were Bosnia and Herzegovina (BA) and Finland (FI) with the lowest mean $r$ values of -0.10 $\pm$ 0.18 and 0.01 $\pm$ 0.14, respectively. Among other things (mineral fertilisers and

manure distributions), variations in data sources and crop removal coefficients could explain the differences between our and previous estimates (Lun et al., 2018; Zou et al., 2022; Ludemann et al., 2023).

Overall, the narrow spread of the $r$ values (Figure 6 – Figure 7) across different studies — Ludemann et al. (2023), Zou et al. (2022), and Lun et al. (2018) — suggests consistent temporal dynamics in our P surplus estimates. The standard deviations (SD) of $r$ were less than 0.20 for the majority of countries (84% for FAOSTAT; Ludemann et al. (2023), 78% for Zou et al. (2022), and 71% for Lun et al. (2018)). Similarly, we observed a strong and positive correlation between our P budget estimates and those of country-specific estimates. Notably, France showed an $r$ of $0.95 \pm 0.01$ (see Figure 9a), while the United Kingdom (UK) recorded an $r$ of $0.98 \pm 0.00$ for the period 1990–2018 (Figure 9b). Slovenia also demonstrated a high correlation with an $r$ of $0.92 \pm 0.01$ for the period 1992–2019 (Figure 9c). For broader regions, the combined EU27 and UK showed an $r$ of $0.91 \pm 0.06$ in 2013 (Figure 9d), while the European countries covered by Eurostat exhibited an $r$ of $0.68 \pm 0.08$ during 2010–2019 (Figure 9e). Additionally, a comparison at the sub-national (NUTS 2) level within the EU for 2013, using estimates from Einarsson et al. (2020), revealed an $r$ value of $0.91 \pm 0.03$ (see Figure 9f). This strong positive association between the sub-national scale dataset and ours underscores the consistency of our reconstructed datasets.

Overall, our 48 estimates of P surplus at both continental and country levels align well with existing estimates in terms of spatio-temporal patterns and relative differences. Although there are discrepancies in the mean P surplus estimates, the full range of our uncertainty estimates captures well the values provided by previous studies (Lun et al., 2018; Einarsson et al., 2020; Zou et al., 2022; Ludemann et al., 2023, ; among others). This underscores the need for a more comprehensive characterization of uncertainties in future studies to improve the accuracy of P surplus estimations. In our analysis, we addressed only a subset of these uncertainties, but it is crucial to acknowledge additional sources that might explain the variations between our results and those from previous studies. For instance, differences in the input data sources for different P surplus components, such as fertilizers and manure applications, can greatly influence the P estimates. The IFA (Heffer et al., 2017) database is primarily based on sales data from fertilizer companies, whereas FAOSTAT (FAOSTAT, 2023a) gathers data through questionnaires from member countries, and Eurostat (2024) relies on country-specific statistics and United Nations Framework Convention on Climate Change (UNFCCC) data (Eurostat, 2023) for gaps. Another contributing factor could be the exclusion or inclusion of minor components such as P inputs from atmospheric deposition, chemical weathering, seed and planting materials and P removal via crop residues. FAOSTAT, for instance, does not account for P inputs from atmospheric deposition and chemical weathering, while our study includes them, which might lead to some differences, especially in regions with higher atmospheric P deposition or weathering components, such as areas with high industrial activity or regions with specific geological formations (Hartmann et al., 2014). Moreover, differences in P budget can largely be attributed to different coefficients used by different studies for estimating for instance P content in harvested crops.

Despite the challenges described above, our dataset represents a substantial advance as it provides 48 different uncertainty estimates for key components of P surplus. In contrast, previous datasets (Ludemann et al., 2023; Zou et al., 2022; Lun et al., 2018; Einarsson et al., 2020; DEFRA, 2022; Verbič and Sušin, 2022) typically lack detailed uncertainty reporting for P surplus. Our comparison of P surplus estimates with existing datasets underscores the importance of further addressing uncertainties in

future updates of the P surplus dataset. This also requires further refinement of our datasets to narrow down the uncertainty, for example, by considering region specific mineral fertilisers, manure applications or crop removal rates.

## 4    Potential use and limitations of the dataset

This study provides valuable insights into long-term phosphorus (P) surplus trends across Europe from 1850 to 2019 by incorporating 48 different estimates, that take into account methodological uncertainties. This ensemble approach enhances the robustness of the dataset by offering a range of potential outcomes rather than relying on a single estimate. Nevertheless, certain limitations should be noted when interpreting these results.

First, the reconstruction of P surplus before 1961 is constrained by limited historical data. For the period 1850–1960, we re-

lied on proxy information and extrapolations based on data from 1961 onward, which inherently introduces higher uncertainty. For instance, national wheat production trends were used to estimate other crop productions, but this method may not fully capture the variations of each specific crop types. We assume that the relative values of the fertilizer application rates taken from the International Fertilizer Association (IFA) (Heffer et al., 2017) for 2014–2015 remained constant for the period 1850–2019, and pre-1961 temporal variations were inferred from global phosphate rock production trends. Livestock distributions were

based on GLW3 (Gilbert et al., 2018) data circa 2010 to estimate manure production for the entire 1850–2019 period, and simplifying assumptions were made before 1961, since no country-level manure data were available. Such simplifications may not accurately reflect historical livestock numbers or distribution patterns, influencing P surplus estimates. Furthermore, spatial datasets, especially for land use and crop production, are more detailed and reliable from the mid-1990s onward, making the P surplus estimates more robust for recent decades. Thus, while historical estimates provide general trend insights, recent data

(from the mid-1990s) offer greater reliability.

Second, while the dataset has been downscaled to a high-resolution grid, several underlying inputs, such as fertilizer use and manure application, originate from country-level aggregates. This approach limits the ability of gridded outputs to capture local variability in P surplus. Therefore, while the dataset is valuable at coarser basin, regional, national and continental scales, caution should be exercised for high-resolution applications. We recommend using the dataset at aggregate levels, such as

countries, European socio-economic regions (e.g., NUTS levels), or river basin scales (see Section "Spatio-temporal variation in P surplus, P inputs and P outputs" and Figure 4) to support land and water management activities. Additionally, global datasets, such as the Global Livestock Impact Mapping System (GLIMS) and Hartmann's P weathering rates, were used within the European context. While informative, these datasets may not fully capture regional differences in livestock practices or lithological conditions across Europe.

Further limitations include the simplification of parameters that likely vary across space and time. While the coefficients used were based on prior research (Bouwman et al., 2005; Kaltenegger et al., 2021), PUE can be highly variable across regions and management practices (Lun et al., 2018; Chowdhury and Zhang, 2021). Our use of fixed coefficients may not fully capture this variability, especially in countries with varying level of grazing intensities or grassland management practices. Furthermore, climate-related factors such as changes in precipitation, temperature, and soil moisture directly affect pasture productivity

and thus P uptake and cycling (Martins-Noguerol et al., 2023), which our static approach does not fully encompass. This simplification was necessary due to the lack of detailed historical agricultural records but introduces some degree of uncertainty in our P removal from pasture areas. Likewise, crop-specific uptake rates for P removal from cropland likely vary with soil quality, crop varieties, and management practices. This simplification of using fixed coefficients was necessary due to the lack of detailed historical agricultural records but introduces some degree of uncertainty in our removal estimates. Furthermore, while including P inputs from atmospheric deposition and chemical weathering offers a more comprehensive view of P dynamics, using a uniform P weathering rate from Hartmann et al. (2014) across all non-agricultural areas may oversimplify spatial variability in lithology and mineral composition.

Finally, our primary focus is on P inputs from well-documented sources, such as agricultural activities (fertilizer and manure) and natural processes (atmospheric deposition and weathering) (Panagos et al., 2022b; Ringeval et al., 2024). However, certain other sources, such as P inputs from urban areas (e.g., urban fertilizers and human waste), were not explicitly accounted for.

Despite these limitations, this study provides a valuable dataset that offers a comprehensive view of P surplus dynamics across Europe over a long historical period at gridded resolution. The construction of multiple estimates of P surplus based on different assumptions provides a nuanced understanding of the uncertainties involved. The uncertainty in P surplus estimates arises from various factors, including different assumptions about fertilizer and manure distribution to cropland and pasture, crop uptake coefficients, and historical data quality. The importance of accounting for uncertainty is increasingly emphasized in nutrient research, as demonstrated by recent work, such as Guejjoud et al. (2023), Sarrazin et al. (2024) and Zhang et al. (2021), which emphasise how datasets incorporating uncertainty provide a stronger foundation for robust nutrient and water quality studies.

Furthermore, our gridded dataset supports a greater degree of flexibility for aggregating data at various spatial scales, such as national, regional, and river basin levels, which is helpful in analyzing trans-boundary nutrient flows across Europe. This flexibility allows the dataset to address the needs of cross-regional and trans boundary applications, including in major European river basins like the Elbe, Danube and Rhine, thereby facilitating joint nutrient management in shared water bodies (Müller-Karulis et al., 2024). By identifying critical regions of high P surplus or its components (P fertilizer/manure), users can pinpoint locations where nutrient management improvements could have the greatest environmental and economic impact (Malagó and Bouraoui, 2021). Moreover, linking our P surplus dataset to water quality and biodiversity models supports ecological assessments, as users can link P surplus data to eutrophication or freshwater biodiversity models, thus evaluating the ecological consequences of nutrient imbalances in sensitive ecosystems. Additionally, the dataset offers unique insights into both agricultural and non-agricultural land types, providing a comprehensive view of phosphorus (P) dynamics that was previously unavailable. Whereas existing datasets (Eurostat (2024); Einarsson et al. (2020); Lun et al. (2018)) typically focus solely on agricultural land, our approach includes P inputs from atmospheric deposition and chemical weathering on non-agricultural soils, which is particularly relevant for countries with extensive non-agricultural areas, such as Norway, Sweden, and Finland, where nutrient contributions from forests and semi-natural lands (see Figures 3 and Supplementary Figures S6 and S7). A comprehensive understanding of the P surplus budget over a given landscape and water quality assessment at a catchment scale requires considering contributions from both agricultural and non-agricultural areas (Van Meter et al., 2021).

Although pre-1961 data are limited, we have employed reliable sources (e.g., Cordell et al. (2009), Bayliss-Smith and Wanmali (1984)) to infer reasonable temporal dynamics. By covering the period from 1850, our dataset captures pivotal shifts in agricultural practices, land use, and industrial development that directly influenced phosphorus (P) dynamics prior to the Green Revolution (Guejjoud et al., 2023). These early changes laid the groundwork for modern nutrient management (Pratt and El Hanandeh, 2023; Ringeval et al., 2024; Sharpley et al., 2013), making the dataset useful not only for current policy analysis

but also as a historical baseline for exploring how shifts in climate and agricultural practices affect nutrient cycles over time. Coupled with our nitrogen (N) surplus dataset (Batool et al., 2022), this dataset enables integrated nutrient studies, facilitating the development of comprehensive management strategies that support both P and N sustainability goals. Additionally, the dataset's detailed historical record could support climate adaptation studies, enabling stakeholders to examine how nutrient budgets respond to evolving climate conditions and assess the long-term sustainability of various agricultural practices under

changing environmental conditions.

Importantly, to allow for reproducibility, we provide the methodology, which builds on existing studies, and code used in this study to reconstruct P surplus. This approach can be adopted by other researchers to develop similar datasets for other regions, contributing to a more comprehensive understanding of nutrient dynamics and their environmental impacts. Our methodology thus, can be reused to incorporate new data as they become available, such as a new version of the FAOSTAT database. Overall,

this dataset offers a comprehensive, adaptable, and uncertainty-aware resource for analyzing long-term P dynamics across Europe's diverse landscapes, supporting a wide range of applications in environmental, agricultural, and climate adaptation research.

## 5   Directions for future improvement of the dataset

This study opens several pathways for future improvements to better capture long-term phosphorus (P) dynamics across Eu-

rope.

Improving the historical land-use and livestock data is essential for refining long-term phosphorus (P) surplus estimates. Currently, the dataset applies uniform land-use data throughout the study period, but incorporating detailed historical land-use reconstructions, such as changes in cropland expansion, pasture reduction, or urbanization, would enhance the spatial accuracy of P surplus calculations. Additionally, livestock and manure data from the mid-20th century introduce uncertainties, particu-

larly in estimating manure production and distribution. A more detailed reconstruction of historical livestock numbers, along with records on manure management systems, would increase reliability. As part of this effort, promoting standardized data collection and reporting methods across European countries would further enhance data accuracy and consistency. Engaging with agricultural practitioners, policymakers, and environmental organizations could help refine data collection methods, ensure alignment with user needs, and expand the dataset's applicability for practical use. Further explorations of contributing

parameters in priority areas could also help guide future updates.

Our uncertainty analysis highlights that the distribution of mineral fertilizer and manure to cropland and pasture, and P removal estimates are the primary contributors to variability in P surplus estimates. Addressing these uncertainties should

be prioritized to enhance the reliability of nutrient datasets. Sensitivity analyses to identify key sources of uncertainty could also help refine the methodology, allowing researchers to focus on critical parameters that most influence P surplus. Further,

our dataset also does not cover all sources of uncertainties, and further uncertain data and methodological choices worth investigating are discussed in Section "Technical evaluation of reconstructed P surplus".

P inputs from urban areas especially those from human waste (i.e., sewage and wastewater) are not accounted for in our analysis. These inputs are typically classified as point sources, with much of the waste directly discharged directly into water bodies, rather than contributing to diffuse soil inputs. Our study focuses on characterizing major diffuse sources in the P surplus

budget. In parallel, we have also developed a long-term database on nutrient inputs from point sources (urban areas), detailed in a separate study by Sarrazin et al. (2024). Additional datasets, such as the European Pollutant Release and Transfer Register (E-PRTR) (Roberts, 2009) and the nutrient load database by Vigiak et al. (2020), provide valuable information on nutrient contributions from urban and industrial sources across Europe. Our current dataset on diffuse sources of P complements existing datasets on point sources and contributes to ongoing efforts to comprehensively understand P dynamics across terrestrial

ecosystems, spanning both diffuse and point sources. Future work could integrate point and diffuse sources with our existing dataset to provide a more comprehensive characterization of P inputs to terrestrial systems.

The role of non-agricultural P sources requires more attention. While atmospheric deposition and weathering inputs were included in this study, the uniform weathering rates may not reflect the spatial heterogeneity of soil types and mineral compositions across Europe. Future studies should integrate more localized data to capture the nuances of P dynamics in non-

agricultural areas, particularly forests and semi-natural regions. In addition, future work should explore how shifting environmental conditions due to climate change, such as altered precipitation patterns and temperatures, will influence P inputs, outputs, and surplus. Additionally, addressing cross-border P dynamics, particularly within shared river basins like the Danube or Rhine, could provide insights for collaborative nutrient management across European regions, where trans boundary nutrient flows impact water quality and ecosystem health.

Another future enhancement would involve refining parameters to account for temporal and spatial variability. For example, crop-specific P uptake rates vary with soil quality, crop variety, and management practices, while pasture P removal is influenced by phosphorus use efficiency (PUE) and meteorological and hydrological variables such as precipitation, temperature, and soil moisture. Future work should incorporate dynamic PUE estimates to better capture time-varying removal rates driven by regional grazing practices, crop types, and changing weather patterns.

Finally, expanding the dataset's scope to analyze interactions between P and nitrogen (N) cycles could yield a more holistic understanding of nutrient dynamics. By aligning P surplus data with established N datasets (Batool et al., 2022), future research could explore how shifts in one nutrient cycle affect the other. This integrated approach would support more comprehensive nutrient management strategies, helping to mitigate environmental impacts and promote sustainable agricultural practices across Europe. Collaborative efforts between researchers, policymakers, and agricultural stakeholders will be essential to enhance the

accuracy, relevance, and application of these datasets for future nutrient management.

## 6 Code and data availability

The dataset (version 2.0) is available at the Zenodo repository: https://doi.org/10.5281/zenodo.11351028 (Batool et al., 2024). The dataset consists of 48 distinct estimates of P surplus, P inputs and P outputs from 1850 to 2019, provided in Network Common Data Form (NetCDF) format. Each NetCDF file includes an annual variable for P surplus, P inputs and P outputs expressed in $kg\ ha^{-1}$ of physical area $yr^{-1}$, with a spatial resolution of 5 arcminutes (1/12°) over the entire period. In addition, P surplus data is available at aggregated spatial scales, including NUTS1, NUTS2, and specific river basins in and around Europe, in comma-separated values (.csv) format. These datasets are stored in the Zenodo repositoryBatool et al. (2024). The organization of the files is as follows:

   – 48 NetCDF files named "`P_sur_total_kg_ha_grid_1850_2019_method_xx`", each representing a unique method (1 through 48) and covering 170 years (1850-2019) of gridded P surplus data.

   – 1 NetCDF files named "`P_inp_total_kg_ha_grid_1850_2019`", representing mean of 48 P inputs and covering 170 years (1850-2019) of gridded P input data.

   – 1 NetCDF files named "`P_out_total_kg_ha_grid_1850_2019`", representing mean of 48 P outputs and covering 170 years (1850-2019) of gridded P output data.

   – 2 NetCDF files named "`P_fert_total_kg_ha_grid_1850_2019_method_xx`", each representing a unique method (1-2) and covering 170 years (1850-2019) of gridded P inputs from mineral fertilizer.

   – 6 NetCDF files named "`P_man_total_kg_ha_grid_1850_2019_method_xx`", each representing a unique method (1 through 6) and covering 170 years (1850-2019) of gridded P inputs from animal manure.

   – 3 CSV files labeled as"`P_sur_total_kg_ha_NUTS_xx`" containing P surplus data at various aggregated spatial levels. Here, xx represents NUTS 1, 2, or 3, and each file includes the mean and standard deviation of the 48 P surplus estimates.

   – 1 CSV file called "`P_sur_total_kg_ha_river_basin`" which provides P surplus information for selected river basins, including the mean and standard deviation of the 48 estimates.

Additional readme files accompany the datasets, detailing the NUTS and river basin IDs.

We used the RStudio version (2023.12.1+402) for data processing. The R scripts to assists the data analysis are provided on the Zenodo repository: https://doi.org/10.5281/zenodo.11351028 (Batool et al., 2024)

## 7 Conclusions

This study presents a comprehensive dataset of long-term annual P surplus estimates for Europe, covering both agricultural and non-agricultural soils from 1850 to 2019 and focusing only on diffuse sources. The dataset, with a high spatial resolution

of 5 arcmin (approximately 10 km at the equator), includes 48 different estimates that account for uncertainties arising from varying methodological choices and coefficients in major components of the P surplus. Our data construction is the result of extensive data collection and integration from a multitude of sources to ensure the reliability of the estimates. With a broader aim of analyzing P dynamics across diverse land use types, the dataset highlights the importance of understanding both inputs and outputs across all land types, including areas where agricultural P sources are not dominant. Specifically, studies on water quality assessment at a catchment scale require P surplus data from both agricultural and non-agricultural areas to quantify and analyze total catchment P export.

The magnitude of P surplus shows significant variability over time and space, emphasizing the need for careful consideration in water quality and land management studies. Specifically, the total P surplus across EU-27 has tripled within 170 years, from $1.19 \pm 0.28$ $kg\ ha^{-1}$ of physical area in 1850 to around $2.48 \pm 0.97$ $kg\ ha^{-1}$ of physical area in recent years. This highlights the importance of understanding long-term P surplus evolution to address persistent P-related environmental issues.

Overall, the dataset offers a valuable resource for researchers and policymakers, providing a detailed view of P dynamics that can be used to develop strategies for reducing environmental impacts while maintaining agricultural sustainability. Furthermore, by providing detailed components of the gridded P surplus budget—including P inputs (e.g., mineral fertilizers, manures) and P outputs — our databases are designed to be flexible and user-oriented. This flexibility, for example, enables users to conduct P surplus analysis focusing on the components that are dominant in agricultural areas. Future efforts to refine these estimates and incorporate additional data will further enhance the utility of the dataset for addressing persistent nutrient management challenges across Europe.

*Author contributions.* MB, FS and RK conceptualised the study and designed the methodology. MB led the analyses, collected and processed the data with inputs from FS and RK. MB produced the original draft of the paper that was reviewed and edited by all authors.

*Competing interests.* The contact author has declared that none of the authors has any competing interests.

*Acknowledgements.* Partial support for this work was provided by the Global Water Quality Analysis and Service Platform (GlobeWQ) project financed by the German Ministry for Education and Research (grant number 02WGR1527A) and the Development Bank of Saxony, Research Project Funding on Resilient Zero-Pollution Wastewater Systems in Climate Change – Case Study Saxony (Project No. 100669418). We are also thankful to UFZ for providing computing power and technical support to EVE super computing facility. We would like to thank people from various organizations and projects for kindly providing us with the data that were used in this study, which includes among others: FAO, Eurostat, HYDE, and IFA.

**Table 1.** Phosphorus (P) excretion coefficients and P/N ratios for various livestock categories as utilized in this study. The P excretion values are predominantly adapted from Sheldrick et al. (2003). For animal categories not listed in Sheldrick et al. (2003), (P) excretion coefficients are estimated based on the values of the P/N ratios as reported byLun et al. (2018).

| Livestock Category | P Excretion coefficients (kg head$^{-1}$ yr$^{-1}$) | P/N Ratio |
|---|---|---|
| Cattle | 10 | 0.18 |
| Pigs | 4 | 0.28 |
| Sheep | 2 | 0.15 |
| Goats | 2 | 0.19 |
| Horses | 8 | 0.19 |
| Chickens | 0.19[a] | 0.24[b] |
| Ducks | 0.09[c] | 0.25 |
| Mules | 8[d] | 0.19 |
| Buffaloes | 10[e] | 0.18 |
| Asses | 8[f] | 0.19 |
| Camels | 8[g] | 0.19 |

[a] The value for chickens is adopted from the "Poultry" category in Sheldrick et al. (2003).

[b] The P/N ratio for chickens is consistent for both layers and broilers, as per Lun et al. (2018), hence a single value is utilized.

[c] The excretion value for ducks is sourced from OECD 1997.

[d] The P/N ratio for mules, as per Lun et al. (2018), aligns with that of horses, justifying the identical P excretion rate adopted for mules in the absence of specific data from Sheldrick et al. (2003).

[e] Given the matching P/N ratios for buffaloes and cattle in Lun et al. (2018), the P excretion rate for buffaloes is inferred to be the same as that for cattle.

[f] The P excretion rate for asses is estimated to be equivalent to that of horses due to the lack of specific data.

[g] For camels, the P excretion rate is presumed to be the same as that for horses, based on identical P/N ratios found in Lun et al. (2018), compensating for the missing values in Sheldrick et al. (2003).

**Table 2.** Existing studies for P surplus

| Studies | Land type | Temporal feature | | Spatial feature | | |
|---|---|---|---|---|---|---|
| | | Extent | Resolution | Extent | Level | Resolution |
| Ludemann et al. (2023) | Cropland | 1961 – 2020 | Annual | Global | Country | - |
| Zou et al. (2022) | Cropland | 1961 – 2019 | Annual | Global | Country | - |
| Panagos et al. (2022b) | Agriculture | 2011 – 2019 | Mean | EU and UK | NUTS 2 | - |
| Eurostat (2024) | Agriculture | 2004 – 2019 | Annual | EU27, Norway, and Switzerland | Country | - |
| Einarsson et al. (2020) | Agriculture | 2013 | Snapshot | EU28 | NUTS 2 (except Germany which is at NUTS1) | - |
| Guejjoud et al. (2023) | Agriculture | 1990-2020 | Annual | France | NUTS 3 (90 geographic entities) | - |
| DEFRA (2022) | Agriculture | 1990, 1995 and 2000 – 2019 | Snapshots and Annual | United Kingdom | Country | - |
| Verbič and Sušin (2022) | Agriculture | 1992 – 2019 | Annual | Slovenia | Country | - |
| Antikainen et al. (2008) | Agriculture and Forest sectors | 1910 – 2000 | Annual | Finland | Country | - |
| Senthilkumar et al. (2012) | Agriculture | 1990-2006 | Annual | France | 21 regions | - |
| Lun et al. (2018) | Agriculture | 2002-2010 | Annual | Global | Country, regional, national | - |
| Muntwyler et al. (2024) | Agriculture | 1980-2019, 2019-2050 | Annual | EU27 and UK | Grid | 1 km$^2$ |
| Kopiński et al. (2006) | Agriculture | 2002-2004 | Annual | Poland | Regional and country level | - |
| Bergström et al. (2015) | Agriculture | 1960-2011 | Annual | Sweden | Country | - |
| Laruelle et al. (2014) | Agriculture soils, river basin | 1961-2009 | Annual | EU-27 and Seine River | Country, river basin | - |
| Özbek (2014) | Agriculture | 2011 | Snapshot | Turkey | NUTS2 | - |
| Ringeval et al. (2024) | Agriculture | 1900 - 2018 | Annual | Global | Grid | 0.5 degree |
| This study | Agriculture and non-agriculture | 1850 - 2019 | Annual | Europe | Grid | 0.083 degree |

**Table 3.** Components of P surplus in existing studies

| Studies | P inputs | | | | | P outputs | | | |
|---|---|---|---|---|---|---|---|---|---|
| | Fertilizer | Manure | Deposition | Weathering | Others | Non-fodder crop removal | Erosion | Leaching | Others |
| Ludemann et al. (2023) | ✓ | ✓ | | | | ✓ | | | |
| Zou et al. (2022) | ✓ | ✓ | | | | ✓ | | | |
| Panagos et al. (2022b) | ✓ | ✓ | ✓ | ✓ | | ✓ | ✓ | | ✓[a] |
| Kremer (2013) | ✓ | ✓ | ✓ | | ✓[b] | ✓ | | | ✓[c] |
| Einarsson et al. (2020) | ✓ | ✓ | | | | ✓ | | | |
| Guejjoud et al. (2023) | ✓ | ✓ | ✓ | | | ✓ | | | |
| DEFRA (2022) | ✓ | ✓ | ✓ | | ✓[d] | ✓ | | | ✓[a] |
| Verbič and Sušin (2022) | ✓ | ✓ | ✓ | | ✓[d] | ✓ | | | |
| Antikainen et al. (2008) | ✓ | ✓ | ✓ | ✓ | | ✓ | | | ✓[e] |
| Senthilkumar et al. (2012) | ✓ | ✓ | ✓ | | ✓[f] | ✓ | ✓ | ✓ | |
| Lun et al. (2018) | ✓ | ✓ | | | | ✓ | | ✓ | ✓[g] |
| Muntwyler et al. (2024) | ✓ | ✓ | | ✓ | | ✓ | ✓ | ✓ | ✓[a] |
| Kopiński et al. (2006) | ✓ | ✓ | | | ✓[h] | ✓ | | | |
| Bergström et al. (2015) | ✓ | ✓ | | | ✓[i] | ✓ | | | ✓[a] |
| Laruelle et al. (2014) | ✓ | ✓ | ✓ | | ✓[j] | ✓ | | | |
| Özbek (2014) | ✓ | ✓ | ✓ | | ✓[j] | ✓ | | | ✓[c] |
| Ringeval et al. (2024) | ✓ | ✓ | ✓ | | ✓[k] | ✓ | | | ✓[a] |
| This study | ✓ | ✓ | ✓ | ✓ | | ✓ | ✓ | | ✓[l] |

[a] Crop residues

[b] Net manure import/export, Other organic fertilizers (compost, sewage sludge, residues from biogas plants using crops, crop residues or grassland silage, industrial waste etc), Seed and planting material

[c] P removal from fodder crops, crop residues

[d] Seed and planting material

[e] Round wood harvest (forest), Net import of P embedded in internationally traded agricultural commodities, P in the human diet, detergent consumption

[f] Crop residues, compost, sludge, seed

[g] P emission from agriculture fires, P from household, bio energy

[h] Seed and tuber

[i] Seed, and sewage sludge

[j] Net import/export, withdrawal, stocks, other organic fertilizers

[k] P inputs from sludge

[l] P removal from fodder crops and pasture

**Table 4.** The range (minimum–maximum) of Phosphorus (P) content coefficients for the crops derived based on previous studies (Ludemann et al., 2023; Hong et al., 2012; Guejjoud et al., 2023; Panagos et al., 2022a; Einarsson et al., 2020; Lun et al., 2018; Zou et al., 2022; Antikainen et al., 2008).

| Crops | P content (kg t$^{-1}$ of product) (min. - max.) |
|---|---|
| Barley | 2.80 - 4.06 |
| Buckwheat | 1.60 - 3.90 |
| Maize | 1.80 - 3.40 |
| Millet | 2.60 - 4.20 |
| Oats | 3.10 - 4.39 |
| Potatoes | 0.54 - 3.30 |
| Pulses (total) | 3.53 - 4.97 |
| Rapeseed | 5.50 - 7.30 |
| Rice | 2.07 - 3.90 |
| Rye | 2.80 - 3.60 |
| Sesame Seed | 2.60 - 6.34 |
| Sorghum | 3.10 - 4.50 |
| Soybean | 4.40 - 8.10 |
| Sugar Beet | 0.40 - 1.74 |
| Sunflower Seed | 2.30 - 6.08 |
| Triticale | 2.80 - 4.20 |
| Wheat | 2.80 - 4.20 |

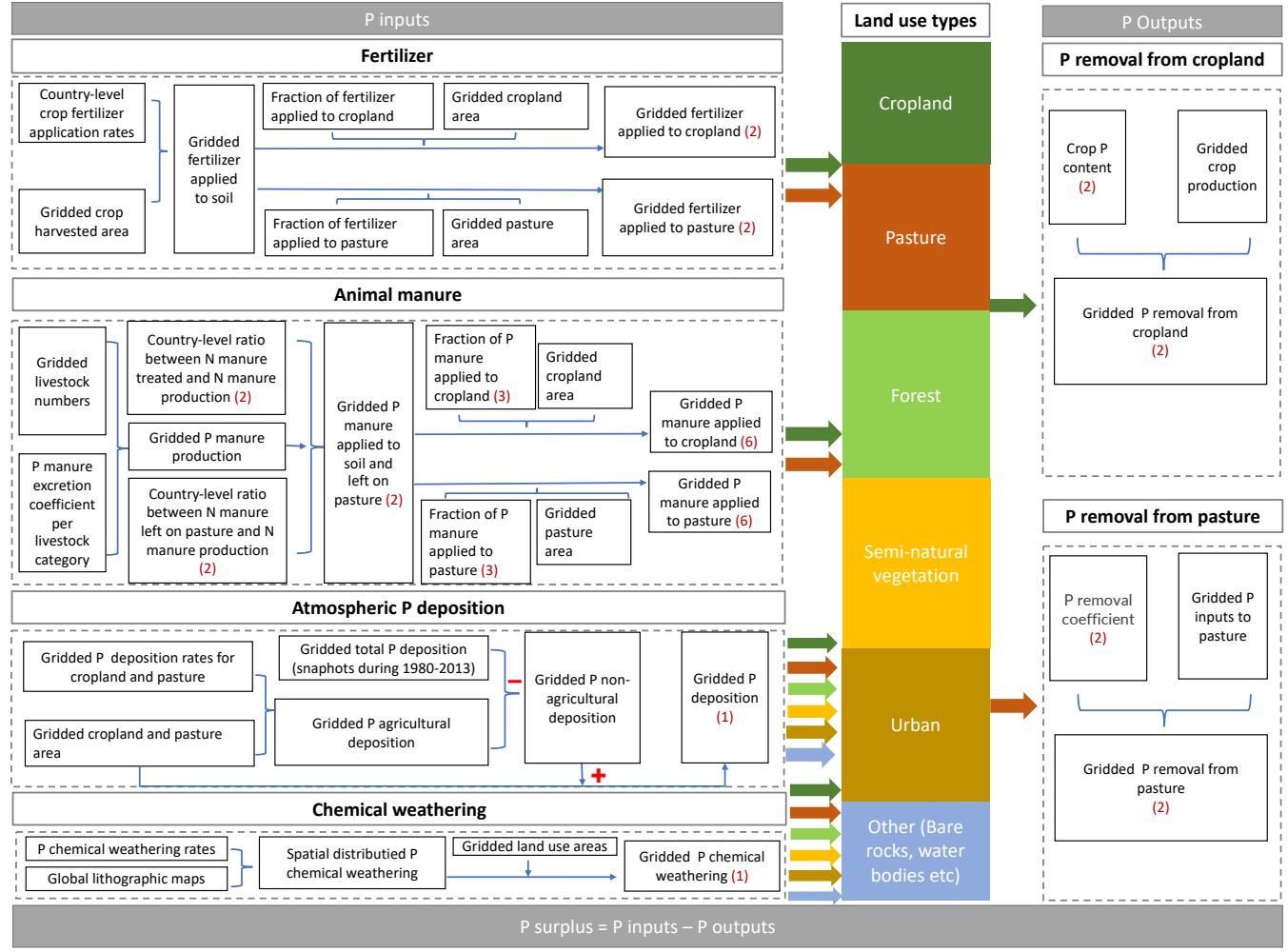

**Figure 1.** Workflow for constructing the long-term annual dataset of P surplus during the period 1850 – 2019. The number in brackets with red colors present different combination of datasets that we used to account for the uncertainties, resulting in 48 P surplus estimates. Arrow colors denote land use types: dark green (cropland), orange (pasture), light green (forest), yellow (semi-natural vegetation), brown (urban), and blue (other land uses such as bare rocks and water bodies).

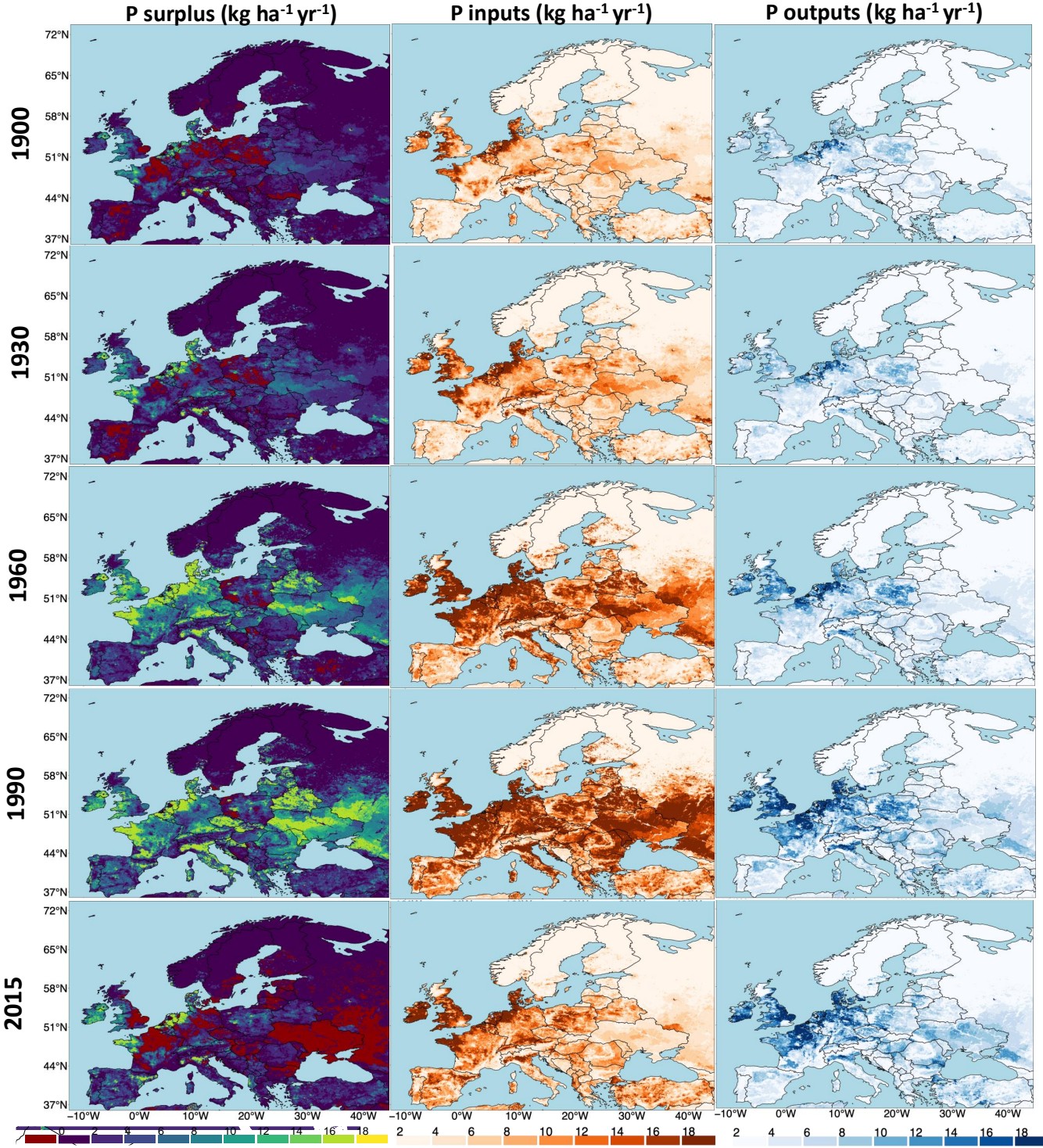

**Figure 2.** Snapshots of P surplus, P inputs and P outputs ($kg\ ha^{-1}$ of physical area $yr^{-1}$) across Europe. The figure shows the annual spatial variation in P surplus, P inputs and P outputs given as the mean of our 48 P surplus, P inputs and P outputs estimates for the selected years.

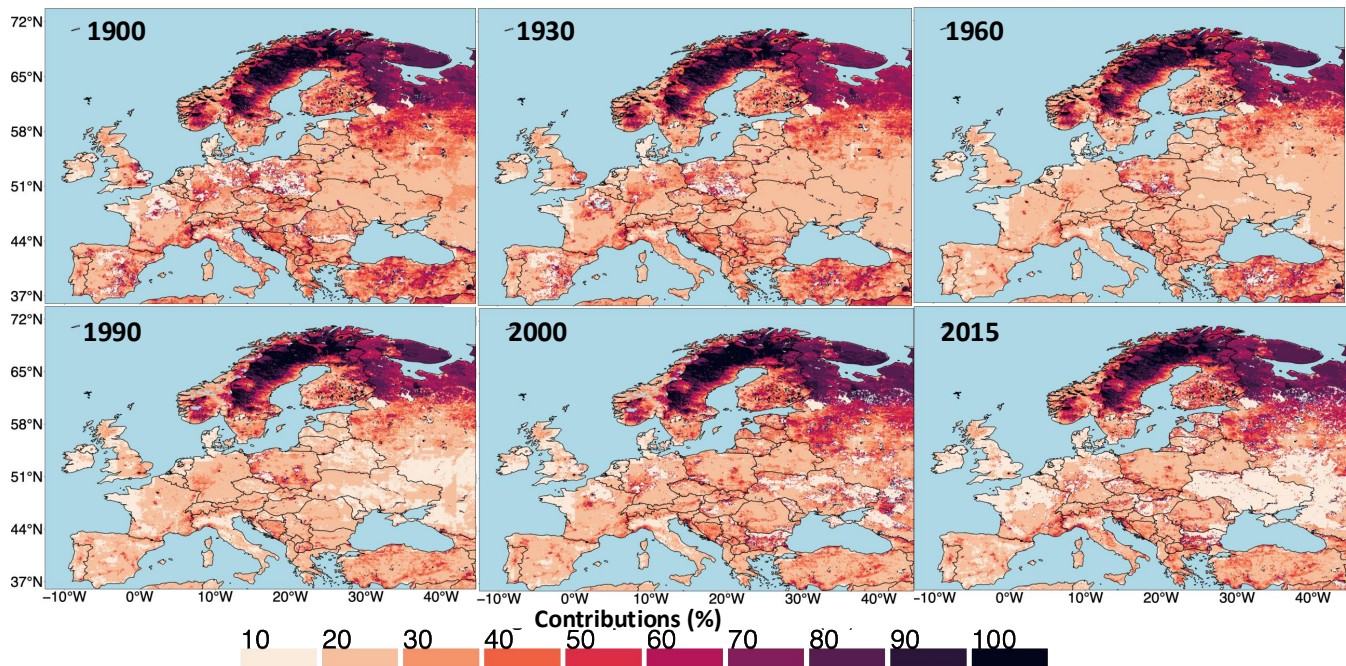

**Figure 3.** Snapshots showing the spatial distribution of the contribution (%) of non-agricultural P surplus to the total P surplus across Europe for selected years. The figure highlights the annual variation in the proportion of non-agricultural P surplus to the total P surplus (averaged from 48 P surplus estimates) across different regions, illustrating the evolving role of non-agricultural sources in European P dynamics over time.

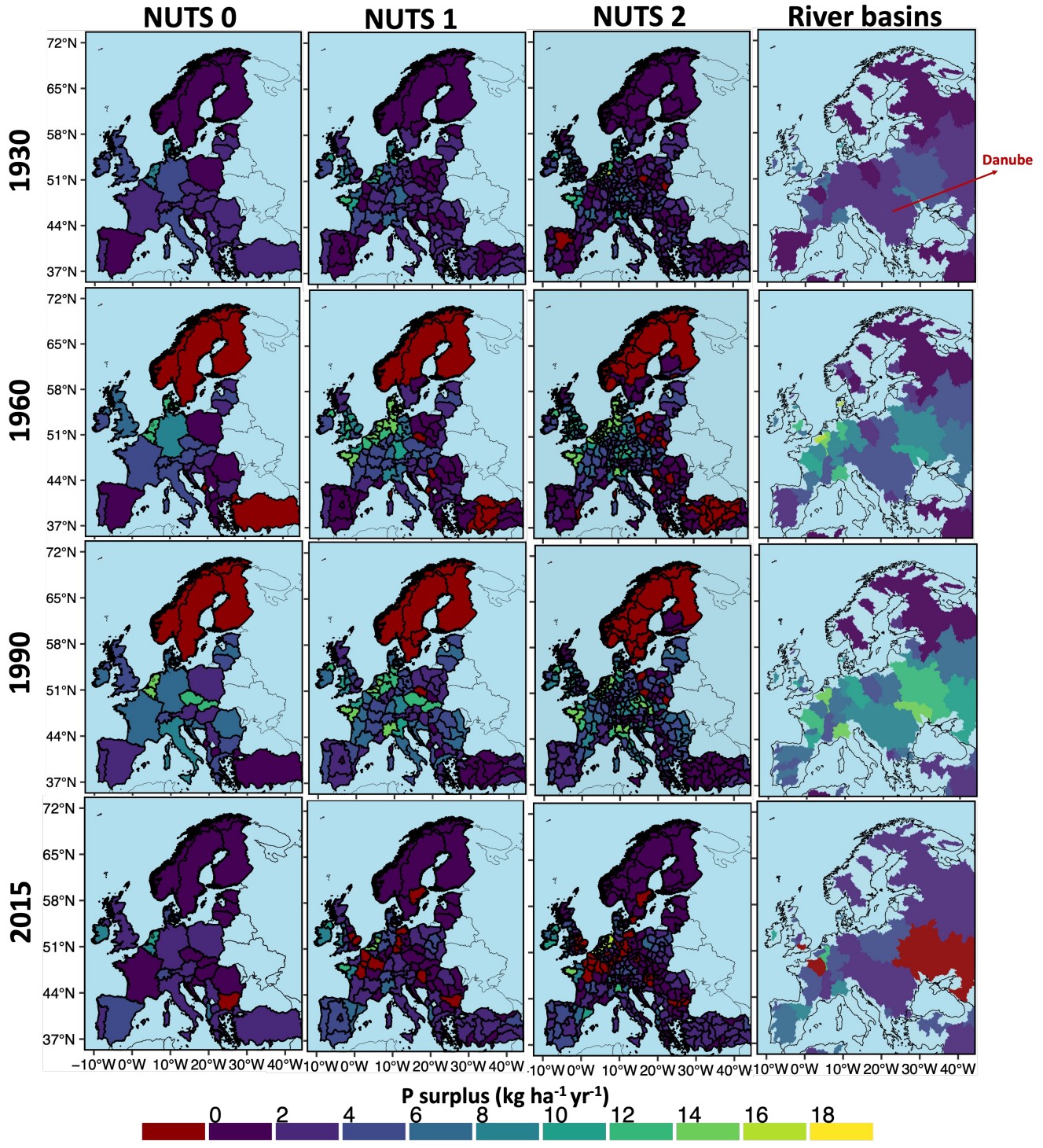

**Figure 4.** Total P surplus ($kg\ ha^{-1}$ of physical area $yr^{-1}$) at multiple spatial levels for four years (1930, 1960, 1990, 2015). P surplus is given as the mean of our 48 P surplus estimates. NUTS: Nomenclature of Territorial units for statistics

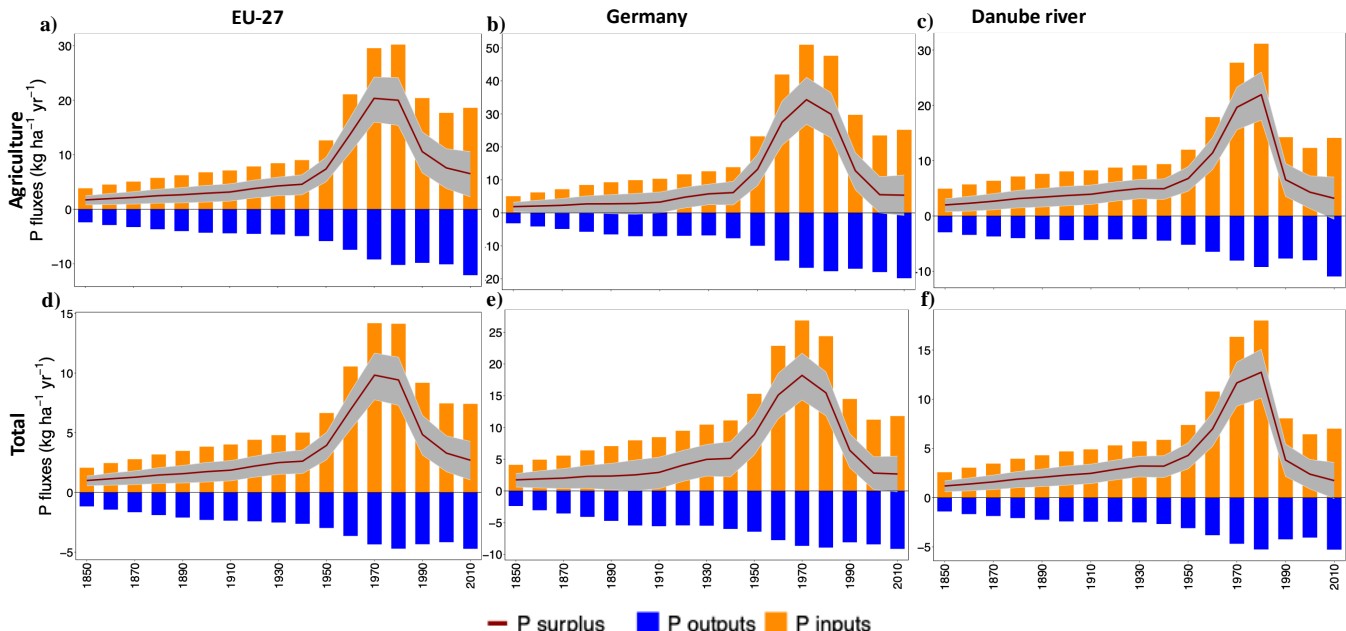

**Figure 5.** Decadal trajectory of agricultural P surplus ($kg\ ha^{-1}$ of agricultural area $yr^{-1}$) and total P surplus ($kg\ ha^{-1}$ of physical area $yr^{-1}$) and its contributing components for the EU-27, Germany, and the Danube river basin from 1850 to 2019. Upward orange bars represent the average of 48 P inputs, while downward blue bars indicate the average of 48 P outputs, showing decadal means. The grey ribbon shows the range (min and max) of the 48 P surplus estimates, with the red line representing the average value. **(a-c)** Agricultural P surplus for EU-27, Germany and Danube river, **(d-f)** Total P surplus for EU-27, Germany and Danube river

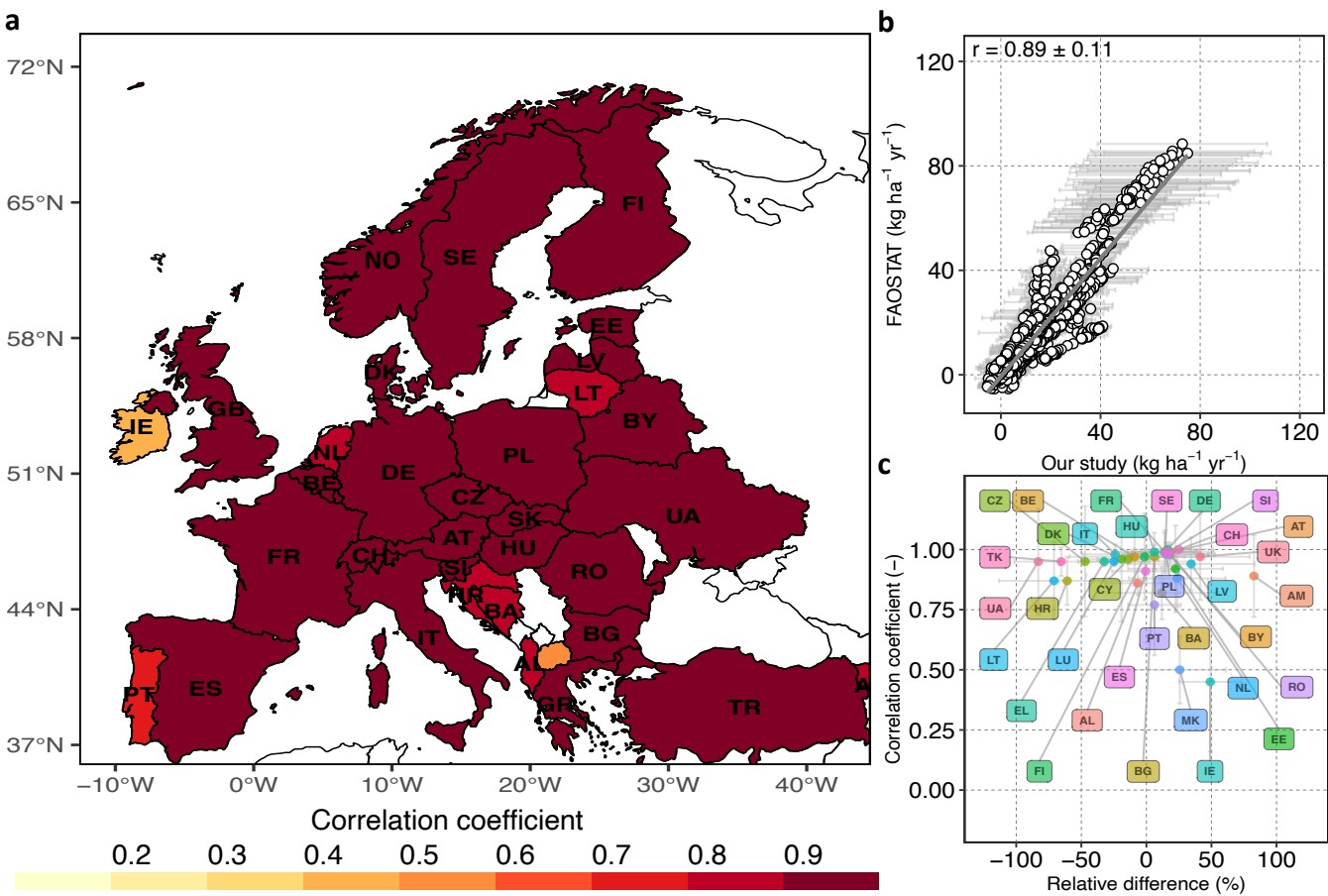

**Figure 6.** Country level comparison between P surplus for cropland estimated by Ludemann et al. (2023) and this study for the period 1961 – 2019. The circles in panels **b)** and **c)** denote the average of the 48 P surplus estimates reconstructed in this study, whereas the bars show the standard deviation. **(a)** Pearson correlations coefficients (r) values for every country. Countries with white color are excluded from the comparison because they are not part of the our dataset.**b)** Linear fit between the P surplus values for all countries and all years in the two studies: x-axis shows the P surplus calculated in this study and y-axis presents the P surplus given by Ludemann et al. (2023) **(c)** Relative difference (defined in equation 63) in P surplus in this study with respect to FAOSTAT against correlation coefficient for each country. Norway (NO) and Slovakia (SK) are not shown in this graph as they present an outlier value of -131% and -239% for the mean relative difference and an SD of 24% and 159% around the mean, respectively. However, the correlation for both Norway and Slovakia is high (0.99±0 for Norway and 0.97±0.02 for Slovakia).

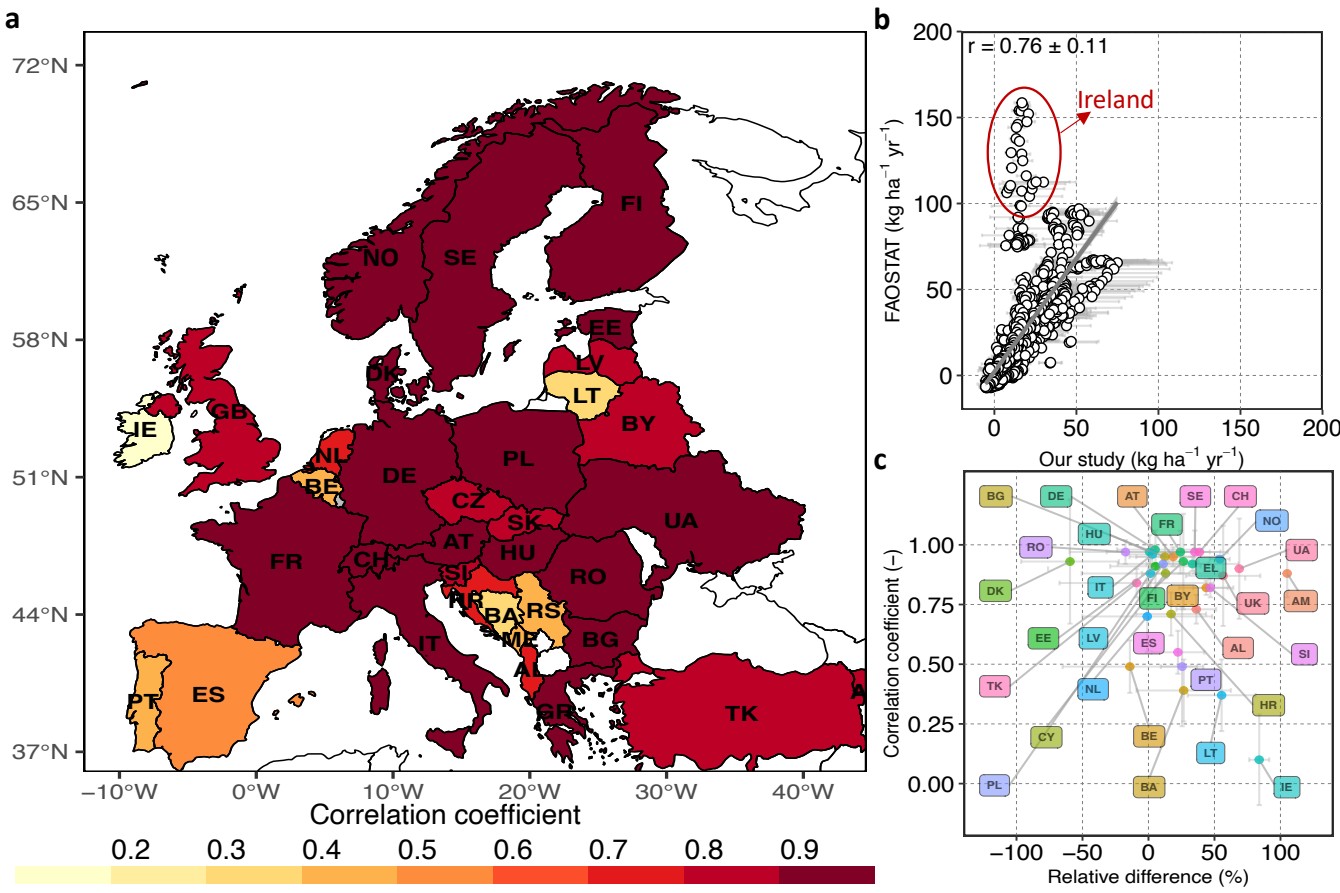

**Figure 7.** Country level comparison between P surplus for cropland estimated by Zou et al. (2022) and this study for the period 1961 – 2019. The circles in panels **b)** and **c)** denote the average of the 48 P surplus estimates reconstructed in this study, whereas the bars show the standard deviation. **(a)** Pearson correlations coefficients (r) values for every country. Countries with white color are excluded from the comparison because they are not part of the our dataset.**b)** Linear fit between the P surplus values for all countries and all years in the two studies: x-axis shows the P surplus calculated in this study and y-axis presents the P surplus given by Zou et al. (2022). The areas highlighted in red show the estimates for Ireland, which are relatively high compared to our estimates and represent outliers. **(c)** Relative difference (defined in equation 63) in P surplus in this study with respect to Zou et al. (2022) against correlation coefficient for each country. Serbia and Montenegro are not shown in this graph as they present an outlier value of -100% for the mean relative difference and an SD of 186% around the mean. However, the correlation for Serbia and Montenegro is moderate (0.53±0.31).

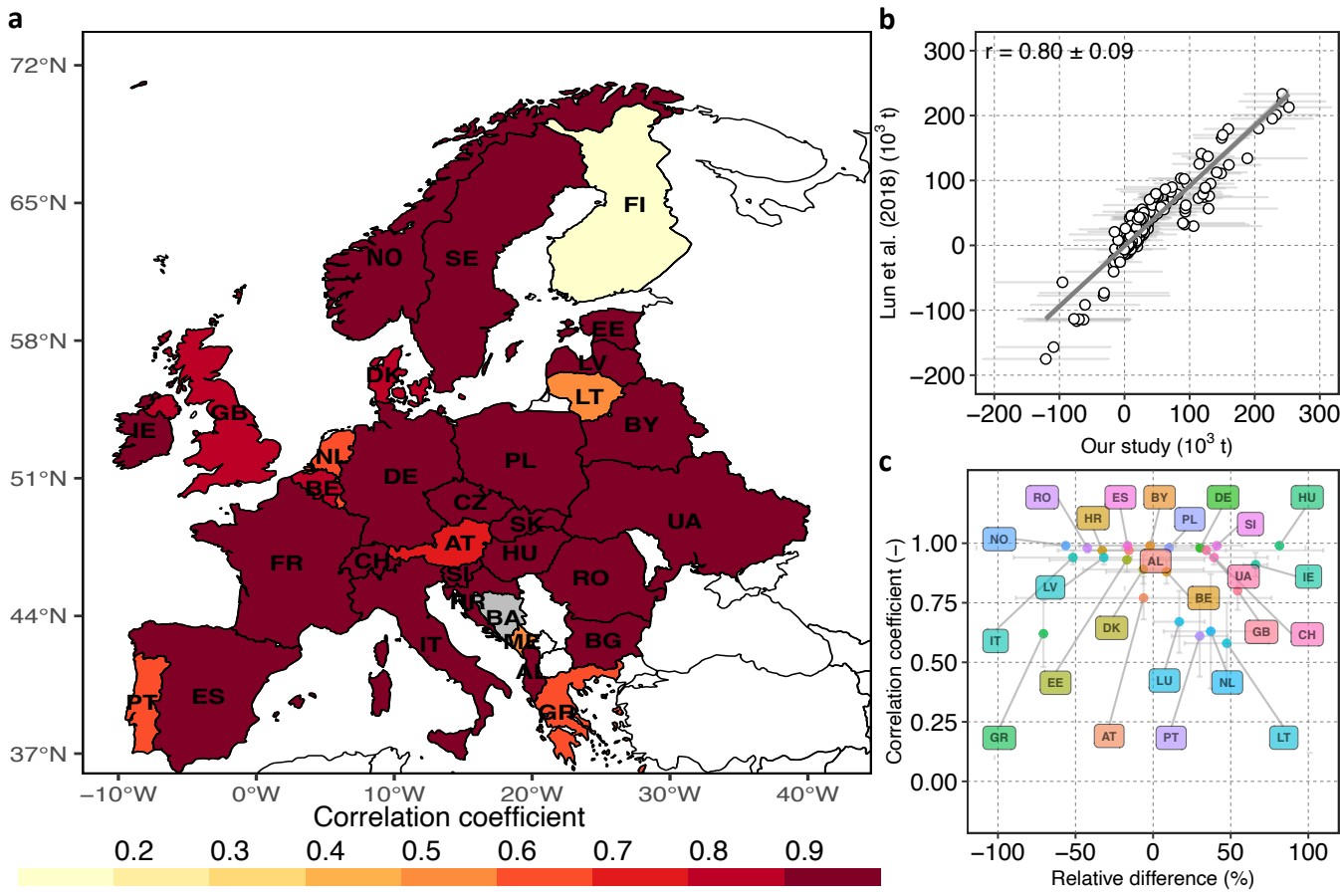

**Figure 8.** Country level comparison between P surplus for cropland estimated by Lun et al. (2018) and this study for the period 2002 – 2010. The circles in panels **b)** and **c)** denote the average of the 48 P surplus estimates reconstructed in this study, whereas the bars show the standard deviation. **(a)** Pearson correlations coefficients (r) values for every country. Countries with white color are excluded from the comparison because they are not part of the our dataset. **b)** Linear fit between the P surplus values for all countries and all years in the two studies: x-axis shows the P surplus calculated in this study and y-axis presents the P surplus given Lun et al. (2018). **(c)** Relative difference (defined in equation 63) in P surplus in this study with respect to Lu and Tian (2017) against correlation coefficient for each country. Countries with values below -100% are not listed as they present an outlier value, which includes France, Finland, Sweden and Bulgaria with a relative difference and SD around the mean of -271%±497, -184%±82, -169%±216 and -114%±126 respectively. However, the correlation for these countries is very high with a value of 0.95±0.02 for France, 0.97±0.02 for Sweden and 0.99 for Bulgaria, with the exception of Finland, which has an r-value of 0.110±0.34

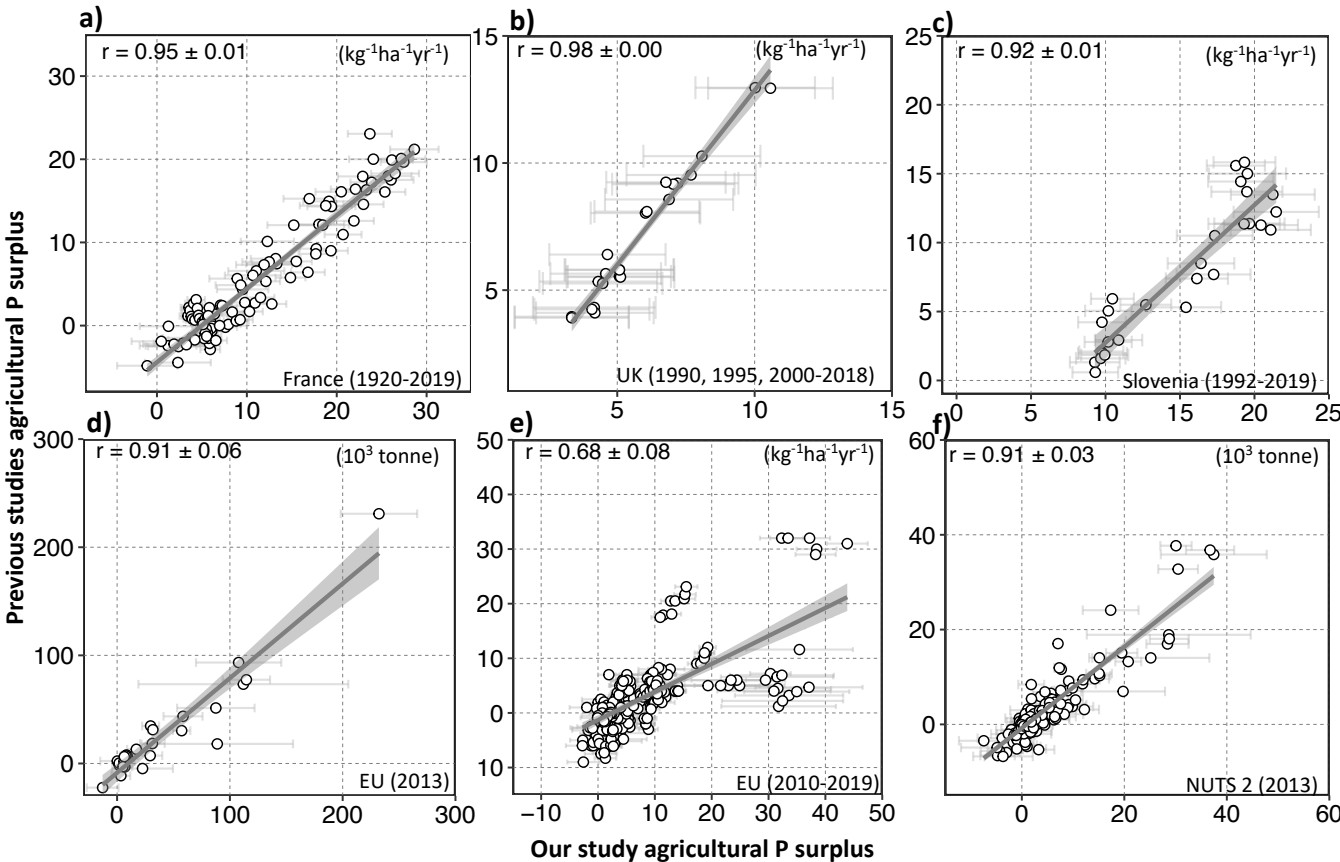

**Figure 9.** Scatter plots between the P surplus for agricultural soil from various previous studies and our study for different countries and corresponding linear fits. The circles in each panel denote the average of 48 P surplus reconstructed in this study, whereas the bars show the standard deviation. **(a)** France for each year in 1920 – 2019 at country level (Guejjoud et al., 2023) **(b)** United Kingdom (UK) for 1990, 1995 and each year in 2000 – 2018 at country level (DEFRA, 2022) **(c)** Slovenia for each year in 1992 – 2019 at country level (Verbič and Sušin, 2022) **(d)** European countries in 2013 (each point in this plot represent one country (Einarsson et al., 2020)) **(e)** Each European country for each year in 2010 – 2019 (Eurostat, 2024) **(f)** Sub-national level (NUTS 2) for the year 2013 (Einarsson et al., 2020) (each point in this plot represent NUTS 2 unit)

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

## Appendix A

Table A1: Table of variables used in the P surplus calculation and related sections, listing each variable, its description, and units. The variables cover multiple components of the phosphorus budget, including inputs, outputs, and surplus calculations across different land-use categories.

| Variable | Description | Unit |
|---|---|---|
| **General Variables** | | |
| $P$ | Phosphorus | - |
| $N$ | Nitrogen | - |
| $i$ | Grid cell index | - |
| $y$ | Year index | - |
| $u$ | Country index | - |
| $c$ | Crop index | - |
| $l$ | Livestock category index | - |
| $n_u$ | Number of grid cells in country u | - |
| $n_c$ | Number of grid cells for each crop category c | - |
| $n_l$ | Number of grid cells for each livestock category l | - |
| **P Surplus** | | |
| $Surp_{\text{soil}}$ | Total P surplus | $kg\ ha^{-1}$ of physical area $yr^{-1}$ |
| $Surp_{\text{agri}}$ | P surplus in agricultural areas (cropland and pasture) | $kg\ ha^{-1}$ of physical area $yr^{-1}$ |
| $Surp_{\text{NonAgri}}$ | P surplus in non-agricultural areas | $kg\ ha^{-1}$ of physical area $yr^{-1}$ |
| $Surp_{\text{cr}}$ | P surplus in cropland | $kg\ ha^{-1}$ of physical area $yr^{-1}$ |
| $Surp_{\text{past}}$ | P surplus in pasture | $kg\ ha^{-1}$ of physical area $yr^{-1}$ |
| $Inp_{\text{cr}}$ | Total P inputs in cropland | $kg\ ha^{-1}$ of physical area $yr^{-1}$ |
| $Out_{\text{cr}}$ | Total P removal from cropland | $kg\ ha^{-1}$ of physical area $yr^{-1}$ |
| $FERT_{\text{cr}}$ | Fertilizer P inputs in cropland | $kg\ ha^{-1}$ of physical area $yr^{-1}$ |
| $MAN_{\text{cr}}$ | Manure P inputs in cropland | $kg\ ha^{-1}$ of physical area $yr^{-1}$ |
| $Dep_{\text{cr}}$ | Atmospheric P deposition in cropland | $kg\ ha^{-1}$ of physical area $yr^{-1}$ |
| $CW_{\text{cr}}$ | P inputs from chemical weathering in cropland | $kg\ ha^{-1}$ of physical area $yr^{-1}$ |
| $Rem_{\text{cr}}$ | P removal by crop uptake from cropland | $kg\ ha^{-1}$ of physical area $yr^{-1}$ |
| $Inp_{\text{past}}$ | Total P inputs in pasture | $kg\ ha^{-1}$ of physical area $yr^{-1}$ |
| $Out_{\text{past}}$ | Total P removal from pasture | $kg\ ha^{-1}$ of physical area $yr^{-1}$ |
| $FERT_{\text{past}}$ | Fertilizer P inputs in pasture | $kg\ ha^{-1}$ of physical area $yr^{-1}$ |
| $MAN_{\text{past}}$ | Manure P inputs in pasture | $kg\ ha^{-1}$ of physical area $yr^{-1}$ |
| $Dep_{\text{past}}$ | Atmospheric P deposition in pasture | $kg\ ha^{-1}$ of physical area $yr^{-1}$ |
| $CW_{\text{past}}$ | P inputs from chemical weathering in pasture | $kg\ ha^{-1}$ of physical area $yr^{-1}$ |
| $Rem_{\text{past}}$ | P removal by grass uptake in pasture | $kg\ ha^{-1}$ of physical area $yr^{-1}$ |
| $Surp_{\text{For}}$ | P surplus in forest areas | $kg\ ha^{-1}$ of physical area $yr^{-1}$ |
| $Surp_{\text{NatVeg}}$ | P surplus in semi-natural vegetation areas | $kg\ ha^{-1}$ of physical area $yr^{-1}$ |
| $Surp_{\text{NonVeg}}$ | P surplus in non-vegetated areas | $kg\ ha^{-1}$ of physical area $yr^{-1}$ |
| $Surp_{\text{Urban}}$ | P surplus in urban areas | $kg\ ha^{-1}$ of physical area $yr^{-1}$ |
| $DEP_{\text{For}}$ | Atmospheric P deposition in forest areas | $kg\ ha^{-1}$ of physical area $yr^{-1}$ |

| | | |
|---|---|---|
| $CW_{\text{For}}$ | P inputs from chemical weathering in forest areas | $kg\ ha^{-1}$ of physical area $yr^{-1}$ |
| $DEP_{\text{NatVeg}}$ | Atmospheric P deposition in semi-natural vegetation areas | $kg\ ha^{-1}$ of physical area $yr^{-1}$ |
| $CW_{\text{NatVeg}}$ | P inputs from chemical weathering in semi-natural vegetation areas | $kg\ ha^{-1}$ of physical area $yr^{-1}$ |
| $DEP_{\text{NonVeg}}$ | Atmospheric P deposition in non-vegetated areas | $kg\ ha^{-1}$ of physical area $yr^{-1}$ |
| $CW_{\text{NonVeg}}$ | P inputs from chemical weathering in non-vegetated areas | $kg\ ha^{-1}$ of physical area $yr^{-1}$ |
| $DEP_{\text{Urban}}$ | Atmospheric P deposition in urban areas | $kg\ ha^{-1}$ of physical area $yr^{-1}$ |
| **Land use** | | |
| $R_{\text{HYDE-cr}}$ | Ratios represent the relative change (temporal variability) in cropland over time, normalized to the year 2000 | - |
| $A_{\text{HYDE-cr}}$ | Cropland area from HYDE | $ha$ |
| $R_{\text{HYDE-past}}$ | Ratios represent the relative change (temporal variability) in pasture area over time, normalized to the year 2000 | - |
| $A_{\text{cr}}$ | Cropland area in our study | $ha$ |
| $A_{\text{Ramankutty-cr}}$ | Cropland area from Ramankutty et al. (2008) | $ha$ |
| $A_{\text{past}}$ | Pasture area in our study | $ha$ |
| $A_{\text{Ramankutty-past}}$ | Pasture area from Ramankutty et al. (2008) | $ha$ |
| $R_{A_{\text{cr}}}$ | Ratios between cropland area from FAOSTAT and cropland area calculated in our study | - |
| $A_{FAO-\text{cr}}$ | Cropland area from FAOSTAT | $ha$ |
| $R_{A_{\text{past}}}$ | Ratios between pasture area from FAOSTAT and pasture area calculated in our study | - |
| $A_{FAO-\text{past}}$ | Pasture area from FAOSTAT | $ha$ |
| $A_{\text{cr}}^{\text{cor}}$ | Corrected cropland area (i.e. harmonized at country level with FAOSTAT country level cropland data) in our study | $ha$ |
| $A_{\text{past}}^{\text{cor}}$ | Corrected pasture area (i.e. harmonized at country level with FAOSTAT country level pasture area data) in our study | $ha$ |
| $A_{\text{agri}}^{\text{cor}}$ | Corrected agriculture area in our study | $ha$ |
| **Crop specific harvested area** | | |
| $R_A$ | Ratio between crop-specific harvested areas calculated in our study and crop-specific harvested areas from FAOSTAT | - |
| $A_{crops_{FAO}}$ | Crop-specific harvested areas from FAOSTAT | $ha$ |
| $A_{\text{crops}}$ | Crop-specific harvested areas in our study | $ha$ |
| $A_{\text{crops}}^{\text{cor}}$ | Corrected crop-specific harvested areas in our study | $ha$ |
| **Mineral Fertilizer** | | |
| $Pfer_{Missing_{East_{EU}}}$ | Total fertilizer application for the missing countries and time period from FAOSTAT | $kg\ yr^{-1}$ |
| $Pfer_{East_{EU}}$ | Total fertilizer application for Eastern European countries (Czechoslovakia, Hungary and Bulgaria) with available data in FAOSTAT | $kg\ yr^{-1}$ |
| $Pfer_{soil}$ | Total P fertilizer application | $kg\ yr^{-1}$ |
| $Pfer_{crops}$ | P fertilizer application for different non-fodder crop types | $kg\ yr^{-1}$ |
| $P_2O_5fer_{crops}$ | P fertilizer application in the form of phosphate for different non-fodder crop types from IFA | $kg\ yr^{-1}$ |
| $Pfer_{crops_{Rate}}$ | Fertilizer application rates for different non-fodder crop types | $kg\ ha^{-1}$ of crop harvested areas $yr^{-1}$ |
| $A_{crops_{FAO}}$ | Crop-specific harvested area for different non-fodder crop types from FAOSTAT | $ha$ |
| $Pfer_{grass_{Rate}}$ | Fertilizer application rates for grassland | $kg\ ha^{-1}$ of grassland areas $yr^{-1}$ |

| | | |
|---|---|---|
| $Pfer_{grass}$ | P fertilizer application for grassland (covering temporary and permanent grasslands and pastures for silage, hay, and grazing) from IFA | $kg\ yr^{-1}$ |
| $A_{grass_{FAO}}$ | Grassland area (including both permanent pastures and temporary grassland) from FAOSTAT | $ha$ |
| $Pfer_{past}$ | P fertilizer application for permanent pasture | $kg\ yr^{-1}$ |
| $Pfer_{fodder}$ | P fertilizer application for fodder crops | $kg\ yr^{-1}$ |
| $Pfer^{cor}_{crops_{Rate}}$ | Adjusted/corrected fertilizer application rates for crops | $kg\ ha^{-1}$ of crop harvested areas $yr^{-1}$ |
| $Pfer^{cor}_{grass_{Rate}}$ | Adjusted/corrected fertilizer application rates for grassland | $kg\ ha^{-1}$ of grassland areas $yr^{-1}$ |
| **Animal Manure** | | |
| $R_{man}$ | Ratio of animal manure production between 1850–1889 and the base year 1890 | - |
| $man$ | Animal manure production for a given country | $kg\ yr^{-1}$ |
| $L$ | Estimated livestock number for each livestock category in a country | $head$ |
| $Wratio$ | Weighted livestock ratio per grid cell | - |
| $P_{man}(l)$ | Total P manure production for each livestock category | $kg\ head^{-1}\ yr^{-1}$ |
| $P_{coeff}$ | P manure excretion coefficient for each livestock category | $kg\ head^{-1}\ yr^{-1}$ |
| $R_{P_{man}}$ | Country-level correction factor (as ratios) for P manure produced to match with FAO-STAT country-level estimates | - |
| $P_{man_{FAO}}$ | Total P manure production for each livestock category derived from FAOSTAT | $kg\ head^{-1}\ yr^{-1}$ |
| $P^{cor}_{man}$ | Adjusted/corrected P manure produced in our study to match with FAOSTAT country-level estimates | $kg\ head^{-1}\ yr^{-1}$ |
| $P_{man}$ | Total P manure production | $kg\ yr^{-1}$ |
| $R_{Nman_{treat/prod}}$ | Country-level ratios of treated N manure to total excreted N manure | - |
| $Nman_{treat}$ | Treated N manure | $kg\ yr^{-1}$ |
| $Nman_{prod}$ | Total N manure produced | $kg\ yr^{-1}$ |
| $R_{Nman_{left/prod}}$ | Country-level ratios of N manure left on pasture to total excreted N manure | - |
| $Nman_{left}$ | N manure left on pasture | $kg\ yr^{-1}$ |
| $Man_{treat}$ | Treated manure | $kg\ yr^{-1}$ |
| $Man_{left}$ | Manure left on pasture | $kg\ yr^{-1}$ |
| $Man_{cr}$ | Manure applied to cropland | $kg\ ha^{-1}$ of physical area $yr^{-1}$ |
| $Prop_{A_{cr}}$ | Proportion of cropland area in the total agricultural area in a grid cell | - |
| $A_{grid}$ | Area of grid cell | $ha$ |
| $Man_{app_{past}}$ | Manure applied to pasture areas | $kg\ ha^{-1}$ of physical area $yr^{-1}$ |
| $Prop_{A_{past}}$ | Proportion of pasture area in the total agricultural area in a grid cell | - |
| $Man_{past}$ | Total manure applied to pasture (sum of manure applied to pasture and manure left on pasture) | $kg\ ha^{-1}$ of physical area $yr^{-1}$ |
| **Chemical Weathering** | | |
| $CW_{Rate}$ | Rates of P release from chemical weathering | $kg\ ha^{-1}$ of physical area $yr^{-1}$ |
| $A_{For}$ | Forest area | $ha$ |
| $A_{NatVeg}$ | Semi-vegetation area | $ha$ |
| **P Removal** | | |
| $P_{content}$ | Crop-specific P content of each crop type | $kg\ t^{-1}$ |
| $Pro_{crops}$ | Crop production | $t\ yr^{-1}$ |
| $Pro_{crops_{Monfreda}}$ | Crop production derived from Monfreda et al. (2008) | $t\ yr^{-1}$ |

| $Rem_{past}^{coeff}$ | Coefficient for P removal in pasture | - |
| | | |