# Peer review of "Century Long Reconstruction of Gridded Phosphorus Surplus Across Europe (1850 – 2019)"

_Earth System Science Data, 2024_

## Author Comment (AC2)

**Response to the Reviewer**

PAPER:#essd-2024-294

TITLE: Century Long Reconstruction of Gridded Phosphorus Surplus Across Europe (1850–2019)

We appreciate constructive feedback of reviewer on our manuscript. We have carefully considered each comment and revised our manuscript accordingly. A point-by-point response to the reviewers' comments is provided below. All line numbers refer to the revised manuscript file. We hope that we have addressed all the comments satisfactorily.
* * *
**Reviewer 1**

*I recently read your preprint in Earth System Science Data titled "Century Long Reconstruction of Gridded Phosphorus Surplus Across Europe (1850 – 2019)". Nice topic of research. I wanted to request a few specific corrections regarding the citation of a paper.*

**Reply**: We thank the Reviewer for their positive comments to the manuscript. We reply below to the points raised by the Reviewer.

**1.1** — *Ensure that Muntwyler et al. (2024) is consistently cited as Muntwyler et al. (2024) in line 39 and Table 3, where currently 2023 is referenced.*

**Reply**: Thank you for this remark. We have updated the citation in the revised manuscript as *". . . Muntwyler et al. (2024)"* in line 39 and Table 3.

**1.2** — *Please add "Crop residues" to the description of paper (Muntwyler et al. (2024) in Table 3 to accurately reflect the removal of P from the budget with crop residues (you might also want to merge indices a and e in the same table to maintain consistency).*

**Reply**: We have revised the Table 3 by adding "Crop residues" to the "Other" columns for the study Muntwyler et al. (2024) to reflect the removal of P from the budget with crop residues. As suggested, we also have merged indices (a) and (e) in Table 3, which were otherwise repeated.

**1.3** — *Distinguish more clearly between the studies by Panagos et al. (2022a) and Einarsson et al. (2020) and Muntwyler et al. (2024). The methodologies of Muntwyler et al. (2024) and the purposes of the P budgets differ significantly; Muntwyler et al. (2024) is derived from a process-based model, whereas other two studies are empirical.*

**Reply**: Thank you for this suggestion. We have updated the sentence to make the differences between the studies clearer. In particular, we have revised the information on Muntwyler et al.

(2024). In this context, we have changed the text in line 39 of the revised manuscript, which now reads as follows:

*"At the European scale, Muntwyler et al. (2024) provide current (average 2011-2019) and future projections (2020-2029 and 2040-2049) of the P budget in agricultural soils at a higher spatial resolution of 1 $km^2$ using a process-based biogeochemical model (DayCent)".*

We also mentioned that the other two studies (Panagos et al. (2022a) and Einarsson et al. (2020)) are based on empirical methods (line 44).

**1.4** — Note that the scale of Panagos' phosphorus budget is at the NUTS2 level, only the soil stock resolution is 500m.

**Reply**: Thank you for this comment. We have updated the "Spatial feature" (i.e. instead of 500m we have replaced it with NUTS 2) of Panagos et al. (2022b) in Table 2.

---

## Author Comment (AC3)

**Reviewer 1**

**Overall comment:** This study reconstructs and analyzes long-term phosphorus (P) surplus in Europe from 1850 to 2019 at a high spatial resolution, providing a comprehensive dataset that includes 48 P surplus estimates and accounts for various uncertainties. The findings reveal that P surplus across the EU-27 has tripled over 170 years, highlighting the need for targeted management strategies. Additionally, the study identifies big variation in P surplus estimates by comparing its database with major previously published P databases and highlights areas for improvement. This research contributes to the improvement of quantification of historical and spatial P budgets in Europe by considering varying methods, including more budget terms, extending the temporal span of current studies, increasing resolution, and comparing various datasets. However, a few comments regarding the calculation details and discussions are provided for the authors' consideration.

**Reply**: We thank the Reviewer for their positive comments to the manuscript. We reply below to the points raised by the Reviewer.

**1.1** — Section 2.1:

– A more detailed definition of each rate term needs to be provided wherever a rate appears. Since the rates are always expressed in $kg\ ha^{-1}\ yr^{-1}$ in this study, is the area being referred to the grid area or a specific land type area? Only part of the rate terms in this manuscript indicate which area is used as the denominator.

**Reply**: Thank you for this valuable feedback. In response to the suggestion, we have thoroughly reviewed and updated the manuscript to provide clearer definitions of all rate terms. Specifically, we have clarified the land use area associated with each rate term to ensure a consistent and precise interpretation of phosphorus (P) surplus values and associated components. Throughout the manuscript, we now explicitly state the specific land use areas (such as grid physical area, agricultural area, cropland, or pasture) that correspond to the units presented.

The rates are now expressed as follows:

– $kg\ ha^{-1}\ yr^{-1}$ of physical area for grid-level estimates,
– $kg\ ha^{-1}\ yr^{-1}$ of agricultural area for agricultural land,
– $kg\ ha^{-1}\ yr^{-1}$ of cropland area for croplands,
– $kg\ ha^{-1}\ yr^{-1}$ of pasture area for permanent pastures, and
– $kg\ ha^{-1}\ yr^{-1}$ of grassland area for grassland (both temporary and permanent pastures).

For instance, in line 9 of the revised manuscript, we have updated the phrasing to explicitly refer to "physical area" as follows: *"Specifically, the total P surplus across the EU-27 has tripled over 170 years, from 1.19 ($\pm$0.28) $kg\ ha^{-1}$ of physical area in 1850 to around 2.48 ($\pm$0.97) $kg\ ha^{-1}$ of physical area $yr^{-1}$ in recent years"*. This updated phrasing ensures that readers are aware of the specific land use area considered in the corresponding rate calculations. We have made similar adjustments throughout the manuscript wherever needed to ensure that the respective land type (physical area, cropland, pasture, grassland, or agricultural area) is clearly specified.

- Since more definitions of land types and other budget terms are provided in the following sections, I suggest that the authors move Section 2.1 to the end of Section 2.

**Reply**: Thank you for your valuable feedback regarding the structure of Section 2. We understand your suggestion to move Section 2.1, but we believe that the current organization is structured in a way to retain the overall clarity and coherence of the manuscript. To this end, our intention was to provide an initial overview of phosphorus (P) surplus, presenting both agricultural and non-agricultural contexts, to establish a basic foundation for the subsequent detailed definitions and methodologies. We believe that this approach allows readers to grasp the broader significance of P surplus before delving into the specific details of land use classifications and P budget components.

By outlining the P surplus early in the manuscript, we guide readers through a step-wise progression, starting with the general concept of P surplus and then moving into more specific discussions about land use types and detailed methodological steps. This structure, in our view, ensures a more intuitive understanding of the complex processes and terms discussed in the manuscript.

To take this point into consideration and to better clarify this structure, we added an introductory texts to the paragraph starting at line 105 in section 2 of the revised manuscript, in which the flow of information is explained as follows:

*"In the following sections, we first outline the definition of P surplus in both agricultural and non-agricultural soils. Next, we present a summary of the methodology used to reconstruct the land use types, including agricultural land, namely cropland and pasture, and non-agricultural land, including non-vegetated areas, semi-natural vegetation, forest, and urban areas. Crop-specific harvested areas for non-fodder and fodder crops are also defined. We refer to Batool et al. (2022) for detailed methodologies. Finally, we describe the steps employed to reconstruct P inputs, including fertilizer, manure, atmospheric deposition, and chemical weathering, and P outputs, focusing on P removal from cropland and pastures."*

**1.2 — Section 2.3:**

- Line 215: Do you mean the IFA report published in 2017 instead of 2013 that contains 2014-2015 data?

**Reply**: Thank you for pointing this out this typing error. We have used the IFA report published in 2017 and therefore updated the reference in line 274 of the revised manuscript.

- Section 2.3.3: Not all terms in each equation are clearly defined in the text after the equation, in equations 15-23. For example, what is P2O5fercrops in Equation 15, what is nu and i in equations 22-23? It may seem redundant since some of the terms are discussed before the equations, but it would be easier for readers who are not familiar with these terms if each term's definition is clearly listed after the equations. Or, since this manuscript includes many parameters, it would be very helpful to include a table at the end listing all parameters' names, definitions, units, and/or calculation methods.

**Reply**:

Thank you for the insightful suggestion regarding the clarity of the terms in the equations. Based on your suggestion, we have reviewed and updated the manuscript to ensure that each term in equations 15–23 (now 27-35 in the revised manuscript) (and others) is clearly defined in the text following the equations.

Further, we have incorporated a **detailed table, Appendix Table A1 in the revised manuscript** that lists all variables, including their description and units, as suggested by the reviewer. We have also attached the Table R1 below for your quick reference. We believe these adjustments will clarify the manuscript and aid readers in following the notations.

[revised manuscript text omitted]

- Line 230: The fertilizer application rates here are defined as total fertilizer use/harvest area. I was wondering if the authors use the same definition for all fertilizer application rates discussed in the study, such as the data from Batool et al. (2022) mentioned later, and in equations 18-20. It seems that the fertilizer application rates are defined as total fertilizer use/land use area in equations 18-20. However, land use areas may not correspond to harvest areas.

**Reply**: Thank you for this remark. We appreciate the opportunity to clarify our approach to defining and applying fertilizer application rates for different land-use types in our study. We consistently used two distinct methods for the application rates based on the land-use type:

- **Non-Fodder Crops:** For non-fodder crops, fertilizer application rates ($Pfer_{crops_{Rate}}$) ($kg\ ha^{-1}$ of crop harvested areas $yr^{-1}$) are calculated as total fertilizer use ($Pfer_{crops}$) ($kg\ yr^{-1}$) divided by the corresponding harvested area ($A_{crops}$) ($ha$) for each crop type ($c$) and country ($u$) for the year 2015 ($y_{2015}$). This approach ensures that the fertilizer rate corresponds specifically to the harvested area of each non-fodder crop, as shown in equation 28 of the revised manuscript (equation R1 here):

$$Pfer_{crops_{Rate}}(u,c,y_{2015}) = \frac{Pfer_{crops}(u,c,y_{2015})}{A_{crops_{FAO}}(u,c,y_{2015})} \tag{R1}$$

- **Grasslands (temporary and permanent):** For grasslands, which include both temporary grassland (represented as one of the six fodder crop categories) and permanent pasture, we calculated a single fertilizer application rate ($Pfer_{grass_{Rate}}$) ($kg\ ha^{-1}$ of grassland areas $yr^{-1}$) by combining IFA fertilizer usage data ($Pfer_{grass}$) ($kg\ yr^{-1}$) with the corresponding grassland area ($A_{grass_{FAO}}$ ($ha$). In this case, the total grassland area reflects both temporary and permanent pasture lands used for grazing and forage production in equation 29 of the revised manuscript (equation R2 here):

$$Pfer_{grass_{Rate}}(u,y_{2015}) = \frac{Pfer_{grass}(u,y_{2015})}{A_{grass_{FAO}}(u,y_{2015})} \tag{R2}$$

- **Application in equations 18–20 (equations 30–32 in Revised Manuscript):** In equation 18 (equation 30 in revised manuscript and equation (R3) here), we applied the country-level fertilizer rates of non-fodder crops ($Pfer_{crops}$) ($kg\ ha^{-1}$ of crop harvested areas $yr^{-1}$) to gridded harvested areas for non-fodder crops ($A_{crops}^{cor}$) ($ha$) for each year to derive fertilizer application amounts ($Pfer_{crops}$) ($kg\ yr^{-1}$) per grid cell for each crop.

$$Pfer_{crops}(i,c,y_{1850,-,2019}) = Pfer_{crops_{Rate}}(u,c,y_{2015}) \times A_{crops}^{cor}(i,c,y1850-2019) \tag{R3}$$

In equation 19-20 (equations 31-32 in revised manuscript and equations (R4–R5) here), we applied the country-level fertilizer rates of grassland to both fodder crop harvested areas and permanent pasture areas. Specifically, for permanent pasture and fodder crops, the fertilizer application rate ($Pfer_{grass_{Rate}}$ (($kg\ ha^{-1}$ of grassland areas $yr^{-1}$))) was multiplied by the gridded pasture area and harvested area of each fodder crop

$(A_{\text{past}}^{\text{cor}}$ and $A_{\text{fodder}}$ in $ha$, respectively) to estimate the fertilizer application amount for pasture and fodder crops $(Pfer_{\text{past}}$ and $Pfer_{\text{fodder}}$, in $kg \ yr^{-1})$, respectively).

$$Pfer_{\text{past}}(i, y_{1850-2019}) = Pfer_{grass_{Rate}}(u, y_{2015}) \times A_{\text{past}}^{\text{cor}}(i, y_{1850-2019}) \tag{R4}$$

$$Pfer_{\text{fodder}}(i, c, y_{1850-2019}) = Pfer_{grass_{Rate}}(u, y_{2015}) \times A_{fodder}(i, c, y_{1850-2019}) \tag{R5}$$

We realize that these distinctions may not have been sufficiently emphasized in the original text. To clarity this, we have updated the manuscript to specify that fertilizer rates are based on crop-specific harvested areas. For clarity, we have revised the text in section 2.3.3 at line 290 of the revised manuscript:

*We calculated fertilizer application rates by combining IFA fertilizer usage data with FAOSTAT's crop-specific harvested area ($A_{crops_{FAO}}$ ($ha$)) and grassland area ($A_{grass_{FAO}}$ ($ha$)). Grassland areas encompass temporary grasslands (represented as one of the six fodder crop categories) as well as permanent pastures, with a single fertilizer application rate applied consistently across all grassland uses for grazing and forage production.*

For further clarification, we have added the following text in line 303:

*Further, it is important to note that fertilizer application rates for non-fodder crops are based on fertilizer use per unit of corresponding harvested area to represent crop-specific fertilizer inputs. For grassland, encompassing both temporary and permanent pastures, the fertilizer application rate is calculated using total grassland area, and ensuring consistent application of this rate across all relevant grassland areas.*

We hope that the above clarifications address potential confusion and ensure transparency regarding how fertilizer application rates are defined and applied to different land-use categories in our analysis.

– Equations 37-38: If the unit of Pman is $kg \ yr^{-1}$ , why are the units of Mantreat and Manleft (=Pman*ratio) $kg \ ha^{-1} \ yr^{-1}$ ? Were any details omitted from the equations? Is the gridded area or the cropland+pasture area used as the denominator to calculate the application rate? It seems that you used the grid's physical area as the denominator based on Line 365. However, based on your Equations 41-42, it seems that the country total cropland+pasture area is used as the denominator (since you assume that "the entire amount of treated P manure is applied to soil" according to line 355 and "equal distribution rates for cropland and pasture within each grid cell" in Line 380)?

**Reply**: We appreciate your careful review of equations 37-38 (now 51-52 of the revised manuscript) and your comments regarding the units of $P_{man}$, $Man_{treat}$, and $Man_{left}$. Upon further inspection, we acknowledge that there was an inconsistency in how these units were represented. In the original equations, $Man_{treat}$ and $Man_{left}$ were incorrectly expressed in $kg \ ha^{-1} \ yr^{-1}$ rather than $kg \ yr^{-1}$.

In the revised version of the manuscript, we have updated this by correcting the text in section 2.3.8 at line 451-460. To clarify, the manure applied to cropland ($Man_{cr}$) ($kg \ ha^{-1}$ of physical area $yr^{-1}$) is calculated by dividing the total treated manure ($Man_{treat}$) ($kg \ yr^{-1}$) by the grid's physical area ($A_{\text{grid}}$) ($ha$) and multiplying it by the proportion of cropland within the grid cell ($Prop_{A_{cr}}$) (-), as outlined in equation R6 (equation 53 of the revised manuscript). Similarly, the manure designated for pasture application ($Man_{app_{past}}$) ($kg \ ha^{-1}$ of physical area $yr^{-1}$) is determined by dividing the treated manure ($Man_{treat}$) ($kg \ yr^{-1}$) by the grid's physical area ($A_{\text{grid}}$)($ha$) and multiplying it by the proportion of pasture area within the grid cell ($Prop_{A_{\text{past}}}$) (-), as indicated in equation R7 (equation 54 of the revised manuscript). Finally, the total manure allocated to pastures ($Man_{past}$) ($kg \ ha^{-1}$ of physical area $yr^{-1}$) is calculated by summing the manure applied to pastures ($Man_{app_{past}}$) ($kg \ ha^{-1}$ of physical area $yr^{-1}$) and the manure left on pastures by grazing animals ($Man_{left}$) ($kg \ yr^{-1}$) normalized by the grid's physical area ($A_{\text{grid}}$) ($ha$) as shown in equation R8 (equation 55 of the revised manuscript).

$$Man_{cr}(i, y_{1850-2019}) = \frac{Man_{treat}(i, y_{1850-2019}) \times Prop_{A_{cr}}(i, y_{1850-2019})}{A_{grid}(i)}, \tag{R6}$$

$$Man_{app_{past}}(i, y_{1850-2019}) = \frac{Man_{treat}(i, y_{1850-2019}) \times Prop_{A_{past}}(i, y_{1850-2019})}{A_{grid}(i)}, \tag{R7}$$

$$Man_{past}(i, y_{1850-2019}) = Man_{app_{past}}(i, y_{1850-2019}) + \frac{Man_{left}(i, y_{1850-2019})}{A_{grid}(i)}. \tag{R8}$$

These adjustments maintain consistency in the units and accurately reflect the distribution of treated manure between cropland and pasture. We hope these clarifications resolve the confusion, and we have also included further explanations in line 451-460 for each variable and equation 53-55 in the revised manuscript.

Thank you again for pointing this out, and we have addressed the issue in both the equations and the corresponding text in the revised manuscript.

– Lines 430-435: You may want to add explanations for CW, C, and A after the equations or refer to a table listing all parameters. Do the area parameters C and A correspond only to the areas that are both of a specific land use type and have chemical weathering?

**Reply**: Thank you for highlighting this point. Upon revising the manuscript, we recognized potential confusion with the area parameters, particularly with the notation for corrected/normalized areas. Previously, we used $C_A$ to denote corrected or normalized areas (i.e., our gridded cropland and pasture areas corrected at the country level to align with FAOSTAT's country-level estimates). To improve clarity, we have now revised this notation to $A_{cr}^{cor}$ throughout the manuscript (for instance, in equations 20-21 and subsequent equations), ensuring that $A$ **consistently represents area terms**, and the "cor" subscript indicates correction.

To clarify on the parameters specifically asked in your comments:

– **CW** refers to the P inputs from chemical weathering, expressed in $kg\ ha^{-1}$ of physical area $yr^{-1}$.

– **A** represents the gridded area for each specific land-use type (cropland, pasture, forest, natural vegetation, and grid area), denoted as $A_{cr}^{cor}$, $A_{past}^{cor}$, $A_{For}$, $A_{NatVeg}$, and $A_{grid}$ (in $ha$), respectively. These parameters represents the area of a particular land-use type within each grid cell or the grid physical area.

In response to your feedback on line 430-435 (now 525-528 in the revised manuscript), we have added explanations for CW and A directly following the equations in the revised manuscript to make them clearer. Furthermore, we have revised equations 44-48 (now 56-59 in the revised manuscript, shows as equations R9- R12 below) to normalize these values using the grid's physical area, providing P inputs in units of $kg\ ha^{-1}$ of physical area $yr^{-1}$, which accurately reflects spatial estimates.

The updated equations (56-59) and subsequent text in lines 530-534 now read as follows:

$$CW_{cr}(i, y_{1850-2019}) = \frac{A_{cr}^{cor}(i, y_{1850-2019}) \times CW_{Rate}(i)}{A_{grid}(i)} \tag{R9}$$

$$CW_{past}(i, y_{1850-2019}) = \frac{A_{past}^{cor}(i, y_{1850-2019}) \times CW_{Rate}(i)}{A_{grid}(i)} \tag{R10}$$

$$CW_{For}(i, y_{1850-2019}) = \frac{A_{For}(i, y_{1850-2019}) \times CW_{Rate}(i)}{A_{grid}(i)} \tag{R11}$$

$$CW_{NatVeg}(i, y_{1850-2019}) = \frac{A_{NatVeg}(i, y_{1850-2019}) \times CW_{Rate}(i)}{A_{grid}(i)} \tag{R12}$$

*where $CW_{cr}$, $CW_{past}$, $CW_{For}$, and $CW_{NatVeg}$ refer to P inputs from chemical weathering for areas covered by cropland, pasture, forest, and natural vegetation, respectively, in $kg\ ha^{-1}$ of physical area $yr^{-1}$; $A_{cr}^{cor}$, $A_{past}^{cor}$, $A_{For}$, $A_{NatVeg}$, and $A_{grid}$ represent the gridded areas of cropland, pasture, forest, natural vegetation, and the grid's physical area, respectively, in $ha$; and $CW_{Rate}$ denotes the P release rate from chemical weathering, based on lithological data, in $kg\ ha^{-1}$ of physical area $yr^{-1}$.*

To further assist the reader, we have included all variables in the Appendix table A1 in the revised manuscript, where each variable, its description, and its unit are listed.

**1.3** — Sections 2.4.1-2.4.2:

    – It seems that Section 2.4.2 would be better placed before Section 2.4.1.?

**Reply**: Thank you for your suggestion regarding the order of Sections 2.4.1 and 2.4.2. We understand the logic behind presenting 'Crop Production' first, as it forms the basis for calculating P removal from pastures. However, we have deliberately structured the manuscript to first describe P removal from pasture, which utilizes crop production data, followed by the detailed description of how we constructed the crop production dataset. This flow ensures that the methodological steps are clearly outlined and remain logically connected. We hope the reviewer now recognizes our viewpoint regarding this structural decision, which supports the overall clarity of the presentation.

**1.4** — Section 3.1:

    – Including a series of maps showing accumulated P surplus in the SI could be informative in illustrating how much P has accumulated or been lost in each area.

**Reply**: Thank you for this suggestion. In response, we have added figures R1(as Supplementary Figure S4). In figure R1, each panel illustrates the spatial distribution of P fluxes across Europe, normalized per year for the selected time intervals to allow spatial and temporal comparisons. Here, along with the P surplus, we also depict the accumulation of corresponding P-inputs and P-outputs. The first row presents accumulated P surplus, the second row illustrates P inputs, and the third row shows P outputs. The four columns correspond to four distinct periods: 1850–1920, 1921–1960, 1961–1990, and 1991–2019. These periods represent key phases in agricultural and environmental history, reflecting pre-modern agriculture, industrialization before the Green Revolution, the Green Revolution and synthetic fertilizer expansion, and the phase of environmental awareness and policy intervention. The graph illustrates key shifts in P flux dynamics, particularly the significant rise in P inputs after 1960 and the notable gap between inputs and outputs, which has driven the P surplus across Europe.

We also have incorporated this in line 618 under section 3.1 in the revised manuscript which reads as follows:

*Furthermore, cumulative P fluxes, including P surplus, inputs, and outputs, are presented for four distinct time periods, which we term as following: (i) 1850–1920 (Pre-modern agriculture), (ii) 1921–1960 (Industrialization before the Green Revolution), (iii) 1961–1990 (Green Revolution and synthetic fertilizer expansion), and (iv) 1991–2019 (Environmental awareness and policy intervention phase) (Supplementary Figure S4). These plots revealed marked shift in P dynamics across Europe over time. During 1850-1920, P surplus was relatively low, averaging 8-10 $t\ yr^{-1}$ in much of the Central and Eastern Europe, with some Western Europe regions like France, the Netherlands, and Denmark exceeding 16 $t\ yr^{-1}$. Northern Europe typically showed much lower values of 2-4 $t\ yr^{-1}$. In the subsequent period (1921–1960), P inputs began to rise modestly, averaging 50-70 $t\ yr^{-1}$, driven by early industrialization and chemical fertilizer use, though P surplus remained moderate due to relatively high P outputs. The*

*Green Revolution period (1961–1990) saw a sharp increase in P inputs, exceeding 80 $t\ yr^{-1}$ in many regions due to agricultural intensification, resulting in substantial P surplus, with most areas surpassing 18 $t\ yr^{-1}$. In the most recent phase (1991–2019), P inputs declined steadily due to improved agricultural practices and environmental policies like the EU Nitrates Directive, while P outputs increased, narrowing P surplus. In some Western and Eastern Europe, P surplus even turned negative, reflecting P mining. These temporal and spatial trends highlight the importance of sustainable nutrient management practices and policies in reducing P surplus over time. Moving forward, strategies like reallocating nutrients inputs based on regional needs, improving the integration of crop and livestock systems could help to further optimize nutrient use efficiency. Such measures, coupled with continued monitoring of P indicators-P surplus and PUE- are essential to address P-related environmental challenges and promote sustainable agricultural practices (Zou et al., 2022).*

 **1.5** — Figure 4:

– Are there any hypotheses that could explain the peaks in P surplus around 1980 and the subsequent decrease afterward?

**Reply**: Thank you for your question regarding the peaks in P surplus around 1980 and the subsequent decline. To provide clarity, we have added Figure R2 (corresponding to Supplementary Figure S5), which illustrates along with P surplus their inherent major components, i.e. P inputs from fertilizer and manure and P outputs across various European countries. In general, as shown in this figure, P surplus dynamics closely follow the temporal variability in mineral fertilizer inputs across most countries, with a notable peak around 1980. This peak might be linked to the Green Revolution, a period of agricultural intensification that likely drove a significant increase in synthetic fertilizer use, including phosphorus, to meet growing food demands (Müller-Karulis et al., 2024). The trend in P outputs—i.e., P harvested in crops and grasslands—also shows an increasing trend over time (Supplementary Figure R2), reflecting the need to meet growing crop demand. The interplay between these P inputs and outputs jointly determines the observed behavior in P surplus.

The subsequent decline in P surplus, as shown in Figure R2, could be associated with several factors. In Western Europe, policy changes such as the Nitrates Directive (91/676/EEC) European Commission (2000b), which limits manure application to 170 $kgN\ ha^{-1}$ and consequently restricts P inputs, along with the Water Framework Directive (Directive 2000/60/EC) (European Commission, 2000a) and the recent Farm to Fork Strategy (European Commission) as part of the EU Green Deal (European Commission, 2019) that restricts P inputs (manure and mineral fertilization) (Amery and Schoumans, 2014) may have encouraged more sustainable nutrient management practices, and likely reduced P applications and stabilizing P surplus (Ludemann et al., 2023). Further regional/country-specific legislation also plays a role since a few European countries, including the Netherlands, Ireland, Norway, and Sweden, have specific legislation limiting P applications (Bouraoui et al., 2011). In some cases, the decrease in P surplus began even earlier, as in Denmark and the UK, where P was not a major limiting factor for crop yield since soil P levels had likely reached sufficient levels for crop production without additional inputs (Bouraoui et al., 2011).

In Central and Eastern European countries, the collapse of the Soviet Union in the late 1980s and early 1990s and subsequent (agro-)economic restructuring may have led to a sharp reduction in P inputs, as noticed by the abrupt decline in fertilizer inputs (Figure R2) (Ludemann et al., 2023). This adjustment likely reflects economic changes that reshaped agricultural practices in the region (Csathó et al., 2007; Ludemann et al., 2023). Around this time, awareness was also increasing about the environmental consequences of unsustainable agricultural practices, which may have further motivated efforts to reduce P surplus (Cassou, 2018). For instance, in the late 1980s, the European Union introduced policies that shifted away from direct agricultural subsidies, instead offering incentives aligned with environmental goals (Cassou, 2018).

Overall, the distinct behavior observed across European countries likely reflect a combination of policy, economic pressures, and agricultural adaptations that have collectively influenced P surplus trends across the continent. Such factors are also reflected in a recent study by Zou et al. (2022) at a global scale, which attempts to differentiate and attribute the factors governing the dynamics of long-term P surplus evolution across different countries. Socioeconomic factors such as fertilizer affordability, the intensification of agricultural systems and changes in farm size are among others significant factors governing distinct P evolutions across European countries (Zou et al., 2022).

We would also like to emphasize that our current study is primarily focused on building a database, while in-depth analyses of evolving trends/patterns and underlying reasoning for these trends are left for future studies. Nevertheless, in response to the reviewer's suggestions, we have included brief information on this aspect in line 635-650 under section 3.1 in the revised manuscript:

*The peak in P surplus observed around 1980 likely aligns with the intensified fertilizer use of the Green Revolution (Supplementary Figure R2). The subsequent decline in P surplus after 1990 reflects multiple factors, including policy shifts in Western Europe (e.g., Nitrate Directive (Directive 91/676/EEC) (European Commission, 2000b) and Water*

265 *Framework Directive (Directive 2000/60/EC) (European Commission, 2000a)), regional legislations that restricted P fertilization (Amery and Schoumans, 2014)), economic adjustments, and increased awareness of sustainable nutrient management (Ludemann et al., 2023; Senthilkumar et al., 2012; Cassou, 2018). Country-specific legislation also played a role, since a few European countries, including the Netherlands, Ireland, Norway, and Sweden, have specific legislation limiting P applications (Bouraoui et al., 2011). In some cases, the decrease in P surplus began*

270 *even earlier, as in Denmark and the UK, where P was not a major limiting factor for crop yield since soil P levels had likely reached sufficient levels for crop production without additional inputs (Bouraoui et al., 2011). On the other side, in Central and Eastern European regions, the collapse of the Soviet Union and subsequent (agro-)economic restructuring may have led to reduced P inputs, as indicated by a sharp drop in fertilizer use (Csathó et al., 2007; Ludemann et al., 2023) (Supplementary Figure R2) and subsequently reflected in corresponding P*

275 *surplus budgets. Such distinct P surplus patterns observed across Europe appear to have been shaped by these combined influences, and disentangling the different factors will require careful consideration in future studies. On a global scale, Zou et al. (2022) discussed the distinct roles of socioeconomic and environmental factors governing the dynamics of long-term P surplus evolution across different countries.*

**1.6** — Section 3.2: In addition, are there any reasons that can explain the varying uncertainty range over time based on your
280 modeling methods? For example, could it be due to changes in crop portfolios over time or differences in data availability? Understanding these reasons could provide insights into improving the modeling in the future. The discussion can be included in a discussion section after the results.

**Reply**: Thank you for your insightful comments and for highlighting the importance of understanding temporal variability in uncertainty for our P surplus estimates. Indeed, we observe varying level of uncertainty, likely driven by shifts in crop
285 portfolios, data availability, and evolving agricultural management practices (e.g., manure and mineral fertilizer applications) across different periods. To address these points, we conducted an analysis of P surplus uncertainty across various European countries from 1850 to 2019, as detailed below and supported by four supplementary figures.

To ascertain the evolving nature of uncertainty in P surplus over time, we calculated the coefficient of variation (CV, %) as the ratio of the standard deviation to the mean based on 48 P surplus estimates. Figure R3 (Supplementary
290 Figure R3) illustrates CV for P surplus across various European countries, revealing trends in uncertainty that fluctuate across different time periods. In the early years (1850–1930), countries like Germany, France, and the United Kingdom show high CV values, largely due to the low magnitude of P surplus. From 1930 to 1990, there is a notable decrease in CV for most countries, including Germany and the Netherlands, which aligns with advances in agricultural practices during the Green Revolution. By 1990, CV values had stabilized around 20–30% in countries like France and Denmark.
295 In recent decades, however, CV has modestly increased in some countries. For example, since 1990, Italy, Denmark, Slovakia, Slovenia, and some Eastern European countries have shown slight increases in CV.

The uncertainty in our P surplus budget largely stems from the combined effects of differences in crop P content coefficients, manure databases, and application rates of mineral fertilizers and manures in pasture and croplands. These factors vary across both space and time, as reflected by the evolution of different crop types and their productivity, as well
300 as changes in land cover areas over time, i.e., the areas covered by croplands and pastures. Ascertaining the individual contributions of different sources of uncertainty requires careful consideration, including analyses across different European regions and contexts, as well as their evolving nature over time. We have conducted a preliminary investigation to understand the contributing and driving factors, which provides an initial basis for future detailed analyses. To this end, we analyzed the uncertainty ranges (calculated as maximum minus minimum) of key components and examined their
305 relationships with P surplus, as shown in Figures R3- R3. For most countries, differences in the mineral fertilizer inputs show the strongest association with P surplus uncertainty, with relatively high $R^2$ values across different European Countries. For example, Eastern and Central European countries like Germany, Slovakia, Slovenia, Latvia, and Poland, as well as Mediterranean countries such as Spain, Italy, and Portugal, where $R^2$ values range between 0.7 and 0.9 (Figure R4, Supplementary Figure R3). This indicates that fluctuations in fertilizer inputs are a significant contributor to P surplus
310 uncertainty over time. On the other side, manure inputs display a more variable relationship with P surplus uncertainty, showing an overall relatively weaker associations than fertilizer in most countries (Figure R5). However, in livestock-intensive countries such as Ireland and the Netherlands, differences in manure inputs emerge as a major contributor to P surplus uncertainty, with $R^2$ values of 0.71 and 0.95, respectively, compared to 0.56 for association with fertilizer inputs in both countries. Another key contributor to the uncertainty in the P surplus budget is the variation in P outputs, reflected
315 in P crop removal coefficients. This component showed moderate to strong associations with P surplus uncertainty in countries such as Germany, France, Spain, the UK, the Netherlands, and Italy,indicating that P-output variability also substantially impacts P surplus uncertainty, especially in areas with high agricultural productivity (Figure R6).

Our analysis focused on assessing the uncertainty in key factors contributing to P surplus, examining each factor individually. However, we acknowledge that multiple factors often interact in complex ways, influencing P surplus collectively.
320 Addressing these complexities would require more detailed statistical approaches, such as those employed by (Zhang et al., 2021), which we leave for future consideration. To summarize, based on our analysis, we find that fertilizer inputs emerge as the primary driver of P surplus uncertainty. However, P outputs and manure inputs also play significant roles, with their influence varying according to regional agricultural practices. By highlighting key regions and components, our

study provides a foundation for future efforts to improve the database and refine the understanding of P surplus dynamics. To reflect these aspects, we have revised the section 3.1 by adding the following texts in line 702-721:

*To assess the uncertainty in P surplus estimates, we calculated the coefficient of variation (CV, %), defined as the ratio of the standard deviation to the mean across our 48 P surplus estimates. This analysis, shown in Supplementary Figure S10, offers insights into how relative uncertainty has evolved over time. The CV was highest in the early period (1850–1920) for many countries, including Germany and France and then declined significantly during the mid-20th century (1950–1990). However, in recent decades, relative uncertainty has increased again, especially in countries like Spain and Italy.*

*In addition, we examined the absolute uncertainty ranges (calculated as maximum minus minimum) of P surplus estimates for each year, comparing these against the ranges of key components, including fertilizer, manure, and P output (Supplementary Figures S11–S13). The results indicate that in central, eastern, and Mediterranean countries such as Germany, Spain, Italy, Slovakia, Slovenia, Poland, and Portugal, fertilizer input uncertainty aligns closely with P surplus uncertainty, identifying fertilizer as potentially a primary driver of variation in these regions (Supplementary Figure S11). In contrast, manure inputs show a more variable relationship with P surplus uncertainty across countries, with generally weaker associations than fertilizer. However, in livestock-intensive regions such as Ireland and the Netherlands, manure uncertainty strongly contributes to P surplus variation (Supplementary Figure S12). For P outputs, associations with P surplus uncertainty are moderate to strong in countries including Germany, France, Spain, the UK, the Netherlands, and Italy, suggesting that output variability also plays a role in P surplus uncertainty, especially in areas with high agricultural productivity (Supplementary Figure S13). Overall, fertilizer inputs emerge as the dominant factor influencing P surplus uncertainty, although the impact of P outputs and manure inputs also varies by region, reflecting distinct agricultural practices. These preliminary findings emphasize the substantial spatial and temporal variability in P surplus uncertainties and underscore the value of ensemble datasets in capturing comprehensive nutrient flows. Further statistical analyses would be required to investigate the factors controlling the uncertainties in P surplus in future studies.*

Furthermore, we have included the following texts in line 931 under the section 5 for the future improvement section of the revised manuscript:

*Improving the historical land-use and livestock data is essential for refining long-term phosphorus (P) surplus estimates. Currently, the dataset applies uniform land-use data throughout the study period, but incorporating detailed historical land-use reconstructions, such as changes in cropland expansion, pasture reduction, or urbanization, would enhance the spatial accuracy of P surplus calculations. Additionally, livestock and manure data from the mid-20th century introduce uncertainties, particularly in estimating manure production and distribution. A more detailed reconstruction of historical livestock numbers, along with records on manure management systems, would increase reliability. As part of this effort, promoting standardized data collection and reporting methods across European countries would further enhance data accuracy and consistency. Engaging with agricultural practitioners, policymakers, and environmental organizations could help refine data collection methods, ensure alignment with user needs, and expand the dataset's applicability for practical use. Further explorations of contributing parameters in priority areas could also help guide future updates.*

We appreciate your valuable feedback, which prompted us to explore the contributing factors to the uncertainty of our P surplus estimates. Although our investigation has been preliminary in nature, we hope it provides insights into the roles of different factors contributing to P surplus uncertainty and highlights regions where future efforts can focus on further improving the presented database.

**1.7** — Section 3.2:

- Some of the calculation details could be mentioned in the Methods section.

**Reply**: Thank you for your feedback. We have significantly revised the manuscript and hope it meets your expectations. Regarding the suggestion to include calculation details in the Methods section, it is unclear which specific calculations is referred to here. Please provide further guidance, and we will be happy to address them in the revised manuscript. However, we have now added explanations about land use and harvested area calculations, which were previously only referred to Batool et al. (2022), to provide more clarity and detail in the Methods section.

**1.8** — Lines 645-650:

- The discussion starting here can be included in a new Discussion section. Since your database only includes a limited number of scenarios and your parameter values are borrowed from previous studies, it would be informative to include a brief discussion that lists the key limitations of your database and suggests ways to improve them in the future, instead of simply providing some limitation examples. Other limitations could include, for example, the temporal and spatial variation of parameter values that are not/partially included in the current database and the uncertainty in applying country-level data to a high-resolution map.

**Reply**: Thank you for your valuable suggestion regarding a more comprehensive discussion and limitations section. We have now added a dedicated *Potential use and limitations of the dataset* section to the revised manuscript (refers to section 4 of the revised manuscript). Therein, we describe key limitations in our approach, such as the temporal and spatial uncertainties before 1961, assumptions related to land-use distributions, fertilizer, and manure applications, and variability in P removal calculations. We also identify and outline potential avenues to improve the dataset in the future (see section 5 of the revised manuscript), including enhancing data collection methods, refining assumptions, and conducting more detailed sensitivity analyses.

**1.9** — Technical corrections:

- Abstract & Introduction Line 15: It would be helpful to include a short note defining 'P surplus' (similar to the definitions you provided in Lines 30, 35 and 55) upon its first appearance in the abstract and introduction for readers who are unfamiliar with the term or have different definitions for it.

**Reply**: Thank you for this comment. We have added the definition of P surplus as the difference between P inputs and P outputs with the term P surplus when it first appears in the Abstract (at line 3) and in Introduction section (at line 19).

- Abstract line 5-10: Does the area in ha refer to the country's total land area or specifically to agricultural land?

**Reply**:

Thank you for this suggestion. We have updated the sentence in the Abstract to make the units clearer. In particular, we have added that the ha refer to the total physical area at EU27 level. In this context, we have changed the text in the lines 8-10 of the revised manuscript, which now reads as follows:

*"Specifically, the total P surplus across the EU-27 has tripled over 170 years, from 1.19 ($\pm$0.28) $kg\ ha^{-1}$ of physical area in 1850 to around 2.48 ($\pm$0.97) $kg\ ha^{-1}$ of physical area $yr^{-1}$ in recent years".*

– Line 35: Provide the full name of FAOSTAT the first time it appears.

400     **Reply**: The full name of FAOSTAT has been added as "Food and Agriculture Organization Corporate Statistical Database" when it first appears in line 36 of the revised manuscript.

– Line 55: You may want to move 'phosphorus (P)' to the first time the letter 'P' appears in the manuscript. In addition, there are multiple places in the manuscript where the same abbreviations are explained repeatedly (e.g., 'phosphorus (P)' appears multiple times).

405     **Reply**: Thank you for this suggestion. We have revised the manuscript to use "P" repeatedly instead of "phosphorus (P)" wherever appropriate. In some instances, we retain the full structure of "phosphorus (P)", such as in table and figure captions, to emphasize it for quick reference to clarify what "P" stands for.

– Figure 1: I can guess what the arrows in different colors represent, but it would improve clarity if the authors also mention their meanings in the caption.

410     **Reply**: Thank you for your comment. We have revised the caption of Figure 1 to clarify the meaning of the colored arrows, which now explicitly correspond to different land use types. Specifically, we added the following line at the end of the caption in the revised manuscript:

[revised manuscript text omitted]

**List of Figures**

[Figure]

**Figure R1.** Cumulative total P surplus, P inputs, and P outputs over four historical periods across Europe. The first row shows the accumulated phosphorus (P) surplus, the second row displays P inputs, and the third row illustrates P outputs across Europe for four distinct periods, which we term as following: (i) 1850–1920 (Pre-modern agriculture), (ii) 1921–1960 (Industrialization before the Green Revolution), (iii) 1961–1990 (Green Revolution and expansion of synthetic fertilizers), and (iv) 1991–2019 (Environmental awareness and policy intervention phase). All values are normalized per year within each time period, with units in tonnes per year.

[Figure]

**Figure R2.** Time series of phosphorus (P) inputs from fertilizer and manure, P outputs, and P surplus ($kg\ ha^{-1}$ of physical area $yr^{-1}$) across various European countries from 1850 to 2019. This figure highlights changes in P fluxes over time, showing a peak in P surplus around 1980 followed by a decline after 1990 for most countries. These patterns illustrate the influence of agricultural intensification during the Green Revolution, as well as subsequent policy, economic, and environmental shifts in both Western and Eastern Europe. The red line represents the mean of 48 P surplus estimates, while green, yellow, and blue lines depict temporal dynamics for P inputs from fertilizer, manure, and P outputs, respectively. Together, these pattern provide insight into how various factors may have influenced P surplus dynamics over time.

[Figure]

**Figure R3.** Mean and coefficient of variations (CV, %) of 48 total P surplus estimates across different European countries. The grey ribbon shows the range (min and max) of the 48 P surplus estimates, with the red line representing the average value. The CV is depicted by dashed lines.

[Figure]

**Figure R4.** Scatter plot showing the relationship between uncertainty ranges (calculated as the maximum minus minimum) in P fertilizer applied to cropland (x-axis) and total P surplus (y-axis) across different European countries from 1850 to 2019. Each point represents the annual range of fertilizer inputs and corresponding P surplus for a specific country and year. The linear trend line for each country indicates the strength and direction of the association, with a steeper slope suggesting a greater influence of fertilizer input variability on P surplus uncertainty. This plot highlights the variability in P surplus associated with fertilizer input uncertainty across regions and underscores the role of fertilizer as a major contributor to P surplus fluctuations.

[Figure]

**Figure R5.** Scatter plot showing the relationship between uncertainty ranges (calculated as the maximum minus minimum) in P manure applied to cropland (x-axis) and total P surplus (y-axis) across different European countries from 1850 to 2019. Each point represents the annual range of manure inputs and corresponding P surplus for a specific country and year. The trend line for each country illustrates the association strength, with a steeper slope indicating a stronger influence of manure input variability on P surplus uncertainty. This plot highlights the variability in P surplus associated with fluctuations in manure inputs, particularly in regions with significant livestock production.

[Figure]

**Figure R6.** Scatter plot illustrating the relationship between uncertainty ranges calculated as the maximum minus minimum) in P outputs/P removal from cropland (x-axis) and P surplus (y-axis) across various European countries from 1850 to 2019. Each point represents the annual range of P outputs and the corresponding P surplus for a specific country and year. Trend lines are shown for each country, with steeper slopes suggesting a stronger influence of P output variability on P surplus uncertainty. This plot underscores the extent to which changes in P output (e.g., crop removal) contribute to fluctuations in P surplus, particularly in agriculturally intensive regions.

[Figure]

**Figure R7.** Fractions of manure distribution to cropland based on different methods utilized in this study. Method 1 represents the fraction of distribution of animal manure to cropland based on the equal distribution rates for cropland and pasture within each grid cell. Method 2 shows the fraction using the time-varying national proportions of nitrogen (N) manure applied to both cropland and pasture, as provided by Einarsson et al. (2021). Method 3 shows the manure distribution based on country-level data on manure application proportions to cropland and pasture, as reported by Ludemann et al. (2023).

---

## Author Comment (AC4)

**Reviewer 2**

**Overall comment:** This study developed a dataset of annual P surplus across Europe at a spatial resolution of 5 arcmin during 1850-2019. The uncertainties of P surplus estimation were considered by using two fertilizer estimates, six animal manure estimates, and two cropland and two pasture P removal estimates. Country-level survey data and multiple spatial maps were used to develop this dataset. The manuscript provided a very detailed description of the methodology and was easily understood. However, I still have several concerns regarding the novelty of the dataset and the reliability of the methods.

**Reply**: We thank the reviewer for their careful review of the manuscript and their useful comments. We reply below to the points raised by the Reviewer.

**1.1** — The published 48 P surplus estimates were not very useful, since most users may only use the ensemble mean. I suggest authors publish data of all P input and output variables instead of only publishing P surplus data.

**Reply**:

Thank you for your insightful comments regarding the publication of the 48 P surplus estimates. We acknowledge the reviewer's concern about the potential over-reliance on ensemble mean estimates by some users and the suggestion to publish additional datasets. In response, we now provide not only the 48 P surplus estimates but also detailed datasets on P inputs and P outputs, which will allow users to carry out more targeted analyses based on specific input and output variables.

The objective of our study is to focus on long-term P surplus dynamics across Europe. Therefore, publishing the P surplus estimates remains crucial for understanding the broader environmental implications. However, we agree that providing access to the underlying input and output data significantly enhances the dataset's utility for the broader research community. **Importantly, the inclusion of all 48 estimates is a key strength of our study, as it accounts for uncertainties inherent in P surplus estimation—an aspect often overlooked in the majority of previous studies.** The uncertainty in P surplus estimates arises from various factors, including different assumptions about fertilizer and manure distribution to cropland and pasture, crop uptake coefficients, and historical data quality. The importance of accounting for uncertainty is increasingly emphasized in nutrient research, as demonstrated by recent work, such as Guejjoud et al. (2023); Ringeval et al. (2024), Sarrazin et al. (2024) and Zhang et al. (2021), which underscores how datasets incorporating uncertainty provide a stronger foundation for robust assessment of underlying nutrient dynamics. Indeed, in a recent comprehensive analysis of nitrogen budget compilation, Zhang et al. (2021) emphasizes that *"To improve nitrogen budget estimates, current uncertainties in concepts, data, and methods need to be addressed . . ."*. While the study focused on the nitrogen budget, similar recommendations can be applied to phosphorus budget estimates, as the main underlying components such as crop removal, mineral fertilizer and manure, are consistent among both nutrients. Our study contributes to this direction and takes a step toward uncertainty assessment in the P surplus budget.

While it may happen that many users may rely on the ensemble mean, we believe that providing uncertainty estimates is a good scientific practice, as it offers a more comprehensive understanding of reconstructed datasets. This is true especially in our case for reconstructing the past nutrient budgets where many methodological aspects and underlying components remain uncertain (e.g., fertilizer or manure applications on different land-use types) (Zhang et al., 2021). To this end, by including the uncertainty estimates alongside the ensemble mean P surplus estimates allows users

to assess the range of variability. Although we cannot control how users choose to utilize the data, we have provided recommendations and made all the datasets available to facilitate informed selections/choices.

To reflected these points, we have further emphasized about the importance of accounting for uncertainty in the introduction of the revised manuscript in line 51-60.

*Nutrient budgets tend to have large uncertainties (Zhang et al., 2021; Ludemann et al., 2023). Uncertainties in P budgets can stem from limited knowledge about the distribution of mineral fertilizers and animal manure on cropland and pasture and about the P removal coefficients, among other factors (Ludemann et al., 2023). As a result, the different studies of Table 2 and Table 3 adopted different schemes to allocate mineral fertilizer and animal manure to cropland and different coefficient values. While some studies explicitly consider uncertainties (e.g., Guejjoud et al. (2023); Antikainen et al. (2008); Lun et al. (2018); Muntwyler et al. (2024); Ringeval et al. (2024); Ludemann et al. (2023); Panagos et al. (2022), listed in Tables 2 and 3), the majority do not. Ignoring this uncertainty could lead to inaccurate assessments of P dynamics and, consequently, flawed policy recommendations (Oenema et al., 2003). Recent studies, such as Guejjoud et al. (2023); Ringeval et al. (2024), Sarrazin et al. (2024) and Zhang et al. (2021) underscore the need for uncertainty-aware nutrient datasets to support quantification of nutrient budgets and robust water quality assessments.*

In addition to publishing the data for P inputs and P outputs, we have also enhanced the manuscript with new visualizations. Figure R1 (which corresponds to Figure 2 in the revised manuscript) now includes gridded maps of P inputs and outputs (alongside P surplus), offering a more detailed spatial overview. We have also updated Figure R2 (which corresponds to Figure 5 in the revised manuscript), which now illustrates the decadal trajectories of agricultural and total P inputs (shown in orange) and P outputs (in blue), while the P surplus is represented by a red line for each decade. Figure R3 and Figure R4 (which corresponds to Supplementary Figure S8 and S9, respectively) further shows decadal trajectories of agricultural and total P inputs for different European countries. This provides a clearer illustration of how inputs and outputs have evolved over time and contributed to changing P surplus levels.

In summary, while the focus of our study is on P surplus, the additional datasets of P inputs and outputs, along with the ensemble of 48 estimates, offer a comprehensive tool for researchers interested in exploring P dynamics in greater depth. We believe that this expanded data access addresses the reviewer's concerns and enhances the dataset's utility, offering researchers a broader set of resources for studying P dynamics in greater depth. Thank you for your constructive feedback on this aspect.

**1.2** — The authors claimed that the novelty of this data is considering P surplus on non-agricultural land. First, it is very weird to identify P inputs (atmospheric deposition and weathering) on non-agricultural land as "surplus". Second, there are no figures showing the results of P surplus on non-agricultural land. The inputs of deposition and weathering puts of P are very low compared to fertilizer and manure inputs, and that is one of the reasons the results in this study are very close to Zou et al. and Ludemann et al. Third, the calculation of P weathering in urban is uncommon. Hartmann's data was developed on soil, and it cannot be directly used on impervious land. Fourth, aside from forests, semi-natural vegetation, and urban areas, what about shrubland and other land use types? Overall, calculating P surplus on non-agricultural land does not make this dataset distinct from other previous datasets.

**Reply**:

Thank you for your valuable feedback and for raising important points regarding the calculation and importance of P surplus budget over non-agricultural areas. We have carefully considered each aspect you highlighted, and below, we provide detailed responses and clarifications for each of your points.

– **Terminology for P surplus on non-agricultural land:** We have chosen to use the term **P surplus** across all land types (both agricultural and non-agricultural) to ensure consistency throughout the manuscript. This unified terminology aligns with our broader aim of analyzing phosphorus dynamics across all diverse land use types. It highlights the importance of understanding both inputs and outputs in various land types, including in areas where agriculture P sources are not dominant. P inputs from other sources may still contribute to long-term environmental challenges. Furthermore, studies on water quality assessment (e.g. Van Meter et al., 2021), requires P surplus data not just from agricultural areas but also from other areas/sources to quantify and analyze total catchment P export. We acknowledge the reviewer's concern regarding the potential confusion in referring to P inputs as P surplus over non-agricultural areas. To address this, we have provided a clarification where this term first appears in the revised manuscript at line 63 in Section 1.

– **Non-agricultural P surplus contributions:** We would like to clarify and explain the relevance of providing P surplus on non-agricultural soils. Our analysis shows that the **northern European countries** (Norway, Sweden, Finland) have a higher contribution of non-agricultural P surplus, mainly over the forests and semi-natural vegetation, compared to other regions. In Norway, non-agricultural areas accounted for approximately 28–40% of the total P surplus in the 1850s, and though this decreased to around 13% in recent decades, it remains notable compared to other regions. Similarly, Finland exhibited a non-agricultural P surplus contribution of 36–42% in the 1850s, declining to 19–31% by 2000–2019. We also have added Figures R5 (Figure 3) and R6-R7 (Supplementary Figures S6 and S7) to illustrate this contribution. Although the non-agricultural P surplus is modest compared to the agricultural P surplus in these countries, it is nevertheless important for understanding the overall P budget, especially in regions with substantial forested and semi-natural landscapes. By accounting for different sources of P inputs and outputs across all considered landscapes, our dataset can support further environmental assessments, such as analyzing the fate of P surplus in catchment-wide water quality studies. Other sources of P inputs to river systems, such as point sources (e.g., wastewater treatment plants), are not considered here. However, our group has worked on reconstructing point source nutrient inputs in a separate study (see (Sarrazin et al., 2024)).

– **Comparison with previous studies**: We also would like to clarify that the datasets of Zou et al. (2022) and Ludemann et al. (2023) mentioned by the Reviewer provide P surplus for **cropland** only (as reported for instance in Table 2, Section 3.2 and in the captions of Figures 6 and 7 of our manuscript). Therefore, the comparison between the datasets of Zou et al. (2022) and Ludemann et al. (2023) and our dataset is limited to croplands and cannot be extended to the relevance of P surplus budgets for other areas, including pasture-dominated lands within agricultural regions and non-agricultural areas.

– **P weathering in urban areas:** We recognize the uncommon application of Hartmann's soil-based weathering data to impervious urban areas. In response to your comment, we have excluded P weathering from urban areas in our

revised estimates. While urban P surplus remains a relatively minor contributor, this adjustment ensures that our calculations align more closely with realistic land use characteristics.

- **Clarification on shrubland and other land types:** In response to your query on shrublands and other land types, we clarify that **semi-natural vegetation in our dataset includes shrubland**. Specifically, for non-agricultural areas in our dataset, we used the classification of land cover categories from global land cover (GLC) (Bartholomé and Belward, 2005) that is available at a spatial resolution of 300 m and includes 23 land cover classes. From these classes, we selected all relevant land types representing non-agricultural land, which we divided into four categories, namely semi-natural-vegetation (tree, shrub-land, herbaceous cover), forest (broad-leaved, evergreen and deciduous forest), non-vegetation (bare areas, water bodies) and urban areas. This ensures that all non-agricultural areas contributing to P surplus are represented in our analysis.

- **Distinction from previous datasets:** Our dataset offers several unique contributions that collectively provide a more comprehensive view of the phosphorus (P) surplus budget across Europe. While including non-agricultural land types is crucial for gaining a thorough understanding of the total P surplus budget within a given landscape unit (e.g., catchments or administrative units), our approach is further enhanced by additional aspects such as extended spatial and temporal coverage and the incorporation of uncertainty assessments in the P surplus budgets. Below, we outline some key features that distinguish our dataset from existing ones or complement them beyond considering both agricultural and non-agricultural areas:

  - As highlighted in our first reply to the reviewer, our dataset includes 48 distinct P surplus estimates, providing a comprehensive representation of uncertainty—a critical aspect partially considered in some past studies reported in Tables 2 and 3. This uncertainty-aware approach is essential for reliably assessing the evolutions of P dynamics and supporting effective policy recommendations, as highlighted in recent literature on nutrient budgeting (Guejjoud et al., 2023; Sarrazin et al., 2024; Zhang et al., 2021).

  - With a spatio-temporal coverage at gridded scale and extending from 1850 to 2019, our dataset provides a long-term, detailed view of P surplus budget across diverse European landscapes. Further, in comparison to existing global databases, such as those by Zou et al. (2022) and Ludemann et al. (2023), which are limited to cropland P budgets, our dataset extends further by covering both cropland and pastureland P budgets. This provides a more comprehensive view of P dynamics across agricultural areas, while covering P budgets over the non-agricultural areas as well. The spatial and temporal depth enhance the utility of our dataset for several studies, ranging from historical agriculture and environmental change to biogeochemical cycling, nutrient management, legacy store characterizations, and water quality assessments, where understanding shifts over time is critical.

In response to your feedback, we have incorporated new gridded and country-level visualizations to better emphasis the contribution of non-agricultural P surplus to total P surplus.

Figure R5 (Figure 3 in the revised manuscript) displays this contribution at the gridded level, while Figures R6 and R7 (Supplementary Figures S6 and S7) depict contributions by countries and over time. To further elaborate on this point, we have added the following text in section 3.1 (at line 651) of the revised manuscript:

*The importance of non-agricultural P surplus is highlighted in Fig R5, which illustrates its contribution to total P surplus. Northern European countries, such as Norway, Sweden, and Finland, show a higher contribution of non-agricultural P surplus, with 30–60% contribution across 70% of grid cells during the entire period (1850–2019). Central and Western Europe exhibit more variable contributions over time. For example, in 1900 and 1930, the non-agricultural contribution in these regions ranged between 10–30%, but it decreased to around 10% by 1990, with further declines in recent years. Southern Europe, meanwhile, displayed a moderate and stable contribution of up to 20% from 1960 to 2019. Supplementary Figures S6 and S7 provide additional insights, showing the contribution of non-agricultural P surplus*

*at both the country level and on a decadal scale. Northern and Eastern European countries demonstrate increasing contributions over time, such as Estonia (from 15% in 1850–60 to 30% in 2010–19) and Sweden (from 35% to 40% over the same period). Meanwhile, countries like Belgium, the Netherlands, and Switzerland show a consistent decrease in contribution throughout the period, such as Switzerland dropping from 40% in 1850–60 to 5% in 2010–19.Understanding these dynamics is critical for devising holistic nutrient management strategies that account for the role of non-agricultural P sources. By incorporating non-agricultural P surplus data, our dataset enables a more comprehensive understanding of P fluxes across Europe.*

To clarify the shrubland and other land use category in our dataset and to further explain our approach to constructing non-agricultural land, we have revised the text in section 2.2.2 at line 206 of the revised manuscript, which now reads as follows:

*The non-agricultural area in a grid cell was calculated as the remaining area after allocating cropland and pasture areas. We used the classification of land cover categories from global land cover (GLC) (Bartholomé and Belward, 2005) that is available at a spatial resolution of 300 m. GLC includes 23 land cover classes that we grouped into 5 categories namely, cropland, semi-natural-vegetation (i.e. vegetation not planted by humans but influenced by human actions (Di Gregorio, 2005) including tree, shrub-land, herbaceous cover, Lichen and mosses), forest (broad-leaved, evergreen and deciduous forest), non-vegetation (bare areas, water bodies) and urban area. The proportions of these categories were then applied to the non-agricultural area to estimate their annual development from 1850 to 2019.*

Moreover, to emphasize the unique aspects of our dataset, we have added a detailed section 4 in the revised manuscript, in which we highlight the value of our dataset at line 875-923.

**1.3** — The fertilizer data before 1960 was calculated by using the temporal changes from Holland et al. However, Holland only provides N fertilizer data. The N fertilizer is produced from the Haber-Bosch process while P fertilizer is produced from mineral rock. These two different technology may not lead to a constant N:P ratio of fertilizer before 1960. Therefore, it is not a solid method to directly use temporal changes of N fertilizer on P fertilizer.

**Reply**:

Thank you for your valuable feedback. We fully acknowledge the distinction between the production processes of nitrogen (N) fertilizers and phosphorus (P) fertilizers, as N fertilizers are derived from the Haber-Bosch process while P fertilizers are produced from phosphate rock. Given these technological differences, we recognize that it is not appropriate to assume a constant N:P ratio for fertilizers before 1960.

In response to your comment, we have revised our methodology. Instead of relying on N fertilizer trends from Holland et al. (2005) as a proxy for changes in P fertilizer, we have now incorporated a **global dataset (Cordell et al., 2009) that traces the historical sources of phosphorus fertilizers from phosphate rock (1800–2000)**. This dataset provides a more reliable temporal trend for P fertilizer use. For the period before 1961, we applied the temporal trends from this phosphate rock dataset uniformly across all countries, and adjusted to the country specific 1961 fertilizer estimate. This improved approach for estimating P fertilizer contribution is detailed in the revised manuscript at line 265 under section 2.3.2 of the revised manuscript which reads as follows:

*Regarding the time period of 1850−1960, when country-level P fertilizer data from FAOSTAT were unavailable, we utilized the temporal dynamics from Cordell et al. (2009) that provides global estimates of phosphate rock production during 1800−2000. These estimated P inputs were normalized to align with FAOSTAT data starting in 1961, using 1961 as a reference year for consistency. The global temporal dynamics was then applied across all countries in our study domain for 1850–1960, proportionally scaling the values based on each country's 1961 estimate. This approach allowed*

*us to generate a temporally coherent dataset, using global phosphate rock production as a proxy for P inputs from fertilizer during the period of limited data availability. The completed annual country-level fertilizer data are referred to as $Pfer_{soil}(u, y_{1850-2019})$ (kg yr$^{-1}$).*

We thank the reviewer for bringing this issue to our attention. We believe the new approach and the adjustments we did have significantly enhances the reliability our P surplus dataset. This revision addresses the concerns you raised, and we are confident it strengthens the methodological integrity of our dataset.

**1.4** — I also doubt the assumption of equal distribution rates of treated manure on cropland and pasture. Are there any survey data or studies that can support it?

**Reply**:

Thank you for your insightful comment. We share the reviewer's critical perspective on this aspect, particularly regarding the broader question of the best approach to estimate distribution rates of treated manure across croplands and pastures. Indeed, this remains an important area and it has been broadly consider as uncertain (Zhang et al., 2021). Different approaches have been reported in literature for estimating treated manure application rates in light of lack of detailed survey data. Accordingly, we applied three different methods – one of them is based on the equal distribution rates of treated manure between cropland and pasture following (Xu et al., 2019) based on generalized assumptions to account for the absence of detailed, consistent data. While such assumptions may not fully reflect actual land management practices, the inclusion of the equal distribution assumption in our study is intended to offer a complementary perspective within our ensemble of methodologies rather than asserting it as the most reliable approach. This assumption has the advantage of dynamically adjusting manure distribution based on changing cropland and pasture areas over time and by country, providing insights into how shifts in land use patterns might influence P dynamics in the long term. We recognize that this approach may carry greater uncertainties than other, more specific country-level methods, and we do not assume that it is preferable to other distribution methodologies.

To address these uncertainties, we further employed an ensemble of approaches that incorporate more granular, country-specific data where available. For instance, the national manure distribution ratios from Ludemann et al. (2023) provide detailed, fixed proportions for each country, while the time-varying national N-based distribution from Einarsson et al. (2021) (used as a proxy for P distribution) incorporates animal type-specific data, refining manure allocation further. These proportions are shown in Figure R8. By integrating these varied methods, our uncertainty-aware approach provides a range of possible outcomes, ensuring that no single assumption, including the equal distribution approach, dominates or unduly influences uncertainty estimates. Instead, our aim is to provide multiple scenarios to capture the inherent uncertainties in manure distribution to cropland and pasture across Europe. We have revised the manuscript at the end of section 2.3.8 to clarify this approach, emphasizing that we do not assert one method as definitively more accurate than another but rather use a suite of methodologies to account for uncertainty comprehensively. The revised text in line 484 reads as follows:

*Overall, by integrating two distinct data sources ((FAOSTAT, 2022) and Einarsson et al. (2021)) alongside three manure distribution methods between croplands and pastures, we developed six separate gridded manure estimates for our database. These estimates reflect the uncertainties in our reconstruction, which arise from the selection of different underlying datasets and distribution methods. Each method captures distinct aspects of manure allocation: the equal distribution assumption adjusts dynamically with cropland and pasture area changes over time, while the country-specific ratios from Ludemann et al. (2023) apply fixed national-level allocations. The third method, based on Einarsson et al. (2021), utilizes time-varying nitrogen-based proportions as a proxy for P manure distribution. Supplementary Figure S1 illustrates these proportion of animal manure allocated to cropland and pasture under each method, highlighting the differences and capturing the uncertainties embedded in our approach. By combining these varied assumptions, our*

*estimates provide a comprehensive view of manure distribution across cropland and pasture, supporting a nuanced analysis of P surplus uncertainty.*

235     Further, in support of our response, we have included the figure R8 (which corresponds to Supplementary Figure S1) illustrating the proportion of animal manure allocated to cropland and pasture under each of the three methodologies applied in our study. This visual comparison highlights the distinct allocation patterns produced by each method.

    **1.5** — The calculation of P removal from pasture is very simple. Temporal change of PUE can impact results too. Other impact factors, such as climate, were also not considered.

240     **Reply**:

    Thank you for your valuable comment regarding the simplicity of our approach to calculating phosphorus (P) removal from pasture. We acknowledge the importance of considering temporal changes in phosphorus use efficiency (PUE) and the potential influence of additional factors like climate on P removal estimates.

    Given the inherent complexity of fully estimating P removal from pasture that includes data on changing livestock 245 intake activities, management activities, and other forage activities, in this study we adopted an approach from prior research (Bouwman et al., 2005, 2009; Kaltenegger et al., 2021), utilizing fixed P removal coefficients to approximate PUE for pasture. To capture uncertainty, we used two assumptions: (1) a general P removal coefficient of 0.6, derived from Bouwman et al. (2005), and (2) region-specific PUE values as proxies, based on nitrogen use efficiency (NUE) data from Kaltenegger et al. (2021), with coefficients of 0.4 for Eastern Europe and 0.5 for Western Europe. These approaches 250 allowed us to develop two distinct datasets for P removal from pasture, each reflecting PUE estimates though constant in time. This approach, while simplified, enables an assessment of P removal from pasture in the absence of detailed historical data.

    However, we agree that PUE is not static and can vary over time due to changes in agricultural practices, pasture management, and environmental factors, all of which can influence the accuracy of P removal estimates. Likewise, climate 255 factors such as precipitation, temperature, and soil moisture directly affect pasture growth and, consequently, P uptake and cycling. To address the reviewer's comment, we have included a more detailed discussion of these points for further improvements of our datasets in the manuscript. Specifically, we have emphasized that the temporal variability in PUE and climate impacts, such as droughts or temperature extremes, could lead to under- or overestimations of P removal in our dataset. This point under the section 4 at line 870 of the revised manuscript now reads as follows:

260     *Further limitations include the simplification of parameters that likely vary across space and time. While the coefficients used were based on prior research (Bouwman et al., 2005; Kaltenegger et al., 2021), PUE can be highly variable across regions and management practices (Lun et al., 2018; Chowdhury and Zhang, 2021). Our use of fixed coefficients may not fully capture this variability, especially in countries with varying level of grazing intensities or grassland management practices. Furthermore, climate-related factors such as changes in precipitation, temperature, and soil moisture directly 265 affect pasture productivity and thus P uptake and cycling (Martins-Noguerol et al., 2023), which our static approach does not fully encompass. This simplification was necessary due to the lack of detailed historical agricultural records but introduces some degree of uncertainty in our P removal from pasture areas.*

    Moreover, while dynamic PUE datasets and models, such as those that factor in climate variability (Ijaz et al., 2017) or region-specific grazing practices (Anderson et al., 2020), provide more accurate P removal estimates, they are limited in 270 temporal and spatial scope and do not cover the long historical period of this study. Future work can therefore prioritize integrating such dynamic models where possible. Incorporating these time-varying factors would undoubtedly improve the robustness of our estimates, and we appreciate your suggestion as a pathway for future refinement. Regarding this, we have added the following text in section 5 at line 965 of the revised manuscript:

*Another future enhancement would involve refining parameters to account for temporal and spatial variability. For exam-*
275 *ple, crop-specific P uptake rates vary with soil quality, crop variety, and management practices, while pasture P removal is influenced by phosphorus use efficiency (PUE) and meteorological and hydrological variables such as precipitation, temperature, and soil moisture. Future work should incorporate dynamic PUE estimates to better capture time-varying removal rates driven by regional grazing practices, crop types, and changing weather patterns.*

**1.6** — Since there are so many weaknesses in the method, I strongly suggest adding one section of the limitation of this data.

280 **Reply**: Thank you for your feedback. In response to your suggestion, we have added two dedicated sections — *4. Potential use and limitations of the dataset* and *5. Directions for future improvement of the dataset* — in the revised manuscript. These sections thoroughly outline the key methodological constraints of our dataset. By explicitly addressing the limitations and identifying directions for future improvement, we aim to enhance transparency about potential uncertainties in our findings and provide a balanced perspective on the dataset's strengths and limitations. We believe these additions will
285 offer readers valuable insights into both the current applicability of the dataset and directions for its continued refinement.

**References**

Anderson, K. R., Moore Jr, P. A., Pilon, C., Martin, J. W., Pote, D. H., Owens, P. R., Ashworth, A. J., Miller, D. M., and DeLaune, P. B.: Long-term effects of grazing management and buffer strips on phosphorus runoff from pastures fertilized with poultry litter, Tech. rep., Wiley Online Library, 2020.

[revised manuscript text omitted]

**Figure R6.** Heat map showing the temporal evolution of the contribution (%) of non-agricultural P surplus to the total P surplus across different European countries from 1850-2019. The figure highlights the annual variation in the proportion of non-agricultural P surplus to the total P surplus (averaged from 48 P surplus estimates) across different European countries, illustrating the evolving role of non-agricultural sources in European P dynamics over time. In recent years, countries such as Hungary, Bulgaria, Estonia and Croatia have recorded peak values, which is mainly due to a lower agricultural P surplus, as a result of which the relative share of the non-agricultural P surplus in the total P surplus has amplified.

[Figure]

**Figure R7.** Decadal contribution of non-agricultural P surplus to the total P surplus across different European countries

[Figure]

**Figure R8.** Fractions of manure distribution to cropland based on different methods utilized in this study. Method 1 represents the fraction of distribution of animal manure to cropland based on the equal distribution rates for cropland and pasture within each grid cell. Method 2 shows the fraction using the time-varying national proportions of nitrogen (N) manure applied to both cropland and pasture, as provided by Einarsson et al. (2021). Method 3 shows the manure distribution based on country-level data on manure application proportions to cropland and pasture, as reported by Ludemann et al. (2023).

---

## Author Comment (AC5)

**Reviewer 3**

**Overall comment:** Batool et al. have conducted a comprehensive analysis of the phosphorus budget in Europe, compiling a valuable dataset by integrating various assumptions and parameters from a range of publications. The inclusion of multiple sources of phosphorus inputs and outputs from pasture and non-agriculture provides a broader perspective on the phosphorus budget within terrestrial ecosystems. However, several key assumptions—particularly those related to time series reconstruction and spatial allocation—raise concerns, as they diminish the dataset's robustness and weaken the overall conclusions. Below are my specific comments for further improvement:

**Reply**: We thank the Reviewer for their useful comments to the manuscript. We reply below to the points raised by the Reviewer. We have carefully revised the manuscript, addressing the reviewers' concerns and suggestions, and we hope it now meets their expectations.

**1.1 —**

– Although the datasets have been allocated to gridded maps, they are primarily based on national-level aggregates. Caution should be exercised when claiming that this dataset offers high spatial resolution.

**Reply**:

Thank you for your insightful comment regarding the spatial resolution of our dataset. We acknowledge that while our dataset has been downscaled to a high-resolution grid, much of the underlying data, especially for inputs like fertilizer use and manure application, are indeed based on national-level aggregates. We would also like to emphasis here that we also used several other sources of spatially refined datas in our dowscaling methodology, such as different crop types, HYDE database on agriculture land development, the Global Land Cover database, among others. While we acknowledge the reviewer's point, we believe the reviewer would agree that many existing nutrient databases (Zhang et al. (2017); Lu and Tian (2017); Xu et al. (2019)) are based on national estimates in one way or another, given the long-term consistency and availability of data at this scale.

In response to this point, we have clarified in the revised manuscript that although the dataset offers gridded outputs, users should exercise caution when interpreting these results as high-resolution spatially explicit data. The spatial patterns are influenced by the downscaling of national-level data, and local variations might not be fully captured. We have adjusted the language in the section 5 of the revised manuscript to reflect this more clearly, emphasizing that the dataset is suitable for basin-level, aggregated regional, national and continental-scale assessments but may not be as reliable for detailed local-scale analysis without further refinement of underlying data. Additionally, we have revised the discussion to address the limitations in the spatial allocation process explicitly, particularly with respect to the uncertainties that arise from using national aggregates for gridded outputs. The added text at line 861 under section 4 reads as follows:

*Second, while the dataset has been downscaled to a high-resolution grid, several underlying inputs, such as fertilizer use and manure application, originate from country-level aggregates. This approach limits the ability of gridded outputs to capture local variability in P surplus. Therefore, while the dataset is valuable at coarser basin, regional, national and continental scales, caution should be exercised for high-resolution applications. We recommend using the dataset at aggregate levels, such as countries, European socio-economic regions (e.g., NUTS levels), or river basin scales (see Section "Spatio-temporal variation in P surplus, P inputs and P outputs" and Figure 4) to support land and water management activities.*

We would also like to emphasis here that the grid resolution of our dataset provides users with the flexibility to aggregate data at different spatial scales depending on their research needs. This allows the dataset to be used at different spatial scales, with the most robust applications being at a larger spatial scale. This point emphasizes the potential use of our dataset and has been added under section 4 in line 894 of the revised manuscript as follows:

*Furthermore, our gridded dataset supports a greater degree of flexibility for aggregating data at various spatial scales, such as national, regional, and river basin levels, which is helpful in analyzing trans-boundary nutrient flows across Europe. This flexibility allows the dataset to address the needs of cross-regional and trans boundary applications, including in major European river basins like the Elbe, Danube and Rhine, thereby facilitating joint nutrient management in shared water bodies (Müller-Karulis et al., 2024). By identifying critical regions of high P surplus or its components (P fertilizer/manure), users can pinpoint locations where nutrient management improvements could have the greatest environmental and economic impact (Malagó and Bouraoui, 2021).*

By revising the text in line with the comment of the reviewer, we hope that this will provide readers with a better understanding of the dataset's strengths and potential limitations for high-resolution applications and its potential use. We appreciate the feedback, as it will help to ensure that users of the dataset understand the appropriate context for its use.

– Pasture definition: Clarification is needed on whether "pasture" refers to both grazed and natural pasture, and what is meant by "semi-natural vegetation." The distinction between these categories could significantly affect the phosphorus budget.

**Reply**:

Thank you for your comment regarding the definitions of "pasture" and "semi-natural vegetation".For clarification, we have based these definitions on established sources, as described below:

– **Pasture Definition:** The definition of pasture in our study aligns with the FAO Land Use classification, which defines pasture area under permanent meadow and pasture as *"land used permanently (five years or more) to grow herbaceous forage crops, either cultivated or naturally occurring (e.g., wild prairie or grazing land)"* (FAOSTAT, 2021). This is the definition used in our dataset of pasture area, ensuring consistency with widely accepted land-use classifications and comprehensively accounting for land designated for long-term herbaceous forage, whether actively grazed or in a natural state.

– **Semi-Natural Vegetation Definition:** The term "semi-natural vegetation" is based on the definition from Land Cover Classification System (LCCS), which describes it as *"vegetation not planted by humans but influenced by human actions, such as grazing or selective logging, which alters the floristic composition"* (Di Gregorio, 2005). To spatially delineate semi-natural vegetation, we use the Global Land Cover (GLC) dataset (Bartholomé and Belward, 2005), which offers a high-resolution (300 m) classification of different land-cover classes. Following the GLC database, in our analysis, semi-natural vegetation includes:

  • Naturally occurring tree, shrub, and herbaceous cover
  • Shrubland (including both evergreen and deciduous types)
  • Lichens and mosses
  • Sparse vegetation (such as sparse tree, shrub, and herbaceous cover)
  • Vegetation in flooded areas (including shrub or herbaceous cover in freshwater, saline, or brackish water zones)

By following the FAO definition and utilizing the GLC dataset, we capture the range of vegetation types that fall under semi-natural land. To make this aspect clear, we have revised the definition of pasture in section 2.2.1 (line 156) which now reads as follows:

*Pasture area is the land under permanent meadow and pasture and is defined as land used permanently (five years or more) to grow herbaceous forage crops, either cultivated or naturally occurring (e.g., wild prairie or grazing land) (FAOSTAT, 2021).*

Similarly, we have added definition of semi-natural vegetation in section 2.2.2 (line 206) which now read as follows:

*We used the classification of land cover categories from global land cover (GLC) (Bartholomé and Belward, 2005) that is available at a spatial resolution of 300 m. GLC includes 23 land cover classes that we grouped into 5 categories namely, cropland, semi-natural-vegetation (i.e. vegetation not planted by humans but influenced by human actions (Di Gregorio, 2005) including tree, shrub-land, herbaceous cover, Lichen and mosses), forest (broad-leaved, evergreen and deciduous forest), non-vegetation (bare areas, water bodies) and urban area. The proportions of these categories were then applied to the non-agricultural area to estimate their annual development from 1850 to 2019.*

– Section 2.2.1: The authors mention using HYDE to derive temporal variation in cropland and pasture. It remains unclear whether this variation was applied on a grid-by-grid basis. If it was applied to grids, how did the authors "maintain the spatial distribution from Ramankutty et al. (2008) while accounting for annual temporal changes from HYDE"? Alternatively, if the country-level annual variation was used by aggregating grids within each country, it seems redundant, as FAOSTAT already rescaled the grids. Clarification is needed.

**Reply**:

Thank you for this insightful comment regarding the use of HYDE and Ramankutty et al. datasets for cropland and pasture reconstruction. For clarification, we first utilized the spatial distribution of cropland and pasture areas from Ramankutty et al. (2008), which are available at a spatial resolution of 5 arcmin for the year around 2000. This static spatial dataset serves as the base spatial distribution of agricultural areas. To derive the temporal variability in the gridded values of cropland and pasture area over the period 1850–2019, we employed the HYDE dataset, which provides decadal (and post-2000, annual) grid-level cropland and pasture areas. We generated the temporal variability in cropland and pasture with respect to the year 2000 area from the HYDE dataset for each year during 1850 − 2019 and each grid cell. We then applied these gridded normalised (temporal) values of cropland and pasture area to the respective land use area of Ramankutty et al. (2008) in 2000 at grid-level. Then, we harmonized the resulting grid-level cropland and pasture areas with FAOSTAT's country-level data for 1961–2019 to ensure consistency. This harmonization was achieved by calculating country-level adjustment ratios between FAOSTAT's reported areas and our reconstructed areas. Then, we applied these calculated ratio to our gridded estimates of cropland and pasture areas to ensure that the FAOSTAT country totals are maintained for the period 1961 − 2019.

We understand that this aspect may not have been clear in the initial version of the manuscript, as we referred to our previous study by Batool et al. (2022), where the land-cover-related reconstruction was presented in detail. To ensure the current paper stands alone in terms of necessary information, we have provided additional clarification in the section 2.2.1 of the revised manuscript and included equations illustrating this process. This will help readers understand how we integrated different databases to create the temporal and spatial dynamics of cropland and pasture areas. This revision now reads as follow:

*Cropland is defined as land used for the cultivation of crops, including arable crops and land under permanent crops (Ramankutty et al., 2008; FAOSTAT, 2021). Pasture area is the land under permanent meadow and pasture and is defined as land used permanently (five years or more) to grow herbaceous forage crops, either cultivated or naturally occurring (e.g., wild prairie or grazing land) (FAOSTAT, 2021). To represent the spatial distribution of cropland and pasture areas, we utilized the dataset from Ramankutty et al. (2008), which provides gridded estimates at a 5-arcminute resolution for the year 2000. These gridded values serve as the baseline for cropland and pasture area in our analysis. To account for temporal changes in cropland and pasture areas, we used data from the History Database of the Global Environment (HYDE version 3.2) (Goldewijk et al., 2017). HYDE provides global decadal estimates of cropland and pasture areas from 1700 to 2000, as well as annual values from 2000 to 2017. We*

*generated annual time series of cropland and pasture areas for the period 1850–2019 using linear interpolation for the decadal estimates. For the years 2018 and 2019, we used the same values as 2017 due to a lack of available data.*

*To combine the data from Ramankutty et al. (2008) and from HYDE, we first calculated temporal ratios for the HYDE data for each grid cell using the year 2000 as the reference year. These ratios represent the relative change in cropland $R_{HYDE\text{-}cr}$ (-) and pasture area $R_{HYDE\text{-}past}$ (-) over time, normalized to the year 2000:*

$$R_{\text{HYDE-cr}}(i, y_{1850,-,2019}) = \frac{A_{\text{HYDE-cr}}(i, y_{1850,-,2019})}{A_{\text{HYDE-cr}}(i, y_{2000})} \tag{R1}$$

$$R_{\text{HYDE-past}}(i, y_{1850,-,2019}) = \frac{A_{\text{HYDE-past}}(i, y_{1850,-,2019})}{A_{\text{HYDE-past}}(i, y_{2000})} \tag{R2}$$

*Where $A_{HYDE\text{-}cr}(ha))$ and $A_{HYDE\text{-}past}(ha))$ are the gridded cropland and pasture areas, respectively.*

*Next, we applied these normalized ratios to the baseline gridded values from Ramankutty et al. (2008) to derive annual cropland and pasture areas for each grid cell, as follows:*

$$A_{\text{cr}}(i, y_{1850,-,2019}) = A_{\text{Ramankutty-cr}}(i, y_{2000}) \times R_{\text{HYDE-cr}}(i, y_{1850,-,2019}) \tag{R3}$$

$$A_{\text{past}}(i, y_{1850,-,2019}) = A_{\text{Ramankutty-past}}(i, y_{2000}) \times R_{\text{HYDE-past}}(i, y_{1850,-,2019}) \tag{R4}$$

*Where $A_{Ramankutty\text{-}cr}(ha)$ and $A_{Ramankutty\text{-}past}(ha)$ are the gridded cropland and pasture areas from Ramankutty et al. (2008) for the year 2000, and $A_{cr}(ha)$ and $A_{past}(ha)$ are the estimated cropland and pasture areas.*

*We harmonized our reconstructed cropland and pasture areas with FAOSTAT data available at country-level, which provides consistent information from 1961–2019. To do so, we calculated country-level ratios for cropland and pasture areas by comparing FAOSTAT data with the sum of our gridded estimates for each country. The ratios were calculated as follows:*

$$R_{A_{\text{cr}}}(u, y_{1961,-,2019}) = \frac{A_{\text{FAO-cr}}(u, y_{1961,-,2019})}{\sum_{i=1}^{n_u} A_{\text{cr}}(i, y_{1961,-,2019})} \tag{R5}$$

$$R_{A_{\text{past}}}(u, y_{1961,-,2019}) = \frac{A_{\text{FAO-past}}(u, y_{1961,-,2019})}{\sum_{i=1}^{n_u} A_{\text{past}}(i, y_{1961,-,2019})} \tag{R6}$$

*Whereas $R_{A_{cr}}(-)$ is the country-level ratio of cropland area, $A_{FAO\text{-}cr}(ha)$ represents the country-level cropland area from FAOSTAT, $n_u$ is the number of grid cells in country $u$, and $\sum_{i=1}^{n_u} A_{cr}(ha)$ is the sum of the gridded cropland areas in country $u$ in year $y$. Similarly, $R_{A_{past}}(-)$ is the ratio of pasture area, $A_{FAO\text{-}past}(ha)$ is the country-level pasture area from FAOSTAT, and $\sum_{i=1}^{n_u} A_{past}(ha)$ is the sum of the gridded pasture areas.*

*We applied these ratios to adjust our gridded estimates to match FAOSTAT's country-level data (all variables, except for ratios, are in $ha$):*

$$A_{\text{cr}}^{\text{cor}}(i, y_{1961,-,2019}) = R_{A_{\text{cr}}}(u, y_{1961,-,2019}) \times A_{\text{cr}}(i, y_{1961,-,2019}) \tag{R7}$$

$$A_{\text{past}}^{\text{cor}}(i, y_{1961,-,2019}) = R_{A_{\text{past}}}(u, y_{1961,-,2019}) \times A_{\text{past}}(i, y_{1961,-,2019}) \tag{R8}$$

*Whereas $A_{cr}^{cor}$ represents the corrected gridded cropland, $R_{A_{cr}}$ is the country-level ratio of cropland area as given in equation R5, and $A_{cr}$ is the original gridded cropland area as derived in equation R3. Similarly, $A_{past}^{cor}$ represents the corrected gridded pasture area, $R_{A_{past}}$ is the country-level ratio of pasture area as shown in equation R6, and $A_{past}$ is the original gridded pasture area as derived in equation R4.*

$A_{cr}^{cor}$ and $A_{past}^{cor}$ represent the corrected gridded cropland and pasture areas, respectively. These corrections are based on country-level ratios—$R_{A_{cr}}$ for cropland and $R_{A_{past}}$ for pasture—calculated using equations R5 and R6. The original gridded areas, $A_{cr}$ and $A_{past}$, are adjusted according to these ratios using equations R3 and R4.

For years prior to 1961, we used the same ratios as of 1961 to maintain consistency. In cases where FAOSTAT data were not available before 1992 (e.g., for Estonia, Croatia, Lithuania, Latvia, and Slovenia), we used the ratios from the year 1992 for the period 1850–1991. For countries like Luxembourg and Belgium, and Slovakia and Czech Republic, which were reported as single entities in historical records, we used combined ratios for the respective periods. Finally, the total agricultural area $A_{agri}^{cor}$ ($ha$) for each grid cell was calculated by summing the corrected cropland $A_{cr}^{cor}$ and pasture areas $A_{past}^{cor}$ (all variables are in $ha$):

$$A_{agri}^{cor}(i, y_{1850,-,2019}) = A_{cr}^{cor}(i, y_{1850,-,2019}) + A_{past}^{cor}(i, y_{1850,-,2019}) \tag{R9}$$

We ensured physical consistency by checking that the agricultural area in each grid cell did not exceed the total physical area of the grid cell. In rare cases where this condition was violated due to inconsistencies in data sources (e.g., differences between FAOSTAT (FAOSTAT, 2021), HYDE (Goldewijk et al., 2017), and Ramankutty et al. (2008)), we redistributed the excess agricultural area to neighboring grid cells.

We hope that the revised manuscript presents the clarified our reconstruction of gridded cropland dynamics.

– Section 2.2.3: The authors refer to 17 non-fodder and 6 fodder crops. Do these cover all cropland? The area of these crops might be overestimated after harmonization with FAOSTAT if they do not cover all cropland. Also, how was the temporal dynamic of cropland area applied to Monfreda et al.—on a grid-by-grid or country level? Furthermore, how was the crop-specific area time series harmonized with FAOSTAT data on the map as the total cropland has been harmonized in section 2.21? Does each grid represent only one crop, or multiple crops?

**Reply**:

Thank you for your detailed comments. As we stated above, we understand that this aspect may not have been clear in the initial version of the manuscript, as we referred therein to our previous study by Batool et al. (2022), where the land-cover-related reconstruction was presented in detail. To ensure the current paper stands alone in terms of necessary information, we have made additional changes to the revised manuscript.

In the following, we have first clarified the points concerning the coverage of cropland by the 17 non-forage and 6 forage crops, the harmonization with FAOSTAT data and the handling of grid-specific cropland areas. We then detailed the changes made for the revised manuscript.

– **Coverage of all cropland by selected crops:** We obtained gridded crop specific harvested areas from Monfreda et al. (2008) which is available for 175 different crop types. Among these 175 crops, we selected 17 major crops for which fertilizer application rates are provided Heffer et al. (2017) and that have large production across Europe. These selected crops cover most of the cropland across Europe. However, minor crops not included within the 17 selected crops were excluded due to the lack of long-term, consistent data. This exclusion may result in some discrepancies in the overall coverage of certain crop types – a limitation we acknowledge in the manuscript.

– **Temporal dynamics applied at the grid level:** We used the cropland dataset from Ramankutty et al. (2008) available at a spatial resolution of 5 arcmin for the year around 2000 to get the spatial variability at gridded level. Then, we derived the temporal variability of cropland area at grid-level from the HYDE dataset for each year during $1850-2019$, by referencing it to the year 2000, as in equation 14 of the revised manuscript. We then applied these normalized (temporal) gridded values of cropland area to the gridded cropland area of Ramankutty et al. (2008) in 2000, as in equation 16 of the revised manuscript. This approach ensured that each grid cell's cropland area of Ramankutty et al. (2008) are adjusted consistently over time with HYDE's cropland grid-level temporal dynamics.

- **Harmonization of crop-specific area time series with FAOSTAT:** The crop-specific harvested areas were harmonized with FAOSTAT country-level data using a ratio-based approach. The ratio was derived between FAOSTAT country-level data and the sum of our gridded estimates of crop-specific harvested area. This ratio was then applied to adjust the gridded estimates of crop-specific harvested areas for each grid cell, ensuring harmonization with FAOSTAT data. For years before 1961, we applied the 1961 ratios to maintain consistency in the estimates.

- **Crop representation in each grid:** Each grid cell represents multiple crops, not just a single crop. We used the crop distribution from Monfreda et al. (2008) to allocate the relative proportions of different crops within each grid cell. The proportional distribution of crops was adjusted over time using the temporal dynamics of cropland area, ensuring that the evolution of crop harvested areas in each grid cell is consistent with changes in total cropland.

To clarify these points, we have revised section 2.2.1 for the points related to cropland areas (the revision is mentioned in the reviewer's response above) and section 2.2.3 on crop-specific harvested area. The revised section 2.2.1 reads as follows:

*Reconstruction of crop-specific harvested area: We acquired gridded crop-specific harvested areas from Monfreda et al. (2008) for 175 different crops representing the year 2000. Among these, we selected 17 major non-fodder crops for which mineral fertilizer application rates are available (Heffer et al., 2017) and which are widely grown across Europe, as well as six fodder crop categories. Below we provide a more detailed overview on the selected crops (see also Table ??). These selected crops cover most of the cropland across Europe. The harmonization process ensures that the total cropland area aligns with FAOSTAT estimates.*

*To generate annual time series of crop-specific harvested areas, we applied the temporal dynamics of cropland areas, adjusting the spatial distribution of crops based on the Monfreda et al. (2008) dataset, while referencing FAOSTAT's country-level data to ensure consistency over time. The crop-specific harvested areas $A_{crops}$ $(ha)$ were harmonized with FAOSTAT data $A_{crops_{FAO}}$ $(ha)$ using a ratio-based approach. The ratio $R_A$ (-) between FAOSTAT country-level data and the sum of gridded estimates was calculated as follows:*

$$R_A(u,c,y) = \frac{A_{crops_{FAO}}(u,c,y)}{\sum_{i=1}^{n_u} A_{\text{crops}}(i,c,y)} \tag{R10}$$

*This ratio was then applied to adjust the gridded estimates of crop-specific harvested areas for each grid cell, ensuring harmonization with FAOSTAT data:*

$$A_{\text{crops}}^{\text{cor}}(i,c,y) = A_{\text{crops}}(i,c,y) \times R_A(u,c,y) \tag{R11}$$

*Where $A_{crops}^{cor}$ is the corrected crop-specific harvested areas for grid cell $i$, crop $c$, and year $y$.*

*For years prior to 1961, we applied the ratio from 1961 to maintain consistency across all years:*

$$A_{\text{crops}}^{\text{cor}} = A_{\text{crops}}(i,c,y_{1850,-,1960}) \times R_A(u,c,y_{1961}) \tag{R12}$$

*This method ensured that the crop-specific harvested areas were harmonized with FAOSTAT country-level data, with each grid representing multiple crops.*

*For fodder crops, we utilized country-level data from Einarsson et al. (2021), available from 1961 to 2019 for 26 European countries. This dataset includes six fodder crop categories, namely: temporary grassland, lucerne, other leguminous plants, green maize, root crops (forage beet, turnip, etc.), and other fodder plants harvested from cropland. For the period 1850–1960, we applied the temporal dynamics of reconstructed cropland areas to estimate fodder crop areas. For countries with missing data, we filled gaps by extrapolating ratios from neighboring countries with similar climatic and geographical conditions or using aggregated ratios from comparable regions.*

– Section 2.3.3 and 2.3.8: The refinement of fertilizer application in the second approach requires further explanation. Did the authors multiply the rate by the percentage of treated area? Please clarify if cropland was only partially fertilized/manured, while pasture was 100% fertilized/manured in the second approach. I do not think these two fertilizer and manure datasets are two independent datasets. The one without considering the percentage of treated area is a biased estimate since it does not account this factor. Additionally, as the authors considered the percentage of treated area, the cropland (grid) that receives fertilizer would have greater surplus than the other cropland. Have the authors considered to allocate the fertilizer/manure only to those treated area?

**Reply**: Thank you for your detailed feedback on the allocation of fertilizer and manure in our study. We appreciate your observations and would like to clarify a few aspects regarding the distribution of these inputs to croplands and pastures.

– **Fertilizer Application (Section 2.3.3) :** First of all, we would like to clarify that we considered that fertilizer is applied to 100% of the cropland and pasture, since we did not have more detailed data to determine the spatial variability of fertilizer application rates within a given country. We have clarified this in the revised manuscript in line 271 in section 2.3.3. In addition, in our study, the term **"treated"** is only applied to manure inputs and refers to the share of manure processed through specific management systems (e.g., lagoons, slurry, solid storage) as per FAOSTAT (FAOSTAT, 2023) and the Intergovernmental Panel on Climate Change (IPCC) guidelines (Dong et al., 2020). This concept of treated vs. non-treated does not apply to fertilizers, which are distributed to cropland and pasture areas based on two approaches to account fo underlying uncertainty in application rates. These approaches are elaborated below:

  • **First approach:** In this method, we determined country-specific fertilizer application rates for various crops and grassland using data from the International Fertilizer Industry Association (IFA; https://www.ifastat.org), combined with FAOSTAT's cropland and grassland area statistics. To capture spatial variations, we applied these rates for different crops and pasture to gridded respective areas over the period from 1850 to 2019. This approach provided annual fertilizer application amounts for each crop type (non-fodder and fodder), pastures, and the overall total for each grid cell. Next, the fertilizer application totals were adjusted to ensure consistency with the FAO country-level fertilizer amounts applied to soil during $1850 - 2019$.

  • **Second approach:** To refine the distribution of fertilizer, we considered country-specific data that provides proportion of fertilizer applied to cropland and pasture areas, as reported by Ludemann et al. (2023). This approach follows national-level statistics, which indicate that in some countries, not all fertilizer is applied to croplands and pastures. For instance, a majority of the European countries apply 100% of their fertilizer to croplands, while the proportion differs for a few European countries (e.g., 90% for Austria, Finland, France, Germany, the Netherlands, and Poland)

– **Manure Application (Section 2.3.8):** The term **"treated manure"** refers to the manure processed through different manure management systems (MMS), such as lagoon or slurry storage, based on FAOSTAT and IPCC guidelines. It represents manure that is applied to soils after treatment, excluding manure left on pastures or used as fuel. We would like to clarify that we assume that all of the manure treated within a given grid cell is applied to soils within that grid cell. Therefore, all cropland and pasture areas located in grid cells where the manure treated is not equal to zero receive manure. We used three methodologies for manure distribution between cropland and pasture within each grid cell:

  • **Equal distribution:** Manure is distributed evenly between cropland and pasture within each grid cell, following Xu et al. (2019).

  • **Country-specific proportions:** Using country-level data from Ludemann et al. (2023), we applied national ratios to adjust manure distribution. Some countries apply nearly all manure to croplands (e.g., 100% for several European countries), while others allocate a portion to pastures.

- **Time-varying data:** We also used dynamic country-specific data from Einarsson et al. (2021) to assign manure based on actual practices in each country over time. This method captures the evolving practices of manure application across croplands and pastures, accounting for historical changes.

By employing two distinct approaches for fertilizer and three for manure allocation, we provide a comprehensive representation of uncertainties, allowing users to compare, for example, the simplistic uniform distribution to a more refined national data scenarios. We have also revised the manuscript in line 427 of Section 2.3.7 to clarify that the term "treated" refers specifically to manure-related datasets as per FAOSTAT (FAOSTAT, 2023) and IPCC guidelines (Dong et al., 2020), and not to other aspects (e.g., fertilizers, croplands, or pasture areas), in order to avoid potential confusion, which reads as follows:

*Specifically, from FAOSTAT, we used the 'Treated manure N' estimates, which represent the quantity of manure processed through specific manure management systems (e.g., lagoons, slurry, solid storage) prior to N loss in these systems (FAOSTAT, 2023). Since P losses in these systems are minimal (FAOSTAT, 2023), we considered that the entire amount of treated P manure is applied to soil. It is important to clarify that in this context, the term 'Treated' refers exclusively to manure management and does not extend to fertilizers, which are directly distributed to cropland and pasture areas without similar classification.*

**1.2** — Concerns regarding time-series reconstruction and spatial allocation:

The reconstruction of time-series spatial maps presents several issues, particularly when relying on a single reference year for spatial distribution. This approach is problematic for periods before 1961 due to a lack of country-level control data. I recommend trimming the study period to 1961–2019, as the 1850–1960 period is based on unsupported assumptions. The extrapolation lacks the necessary historical data and should be omitted unless stronger justifications can be provided.

**Reply**:

We appreciate the reviewer's concern regarding the reliability of the time-series reconstruction for the period 1850–1960. While we acknowledge the limitations of available data for this early period, we believe that retaining the full dataset from 1850 to 2019 is helpful for several reasons:

- **Use of Historical and Proxy Data**: Although pre-1961 data are limited, we have employed reliable sources (e.g., Cordell et al. (2009), Bayliss-Smith and Wanmali (1984)) to infer reasonable temporal dynamics. Specifically, we used global phosphate rock production data (Cordell et al., 2009) to estimate P fertilizer inputs and historical wheat yield trends (Bayliss-Smith and Wanmali, 1984) as a proxy for other crops. While detailed, spatially explicit data for this period are unavailable, these proxies provide a robust basis for capturing temporal trends and aligning with established historical patterns. We acknowledge the uncertainties inherent in using proxy data but emphasize that these methods offer the most reliable approach for reconstructing historical patterns given the limitations of existing dataset.

- **Historical Context and Long-term assessment**: Including the period from 1850 allows us to capture pivotal shifts in agricultural practices, land use, and industrial development that directly influenced phosphorus (P) dynamics prior to the Green Revolution (Guejjoud et al., 2023). These early changes laid the groundwork for modern nutrient management (Pratt and El Hanandeh, 2023; Ringeval et al., 2024; Sharpley et al., 2013). If we were to omit the 1850–1960 period, we would lose critical insights into the pre-industrial and early industrial phases of agricultural intensification, which are essential for understanding how historical P inputs have shaped current nutrient surpluses – albeit at a crude level, it is still useful for analyzing and understanding long-term developments at regional scales.

- **Phosphorus Legacy and Policy Implications**: P applied during historical periods continues to influence present-day P dynamics due to legacy effects (McCrackin et al., 2018; Guejjoud et al., 2023; Sharpley et al., 2013). By excluding historical database, we would risk overlooking the long-term environmental consequences of historical P accumulation leading to eutrophication and nutrient runoff, which remain critical issues today. Understanding the legacy of early P inputs is essential for designing policies aimed at mitigating both historical and current P surpluses (McCrackin et al., 2018; Sharpley et al., 2013).

- **Cross-Disciplinary Relevance**: Retaining the period from 1850–1960 enhances the dataset's utility for cross-disciplinary studies, including historical agriculture, environmental change, and biogeochemical cycling. Research into the historical impacts of land use, industrialization, and nutrient legacies often requires long-term datasets. As stated above, while the dataset prior to 1961 represents crude estimates, we believe it will still useful for analyzing and understanding long-term developments at regional scales.

While we recognize the uncertainty associated with the early period, we emphasize that the historical trends from 1850 onward provide valuable context for understanding long-term P dynamics. Nevertheless, we understand the reviewer's concern and have issued a cautionary note accordingly. In section 4 of the revised manuscript, we have made it clear that the 1850–1960 period is based on a combination of proxy data and extrapolations, and we recommend that these early estimates be interpreted with caution. We have balanced the text by detailing cautionary notes (at line 849) and the utility on the use our datasets (at line 910) as follows:

*First, the reconstruction of P surplus before 1961 is constrained by limited historical data. For the period 1850–1960, we relied on proxy information and extrapolations based on data from 1961 onward, which inherently introduces higher uncertainty. For instance, national wheat production trends were used to estimate other crop productions, but this method may not fully capture the variations of each specific crop types. We assume that the relative values of the fertilizer application rates taken from the International Fertilizer Association (IFA) (Heffer et al., 2017) for 2014–2015 remained constant for the period 1850–2019, and pre-1961 temporal variations were inferred from global phosphate rock production trends. Livestock distributions were based on GLW3 (Gilbert et al., 2018) data circa 2010 to estimate manure production for the entire 1850–2019 period, and simplifying assumptions were made before 1961, since no country-level manure data were available. Such simplifications may not accurately reflect historical livestock numbers or distribution patterns, influencing P surplus estimates. Furthermore, spatial datasets, especially for land use and crop production, are more detailed and reliable from the mid-1990s onward, making the P surplus estimates more robust for recent decades. Thus, while historical estimates provide general trend insights, recent data (from the mid-1990s) offer greater reliability.*

*Although pre-1961 data are limited, we have employed reliable sources (e.g., Cordell et al. (2009), Bayliss-Smith and Wanmali (1984)) to infer reasonable temporal dynamics. By covering the period from 1850, our dataset captures pivotal shifts in agricultural practices, land use, and industrial development that directly influenced phosphorus (P) dynamics prior to the Green Revolution (Guejjoud et al., 2023). These early changes laid the groundwork for modern nutrient management (Pratt and El Hanandeh, 2023; Ringeval et al., 2024; Sharpley et al., 2013), making the dataset useful not only for current policy analysis but also as a historical baseline for exploring how shifts in climate and agricultural practices affect nutrient cycles over time. Coupled with our nitrogen (N) surplus dataset (Batool et al., 2022), this dataset enables integrated nutrient studies, facilitating the development of comprehensive management strategies that support both P and N sustainability goals. Additionally, the dataset's detailed historical record could support climate adaptation studies, enabling stakeholders to examine how nutrient budgets respond to evolving climate conditions and assess the long-term sustainability of various agricultural practices under changing environmental conditions.*

**1.3** — Additionally, a limitation section addressing these issues is highly recommended. Cropland and pasture: The authors used cropland and pasture distributions circa 2000 from Ramankutty et al. for 1850-2019. There is no country-level data control before 1960.Non-agriculture: The ratios of these non-agricultural area in each grid cell from GCL circa 2000 were

used for 1850-2019, again with no supporting data before 1960. Crop-specific harvest area: the distribution of specific crops from Monfreda et al circa 2000 was used for 1850-2019. There is a lack of country-level data control before 1960.Fertilizer: Crop-specific fertilizer use was derived from IFA circa 2014-2015 and was rescaled throughout 1961-2019. The temporal trend before 1961 was based on global fertilizer production. There is a no country-level data control before 1960. Manure: The animal distribution was based on GLW3 circa 2010 for 1850-2019. Crop production: The annual trend of production for all other crops was based on wheat production, which is not reasonable. P removal by pasture: The P removal was calculated by multiplying 0.6 (or even using NUE) with P input. The removal of P is more likely influenced by herd size and grazing frequency rather than the P input.

**Reply**:

Thank you for the valuable feedback on the potential and limitations of the dataset. We have carefully considered each of the points raised by the reviewer and addressed them in the revised manuscript. Additionally, we have incorporated a new section that highlights both the limitations and future avenues for improving the dataset. Below, we provide responses to the specific points raised:

- **Cropland and Pasture (1850-2019)**: We acknowledge the limitation in using (Ramankutty et al., 2008)'s cropland and pasture distribution circa 2000 for the entire period from 1850 to 2019, especially in the absence of pre-1960 country-level data. This assumption introduces uncertainty in the spatial distribution of agricultural land before 1960. In the limitation section of the manuscript, we now emphasize that the pre-1960 estimates should be interpreted with caution due to the lack of direct historical land-use data. We also suggest future efforts focus on integrating more granular historical data sources to refine the estimates for earlier periods.

- **Non-Agricultural Areas (1850-2019)**: Similar to the cropland and pasture data, the use of GLC (Global Land Cover) ratios circa 2000 for non-agricultural areas introduces limitations, especially before 1960. We recognize that non-agricultural areas, such as forests and urban zones, likely underwent significant changes over the historical period that are not fully captured in our dataset. This limitation has been highlighted in the manuscript, and we propose that future work consider improved historical land-use reconstructions.

- **Crop-Specific Harvest Areas (1850-2019)**: The use of Monfreda et al.'s crop-specific harvest areas circa 2000 for the entire period similarly introduces uncertainty before 1960. We acknowledge that this method does not account for shifts in crop distributions and varieties over time, which could affect the accuracy of P surplus estimates. This limitation has been explicitly mentioned in the revised manuscript.

- **Fertilizer Use (1850-2019)**: The derivation of crop-specific fertilizer use from IFA data for 2014-2015, rescaled to cover 1850-1960, introduces uncertainty for the period prior to 1961 when global dynamics of phosphate rock production were used. While we agree that there is no country-level data control before 1960, we believe that using global phosphate rock production trends is a reasonable approximation given the lack of alternative data. Nevertheless, we have added this as a limitation and encourage future work to focus on refining early-period fertilizer estimates with more historical data.

- **Manure Production (1850-2019)**: The use of livestock distribution from GLW3 circa 2010 to estimate manure production for the entire 1850-2019 period poses challenges, particularly before 1960 when detailed livestock data are sparse. This simplification may not accurately reflect historical herd sizes or distribution patterns, which could influence P surplus estimates. We have explicitly noted this in the limitations section and recommend future studies incorporate more detailed historical livestock data where available.

- **Crop Production (1850-1960)**: We acknowledge that using wheat production as a proxy for other crops may not fully capture the nuances of different crop production systems. However, in the absence of detailed crop-specific

410 production data before 1961, this approach provides a useful, albeit simplified, estimation. To assess the applicabil- ity of this method, we analyzed the temporal alignment between wheat production and other crop categories for the period 1961-2019 using scatter plots and correlation coefficients for the EU28 region. These analyses, presented in Figure R1 (Supplementary Figure S2), showed reasonable correlations for most crops, supporting the use of wheat production dynamics as a proxy during the reconstruction period. Nonetheless, variations in correlation strength
415 among crops highlight the potential for refinement by incorporating additional crop-specific data where available. We have highlighted this limitation in the manuscript and suggest that future work could improve upon this method by incorporating additional crop data where possible.

– **P Removal by Pasture**: We agree with the reviewer that P removal from pasture is influenced by factors beyond P input, such as herd size and grazing frequency. While we based our estimates on phosphorus use efficiency
420 (PUE), we recognize that PUE is an approximation that may not fully capture these complexities. In response, we have revised the manuscript to discuss this limitation and have included it in the expanded limitations section (refers to section 4 of the revised manuscript). Future work should aim to incorporate more dynamic models that account for herd size, grazing intensity, and climatic factors to improve P removal estimates.

In response to the reviewer's suggestion, we have added a dedicated section —*Potential use and limitations of the*
425 *dataset* in the revised manuscript. This section thoroughly explains these issues by clearly outlining the assumptions and uncertainties related to land use, crop production, fertilizer, and manure data. As stated above, we believe that retaining the historical period (1850-1960) provides valuable insights into long-term P surplus trends, but we have issued a cautionary note on the use of early estimates.

To address the concern regarding the use of wheat production as a proxy, Figure R1 (Supplementary Figure S2) has
430 been added to show the relationship for the temporal alignment between wheat production and other crop categories for the period 1961-2019. The moderate to high correlation coefficient ranging from 0.3 to more than 0.9 for different crops, supports the assumption of our methodology for using the wheat production dynamic as a proxy for other crops. Nevertheless, we also acknowledge the fact that further refinements can be done by incorporating additional crop-specific data where available. To ensure clarity, we have also included the following text in Section 2.4.2 at line 569:

435 *The temporal alignment between wheat production and other crop categories was assessed for the EU28 region (Sup- plementary Figure S2) and resulting correlation coefficients were estimated. Most crops showed a reasonable positive correlation with wheat production, ranging from 0.3 to more than 0.9, indicating consistent temporal dynamics across different crop types. These results supports the use of wheat production dynamics as a proxy for other crops during the reconstruction period (1850–1960). However, variations in correlation strength among crops suggest that future refine-*
440 *ments could benefit from incorporating additional crop-specific data where available.*

**1.4** — Other sources

– While the authors aimed to provide a comprehensive phosphorus budget, additional sources of phosphorus emissions, such as those from burning (both agricultural and non-agricultural) and urban phosphorus use (e.g., gardens, golf courses), should be considered. Has phosphorus from fertilizer and manure applied to urban areas or human waste been
445 accounted for? These could be significant sources, and their omission weakens the comprehensiveness of the dataset relative to other inputs like deposition and weathering.

**Reply**: Thank you for your insightful feedback regarding additional sources to account within the phosphorus (P) budget. Regarding the P emissions from burning, we would like to emphasis that they are accounted for in our analysis within the atmospheric deposition component of our dataset. The underlying data explicitly include

450      atmospheric P deposition from various sources such as mineral dust, primary biogenic aerosol particles, sea salt, natural combustion and anthropogenic combustion (e.g. agricultural residue burning, forest fires, logging fires and fossil fuel burning) (Ringeval et al., 2024). To clarify the inclusion of burning-related emissions, we have revised Section 2.3.9 (line 495) as follows:

455      *In our study, we assessed P inputs from atmospheric deposition for different land types, including agricultural land (cropland and pastures) and non-agricultural land. To estimate P deposition for agricultural soils, we used the dataset provided by Ringeval et al. (2024) which represents global atmospheric deposition rates of P to cropland and pasture from 1900 to 2018 at a spatial resolution of 0.5 degrees. This dataset accounts for various sources, including mineral dust, primary biogenic aerosol particles, sea salt, natural combustion, and anthropogenic combustion (e.g., agricultural residue burning, forest fires, logging fires, and fossil fuel burning) (Ringeval et al., 2024).*

460      We agree that urban P inputs, such as those from gardens, golf courses, urban fertilizer use, could be more nuanced and should be considered in a comprehensive P budget. In the current version of our dataset, we have primarily focused on P inputs from agricultural sources (e.g., fertilizer and manure) and natural processes (e.g., atmospheric deposition and chemical weathering) that are generally well-documented at large spatial scales (Panagos et al., 2022; Ringeval et al., 2024).

465      P inputs from urban areas especially those from human waste (i.e., sewage and wastewater) are not accounted for in our analysis, as these often represent point source inputs. Much of this waste often ends up as direct discharges rather than diffuse sources in soil. Our study focuses on characterizing major diffuse sources in the P surplus budget. In parallel, we have also developed a long-term database on nutrient inputs from point sources (urban areas), detailed in a separate study by Sarrazin et al. (2024). By focusing on diffuse P sources, our dataset complements

470      existing datasets that address point source nutrient contributions, such as the European Pollutant Release and Transfer Register (E-PRTR) (Roberts, 2009) and the nutrient load database by Vigiak et al. (2020). Together, these datasets contribute to ongoing efforts to comprehensively understand P dynamics across terrestrial ecosystems, spanning both diffuse and point sources. Moving forward, we plan to explore the integration of different sources in future effort to characterize total P inputs to terrestrial system.

475      To clarify these distinctions, we have revised relevant sections of the manuscript to highlights the focus on diffuse sources as follows.

In Abstract as: *This study reconstructs and analyzes the annual long-term P surplus for both agricultural and non-agricultural soils from diffuse sources across Europe at a 5 arcmin ($\approx$ 10 km at the equator) spatial resolution from 1850 to 2019.*

480      In the Introduction (line 64), we have emphasized the scope of the study: *To address these limitations, we present here a database of yearly long-term P budgets, termed "P surplus" - defined as the difference between P inputs (mineral fertilizer, animal manure, atmospheric deposition and chemical weathering) and P removals (crop and pasture removals), covering both agricultural (cropland and pastures) and non-agricultural soils at a 5 arcmin (1/12$^\circ$; approximately 10 km at the equator) spatial resolution from 1850 to 2019 across Europe, focusing only on diffuse*
485      *sources.*

Finally, we have added the following paragraph in Section 5 (line 947) to discuss on point sources in more detail:

*P inputs from urban areas especially those from human waste (i.e., sewage and wastewater) are not accounted for in our analysis. These inputs are typically classified as point sources, with much of the waste directly discharged rather than contributing to diffuse soil inputs. Our study focuses on characterizing major diffuse sources in the P*
490      *surplus budget. In parallel, we have also developed a long-term database on nutrient inputs from point sources (urban areas), detailed in a separate study by Sarrazin et al. (2024). Additional datasets, such as the European Pollutant Release and Transfer Register (E-PRTR) (Roberts, 2009) and the nutrient load database by Vigiak et al. (2020), provide valuable information on nutrient contributions from urban and industrial sources across Europe. Our current dataset on diffuse sources of P complements existing datasets on point sources and contributes to*
495      *ongoing efforts to comprehensively understand P dynamics across terrestrial ecosystems, spanning both diffuse*

*and point sources. Integrating these point sources with our existing dataset can provide a more comprehensive characterization of P inputs to receiving water bodies (e.g., rivers, lakes, wetlands, etc.) from different sources.*

**1.5 — Show and publish inputs and outputs:**

– The phosphorus surplus represents the balance between inputs and outputs. I recommend including the temporal and spatial changes of individual input and output categories alongside the surplus to help readers understand the drivers of these trends. Additionally, publishing the input and output datasets would be valuable for broader research use.

**Reply**: Thank you for the valuable suggestion. We agree that providing both temporal and spatial changes for P inputs and outputs alongside P surplus greatly enhances the interpretation of the data.

To address this comment, we have expanded the dataset to include not only P surplus but also the underlying P inputs and outputs for the entire study period. This allows for a further exploration of how specific input and output categories drive P surplus trends. These datasets are made publicly available to support further research and detailed analysis of changing P dynamics across Europe.

In the revised manuscript, we have added new visualizations to better represent these dynamics. Figure R2 (which corresponds to Figure 2 in the revised manuscript) now includes gridded maps for P inputs and outputs alongside P surplus, providing a comprehensive spatial overview. The snapshots of variations in mineral fertilizer and animal manure have been provided in Figure R4 (which corresponds to Supplementary Figure S3). Additionally, Figure R3 (which corresponds to Figure 4 in the revised manuscript) shows the decadal trajectories of agricultural and total P inputs (orange) and outputs (blue), with the P surplus represented by a red line for each decade. Figure R5 and R6 (which corresponds to Supplementary Figures S8 and S9) depict decadal P input and output trajectories for individual countries, illustrating how these factors evolve over time and contribute to P surplus trends.

In this regard, we have revised additional text in the revised manuscript at lines 597-650 in section 3.1.

[revised manuscript text omitted]

These enhancements help clarify the major drivers of P surplus and provide better insights into the evolving P dynamics in Europe. By making both the datasets and visualizations publicly accessible, we hope to offer the research community a valuable resource for understanding the interplay between P inputs and outputs across time and regions.

580 **1.6** — Technique corrections:

– Line 49: "difference" should be "different".

**Reply**: Thank you for pointing this out. We have corrected the typo in Line 49 by replacing "difference" with "different" as suggested.

– Equation 12 and 13: Wrong equations.

585        **Reply**: We have reviewed and corrected Equations 12 and 13 in the manuscript to ensure they accurately reflect the intended calculations. The errors in these equations have been rectified, and the revised versions are now correctly presented.

**References**

[revised manuscript text omitted]

**Figure R7.** Cumulative total P surplus, P inputs, and P outputs over four historical periods across Europe. The first row shows the accumulated phosphorus (P) surplus, the second row displays P inputs, and the third row illustrates P outputs across Europe for four distinct periods, which we term as following: (i) 1850–1920 (Pre-modern agriculture), (ii) 1921–1960 (Industrialization before the Green Revolution), (iii) 1961–1990 (Green Revolution and expansion of synthetic fertilizers), and (iv) 1991–2019 (Environmental awareness and policy intervention phase). All values are normalized per year within each time period, with units in tonnes per year.

[Figure]

**Figure R8.** Time series of phosphorus (P) inputs from fertilizer and manure, P outputs, and P surplus ($kg\ ha^{-1}$ of physical area $yr^{-1}$) across various European countries from 1850 to 2019. This figure highlights changes in P fluxes over time, showing a peak in P surplus around 1980 followed by a decline after 1990 for most countries. These patterns illustrate the influence of agricultural intensification during the Green Revolution, as well as subsequent policy, economic, and environmental shifts in both Western and Eastern Europe. The red line represents the mean of 48 P surplus estimates, while green, yellow, and blue lines depict temporal dynamics for P inputs from fertilizer, manure, and P outputs, respectively. Together, these pattern provide insight into how various factors may have influenced P surplus dynamics over time.

---

## Author Response (AR2)

**Editor Comments and Responses:**

**1.1** — Please address the comments from Reviewer 2 and 3.

**Reply**: Thank you for the opportunity to revise our manuscript. We have fully addressed the comments raised by Reviewer #2. In particular, we have clarified the scope regarding the use of "P surplus" across different land types and provided explanations for the importance of P inputs from non-agricultural areas. Detailed responses to these comments are included below. Additionally, we note that Reviewer 3 did not provide any new comments during this revision.

**1.2** — Please also perform a formatting and consistency check for ESSD Journal:

Manuscript format:

– Verified compliance with the ESSD guidelines for structure, font, and referencing style.

**Reply**: The manuscript has been reviewed to ensure compliance with ESSD guidelines for structure, font, and referencing style.

– All sections (Title, Abstract, Introduction, Methods, Results, Discussion, Conclusions, References) are correctly formatted.

**Reply**: All sections, including Title, Abstract, Introduction, Methods, Results, Discussion, Conclusions, and References, are correctly formatted.

Figures and Tables:

– Ensured that all figures are high resolution.

**Reply**: All figures have been checked and verified to be of high resolution.

– Checked that tables are formatted with appropriate headers and units. .

**Reply**: Tables are appropriately formatted with headers and units clearly defined.

Dataset Compliance:

– Verified that the dataset adheres to FAIR principles (Findable, Accessible, Interoperable, Reusable).

**Reply**: The dataset has been reviewed to ensure adherence to the FAIR principles (Findable, Accessible, Interoperable, Reusable).

– Added README files in the dataset repository for clarity.

**Reply**: README files have been added to the dataset repository to enhance clarity and usability.

References:

30     – Cross-checked all citations in the text with the reference list.

**Reply**:

All in-text citations have been cross-checked against the reference list to ensure accuracy.

– Reformatted references to match ESSD citation style.

**Reply**: References have been reformatted to match the ESSD citation style.

35     Supplementary Materials:

– Organized supplementary materials, ensuring all files are properly labeled and linked in the manuscript.

**Reply**: Supplementary materials have been verified to ensure proper labeling and linking within the manuscript.

**Reviewer #2 Comment and Response:**

40    **Comment**: I still feel it is weird to mix P inputs on non-agricultural land and P surplus on agricultural land together. The contribution of non-agricultural P surplus to the total P surplus cannot prove the importance of P weathering and deposition on non-agricultural land. The so-called P surplus on non-agricultural land is usually efficiently used by ecosystems and may not cause serious P pollution. The high contribution of P surplus on non-agricultural land may mainly result from the large area of non-agricultural land.

45    **Reply**:

Thank you for raising this point. While we understand that P weathering and deposition on non-agricultural land are often efficiently utilized by natural ecosystems and may not directly contribute to significant P pollution, our objective here is to provide a comprehensive overview of the P surplus budget across all landscapes, including both agricultural and non-agricultural areas. This broader perspective is essential for understanding not only the dynamics of P inputs and

50    outputs across diverse landscapes but also the fate of P surpluses in receiving water bodies (e.g., groundwater, lakes, rivers) for effective water quality assessments. Such assessments, particularly conducted at catchment or basin scales, require P surplus data from all areas and sources — not just agricultural land — to reliably quantify and analyze total catchment P export.

Moreover, in response to the reviewer's previous suggestion, we have now explicitly provided the different components

55    of the gridded P surplus budget, including P inputs (e.g., mineral fertilizers, manures), P outputs, and the resulting P surplus. In this context, our databases are designed to be flexible and user-oriented, enabling users to conduct analyses based on their specific objectives – for example, for estimation of the P surplus with components focusing on the components that are dominant in agricultural areas.

We believe our comprehensive approach addresses the reviewer's concerns and enhances the utility of our dataset for

60    a wide range of applications. We have added the following texts in the revised manuscript to further emphasis on these points (in section 7 at line 1008 and 1018). Thank you once again for your constructive feedback.

*With a broader aim of analyzing P dynamics across diverse land use types, the dataset highlights the importance of understanding both inputs and outputs across all land types, including areas where agricultural P sources are not dominant. Specifically, studies on water quality assessment at a catchment scale require P surplus data from both agricultural*

65    *and non-agricultural areas to quantify and analyze total catchment P export.*

*Furthermore, by providing detailed components of the gridded P surplus budget—including P inputs (e.g., mineral fertilizers, manures) and P outputs — our databases are designed to be flexible and user-oriented. This flexibility, for example, enables users to conduct P surplus analysis focusing on the components that are dominant in agricultural areas.*